# Commensal yeast promotes *Salmonella* Typhimurium virulence

Kanchan Jaswal[1], Olivia A. Todd[1], Roberto C. Flores Audelo[1], William Santus[1,2], Saikat Paul[3], Ciera M. Duffy[1], Edward T. Eshoo III[1], Manmeet Singh[4], Jian Miao[5], David M. Underhill[6,7], Brian M. Peters[3,8] & Judith Behnsen[1]✉

Enteric pathogens engage in complex interactions with the host and the resident microbiota to establish gut colonization[1–3]. Although mechanistic interactions between enteric pathogens and bacterial commensals have been extensively studied, whether and how commensal fungi affect enteric infections remain largely unknown[1]. Here we show that colonization with the common human gut commensal fungus *Candida albicans* worsened infections with the enteric pathogen *Salmonella enterica* subsp. *enterica* serovar Typhimurium. The presence of *C. albicans* in the mouse gut increased *Salmonella* caecal colonization and systemic dissemination. We investigated the underlying mechanism and found that *Salmonella* binds to *C. albicans* via type 1 fimbriae and uses its type 3 secretion system to deliver effector proteins into *C. albicans*. A specific effector, SopB, was sufficient to manipulate *C. albicans* metabolism and trigger the release of millimolar amounts of arginine into the extracellular environment. The released arginine, in turn, induced expression of the type 3 secretion system in *Salmonella*, increasing its invasion of epithelial cells. *C. albicans* deficient in arginine production was unable to increase *Salmonella* virulence. Arginine-producing *C. albicans* also dampened the inflammatory response during *Salmonella* infection. Arginine supplementation in the absence of *C. albicans* increased the systemic spread of *Salmonella* and decreased the inflammatory response, phenocopying the presence of *C. albicans*. In summary, we identified *C. albicans* colonization as a susceptibility factor for disseminated *Salmonella* infection and arginine as a central metabolite in the cross-kingdom interaction between fungi, bacteria and host.

Decades of research have illuminated the central role of the gut microbiome for human health. Among a multitude of functions, gut microorganisms provide colonization resistance to pathogens[2–4], train the immune system[5,6], aid in digestion[7] and modulate distal organ functions via microbial products[8–10]. Gut bacteria are the most abundant members of the gut microbiome and have been the focus of mechanistic research. Conversely, our knowledge on the roles of other members of the gut microbiome, such as viruses and fungi, is still lacking[1]. Abundance and composition of the fungal microbiome (mycobiome) is greatly altered in multiple gastrointestinal diseases[11–13]. However, it is largely unknown how fungi metabolically integrate into the gastrointestinal environment and interact with commensal and pathogenic bacteria[14].

Some of the best-studied enteric pathogens are non-typhoidal *Salmonella*, which infect an estimated 100 million individuals per year globally[15]. In healthy individuals, non-typhoidal *Salmonella*, such as *Salmonella* Typhimurium (*Salmonella* or STm), cause a localized infection of the gastrointestinal tract, resulting in inflammatory diarrhoea[16,17]. In immunocompromised individuals, *Salmonella* can

disseminate to peripheral organs causing potentially fatal disease[16,17]. To establish gut colonization, *Salmonella* must compete with resident microorganisms. Even though commensal fungi are found in all tested mammalian species[18], studies have predominantly focused on the role of gut-resident bacteria. The role of the mycobiome during infection with enteric pathogens has been largely unexplored.

One of the most prominent fungal colonizers of human mucosal surfaces is *C. albicans*[19]. Recent studies have determined *C. albicans* to be present in the gut of more than 60% of healthy humans[20,21]. Although usually a commensal[22,23], *C. albicans* can become pathogenic, particularly in immunocompromised patients[19,24,25]. An important virulence mechanism of *C. albicans* is the ability to switch morphology from rapidly growing yeast to epithelium-penetrating hyphae[25]. *C. albicans* is associated with inflammatory bowel disease, specifically Crohn's disease[19,24,25]. Although *C. albicans* cannot induce gut inflammation, it has been shown to exacerbate it[26,27]. *Salmonella* and *C. albicans* therefore both thrive under inflammatory conditions in the gut, and both have high pathogenic potential. *C. albicans* represents an important

[1]Department of Microbiology and Immunology, University of Illinois Chicago, Chicago, IL, USA. [2]Department of Biological Sciences, University of Illinois Chicago, Chicago, IL, USA. [3]Department of Clinical Pharmacy and Translational Science, College of Pharmacy, University of Tennessee Health Science Center, Memphis, TN, USA. [4]Department of Pathology, University of Illinois Chicago, Chicago, IL, USA. [5]Pharmaceutical Sciences Program, College of Graduate Health Sciences, University of Tennessee Health Science Center, Memphis, TN, USA. [6]Department of Biomedical Sciences, Cedars-Sinai Medical Center, Los Angeles, CA, USA. [7]F. Widjaja Inflammatory Bowel and Immunobiology Research Institute, Cedars-Sinai Medical Center, Los Angeles, CA, USA. [8]Department of Microbiology, Immunology, and Biochemistry, College of Medicine, University of Tennessee Health Science Center, Memphis, TN, USA. ✉e-mail: jbehnsen@uic.edu

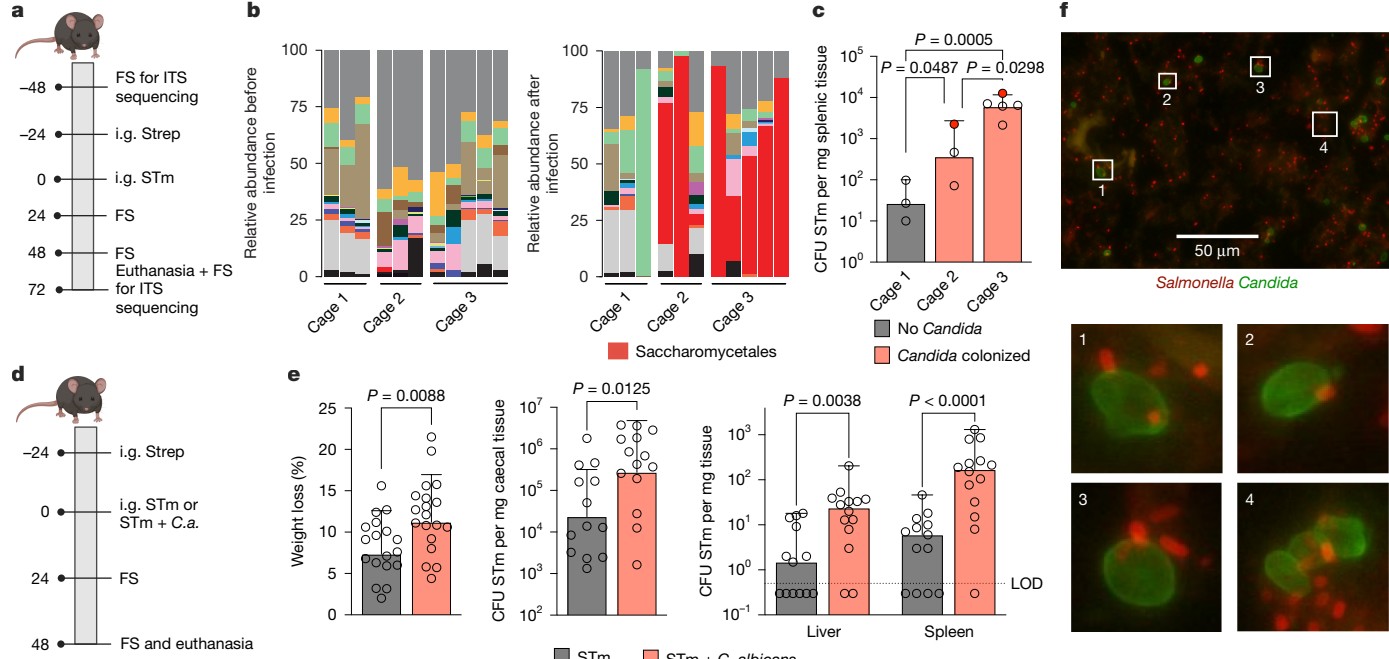

**Fig. 1 | *C. albicans* increases *Salmonella* colonization and dissemination.**
**a**, Schematic representation of the experimental set-up for sequencing analysis.
*C.a.*, *C. albicans*; FS, faecal sample; i.g., intragastrically; strep, streptomycin.
**b**, Relative abundance of fungal genera identified with ITS sequencing in faecal
samples of mice before and after STm infection. *n* = 3 (cages 1 and 2) and *n* = 5
(cage 3) animals. **c**, STm colonization in the spleen 72 h post-infection. Mice
with reads for *C. albicans* before infection are represented as red circles. Data
are geometric mean ± s.d. *n* = 3 (cages 1 and 2) and *n* = 5 (cage 3) animals. Ordinary
one-way analysis of variance (ANOVA) for comparison was used. CFU, colony-
forming units. **d**, Schematic representation of the experimental set-up for STm
or STm + *C. albicans* ATCC infection. The schematics in panels **a**,**d** were created
in BioRender. Behnsen, J. (2025) https://biorender.com/xv05v3d. **e**, Weight

loss and STm colonization in C57BL/6 mice infected with STm or STm + *C. albicans*
ATCC in the streptomycin pre-treatment model at 48 h post-infection. Data are
geometric mean ± s.d. (for weight loss and caecum) and median (for liver and
spleen). *n* = 13 (for STm: caecum, liver and spleen), *n* = 14 (for STm + *C. albicans*:
caecum, liver and spleen), *n* = 18 (for STm: weight loss) and *n* = 19 (for STm +
*C. albicans*: weight loss) animals from three independent experiments.
Significance was determined by two-tailed Mann–Whitney test (for weight loss
and caecal colonization) and mixed-effect analysis with Šídák's multiple
comparisons test (for liver and spleen dissemination). LOD, limit of detection.
**f**, Representative fluorescence image of the lumen of the caecum during mouse
infection with STm and *C. albicans* SC5314. The experiment was repeated
independently with six mice.

human gut mycobiome member and is potentially present in the gut
of a substantial number of patients when they become infected with
*Salmonella*.

Our investigation on mechanistic interactions between *Salmonella*
and *C. albicans* supports an important role for *C. albicans* carriage
in the pathogenesis of *Salmonella* infection and delineates intricate
fungal–bacterial–host crosstalk in the gut.

## *C. albicans* enhances *Salmonella* virulence

Inflammatory diseases, such as inflammatory bowel disease, have often
been associated with changes in the abundance or composition of
the fungal microbiome. We therefore tested whether infection with
*Salmonella*, an enteric pathogen that causes acute gut inflammation,
is associated with changes in the mycobiome (Fig. 1a). After infection
with *Salmonella*, we observed a marked increase in the abundance of
the order Saccharomycetales (Fig. 1b, red bars, and Supplementary
Table 1), particularly in the genus *Candida* (Extended Data Fig. 1a), in
faeces of mice from two cages. *Candida* spp. were only present in two
mice before infection but showed a high relative abundance in almost all
cage mates after infection (Fig. 1b and Extended Data Fig. 1a, red closed
circles). Faeces of mice from cage 1 did not contain any *Candida* spp.
before or after infection. Usage of an antibiotic in the streptomycin
pre-treatment mouse model[28] probably opened a niche for *Candida* spp.
that is usually occupied by bacteria. Although expansion of *Candida*
spp. and spreading to cage mates was therefore not unexpected, the
serendipitous effect of this expansion on the outcome of *Salmonella*
infection was. Mice in cages 2 and 3 with increased *Candida* spp. relative

abundance showed markedly higher dissemination of *Salmonella* to
the spleen and liver (Fig. 1c and Extended Data Fig. 1b) than mice in
cage 1 that had no increase in *Candida* spp.

The species identified by sequencing in mice was *C. albicans*, which
is a frequent commensal of humans but not a known commensal of
mice. The presence of *C. albicans* was probably an environmental con-
tamination that bloomed in the presence of an antibiotic. Given that
*C. albicans* is an important commensal in humans and seemed to be
modulating *Salmonella* disease severity in mice, we further analysed
this cross-domain interaction. We first determined whether the initial
observation of higher *Salmonella* burden was indeed caused by the
presence of *C. albicans*. *Candida*-free mice received either *Salmonella*
alone or *Salmonella* and *C. albicans* (10:1 *Salmonella* to *C. albicans*
ratio; Fig. 1d,e and Extended Data Fig. 1c–e). Co-infected mice lost 50%
more weight, showed higher *Salmonella* colonization in the caecum
(Fig. 1e) and increased *Salmonella* burden in the spleen and the liver
than mice infected only with *Salmonella* (Fig. 1e). Reproducibility of
the phenotype was strong regardless of sex and infectious dose
(Extended Data Fig. 1f). Conversely, colonization of *C. albicans* was
not significantly affected by the presence of *Salmonella* and the yeast
did not disseminate to peripheral organs (Extended Data Fig. 1e). Imag-
ing showed *C. albicans* predominantly in yeast form in the gut and
confirmed that *Salmonella* and *C. albicans* interact with each other in
the lumen of the caecum (Fig. 1f) and in proximity to colonic epithelial
cells (Extended Data Fig. 1g). In humans, *C. albicans* would be present in
the gut at the time of *Salmonella* infection. We therefore first colonized
CBA/J mice with the *C. albicans* strain 529L[29] and then infected the
mice with *Salmonella* in the absence of antibiotics. Also in this model,

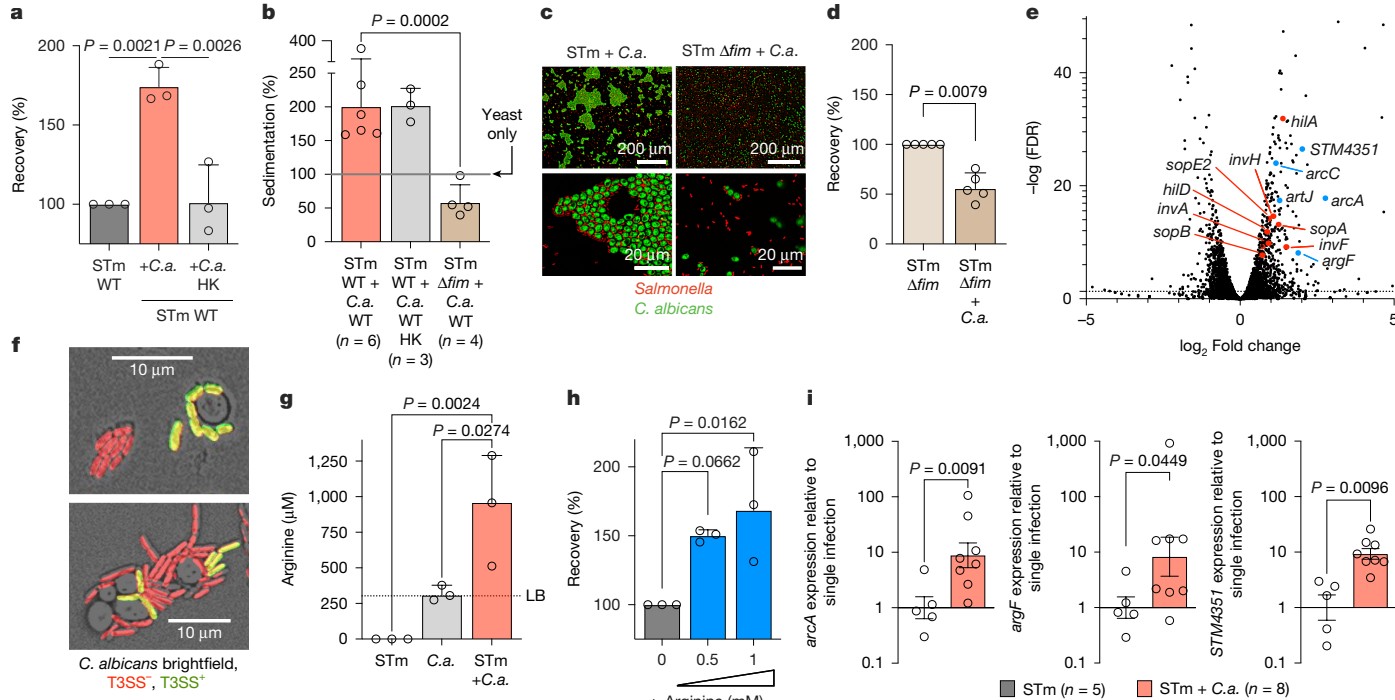

**Fig. 2 | Binding to live *Candida* increases *Salmonella* invasion. a**, Invasion assay of STm infected (multiplicity of infection (MOI) = 1) colonic epithelial cells (T84). STm either alone or with *C. albicans* ATCC in a 10:1 (*Salmonella* to *C. albicans*) ratio were incubated for 2 h before the assay. Data are geometric mean ± s.d.; *n* = 3 independent experiments. Significance was determined using ordinary one-way ANOVA. HK, heat killed. **b**, Sedimentation assay of STm and *C. albicans* SC5314. The line at 100% indicates yeast-only sedimentation. Data are geometric mean ± s.d.; *n* represents independent experiments. Significance was determined using ordinary one-way ANOVA. **c**, Fluorescence image of STm and *C. albicans* SC5314 in vitro. **d**, Invasion assay of STm-infected (MOI = 1) colonic epithelial cells (Caco2). STm either alone or with *C. albicans* ATCC in a 10:1 ratio were incubated for 2 h before the assay. Data are geometric mean ± s.d.; *n* = 5 independent experiments. Significance was determined using a two-tailed Mann–Whitney test. **e**, Volcano plot of differentially regulated STm genes in co-culture with *C. albicans* ATCC compared with STm alone. SPI-1 genes involved in invasion are in red, and genes involved in arginine transport and downstream metabolism are in blue. FDR, false discovery rate. **f**, Fluorescence image of STm expressing mCherry and *Pprgh-gfp*, and *C. albicans* SC5314 (brightfield) in vitro. **g**, Arginine levels in cell-free supernatants of STm, *C. albicans* SC5314 or STm + *C. albicans* SC5314 cultures incubated for 2 h. Data are median with range; *n* = 3 independent experiments. Significance was determined using ordinary one-way ANOVA. The dashed line indicates levels in lysogeny broth (LB). **h**, Invasion assay of STm-infected (MOI = 1) colonic epithelial cells (Caco2). STm either alone or with L-arginine were incubated for 2 h before the assay. Data are geometric mean ± s.d.; *n* = 3 independent experiments. Significance was determined using ordinary one-way ANOVA. **i**, rt-qPCR analysis of STm arginine import and metabolism genes from the caecal content of STm or STm + *C. albicans* ATCC infected mice 48 h post-infection. Data are mean ± s.e.m.; *n* represents mice of two independent experiments. Significance was determined using two-tailed Welch's *t*-test. Bars with no statistics have *P* > 0.9999.

---

the presence of *C. albicans* increased *Salmonella* colonization in the caecum (Extended Data Fig. 1h). However, *Salmonella* dissemination to peripheral organs was not increased, probably due to the lower colonization level of *Salmonella* in the caecum (Extended Data Fig. 1h). The presence of two different *C. albicans* strains thus modulated *Salmonella* colonization or dissemination in two different mouse models.

## Arginine increases *Salmonella* virulence

*C. albicans* may modulate virulence of *Salmonella* through direct contact, secreted molecules or altered immune response to *Salmonella* infection. We first tested whether *C. albicans* modulates gut epithelial cell invasion in vitro (Extended Data Fig. 2a). The presence of live *C. albicans* increased recovery of intracellular *Salmonella* to 180% compared with *Salmonella* alone (Fig. 2a and Extended Data Fig. 2b,c). However, heat-killed *C. albicans* or purified cell wall components (curdlan, that is, β-(1→3)-glucan) did not increase *Salmonella* invasion (Fig. 2a and Extended Data Fig. 2d). Therefore, viable *C. albicans* is required to mediate increased *Salmonella* epithelial cell invasion. Imaging of the gut indicated that *C. albicans* and *Salmonella* directly interact with one another (Fig. 1f and Extended Data Fig. 1g). *Salmonella* bound to both live and heat-killed *C. albicans* yeast cells in vitro, quantitatively measured by higher sedimentation rates (Fig. 2b and Extended Data

Fig. 2e). Binding to yeast and hyphal form was confirmed using fluorescence microscopy (Fig. 2c and Extended Data Fig. 2f). We used a small library of *Salmonella* mutants to identify surface determinants that could mediate binding to *C. albicans* (Extended Data Fig. 2e). A *Salmonella* mutant deficient in type 1 fimbriae, Δ*fim*, was unable to aggregate *C. albicans* (Fig. 2c and Extended Data Fig. 2e). Type 1 fimbriae bind to mannose residues[30] present in the epithelial cell extracellular glycome or mannans present in the cell wall of yeast cells. As expected, binding of *Salmonella* to *C. albicans* was therefore inhibited by the addition of excess mannose (Extended Data Fig. 2e,g). *C. albicans* did not increase invasion of *Salmonella* WT in the presence of mannose or *Salmonella* Δ*fim*, indicating that *Salmonella* binding to *C. albicans* is required to increase *Salmonella* invasion into epithelial cells (Fig. 2d and Extended Data Fig. 2h). Secreted factors could additionally modulate the interaction between *Salmonella* and *C. albicans*. Indeed, co-culture supernatant was sufficient to increase *Salmonella* invasion to 160% (Extended Data Fig. 2i), nearly the same level as exposure to live *C. albicans* (Fig. 2a). However, exposure to the supernatant of a co-culture of *C. albicans* and Δ*fim Salmonella* did not increase wild-type (WT) *Salmonella* invasion (Extended Data Fig. 2i). Therefore, binding of *Salmonella* to *C. albicans* seemed to result in secreted molecules that alter *Salmonella* virulence. This hypothesis was supported by the observation that if *Salmonella* and *C. albicans* were

directly added to epithelial cells without previous contact, *Salmonella* invasion was not increased (Extended Data Fig. 2j).

To determine *Salmonella* genes responsive to *C. albicans*, we compared transcriptomes of monocultures and co-cultures (Supplementary Table 2). KEGG analysis revealed significant regulation of multiple pathways in the presence of *C. albicans*, including genes in 'Salmonella infection' (Extended Data Fig. 3a and Supplementary Table 3). Incubation with *C. albicans* resulted in upregulation of *Salmonella* pathogenicity island 1 (SPI-1) regulator HilA and type 3 secretion system (T3SS) structural proteins and effectors (Fig. 2e and Supplementary Table 2). We confirmed upregulation of *hilA* and *invA*, which encodes a SPI-1 T3SS structural protein (Extended Data Fig. 3b,c). We also visualized SPI-1 induction on a single-cell level, utilizing a GFP reporter construct under the control of the T3SS gene *prgH*. We found a significant increase in T3SS-positive cells in the presence of *C. albicans*, specifically in the cells in proximity to *C. albicans* (Fig. 2f and Extended Data Fig. 3d). Binding of *Salmonella* to *C. albicans* might directly result in increased expression of genes on SPI-1, which are important for *Salmonella* invasion into epithelial cells (Fig. 2a and Extended Data Fig. 2b–d). However, further analysis of RNA sequencing data suggested a more complex regulatory network. Genes involved in arginine uptake and metabolism represented some of the most highly upregulated *Salmonella* genes in the presence of *C. albicans* (Fig. 2e). In our KEGG analysis, arginine and proline metabolism was not significantly regulated (Extended Data Fig. 3a and Supplementary Table 3). However, the upregulated genes presented a specific subset of genes that encoded proteins involved in the catabolism of arginine via the arginine deiminase (ADI) pathway (Extended Data Fig. 3e). *arcA*, encoding the *Salmonella* ADI (and not the oxygen flux sensor with the same identifier) was upregulated 6.8-fold. We confirmed upregulation of *arcA*, *argF* (also annotated as *arcB*) and the arginine transporter subunit *STM4351* (ref. 31) by real-time quantitative PCR (rt-qPCR) using two different *Salmonella* and *C. albicans* strains (Extended Data Fig. 3b,c). As arginine was previously shown to modulate virulence of pathogens, such as *Citrobacter rodentium*[32], we further investigated the role of arginine in the interaction of *Salmonella* and *C. albicans*. Expression of ADI pathway genes is known to be regulated by arginine availability[33,34]. Metabolomic analysis of in vitro cultures indeed revealed significant differences in arginine concentrations. *Salmonella* depleted the arginine available in lysogeny broth within 2 h, whereas growth of *C. albicans* did not change the arginine concentration. During co-culture of *Salmonella* and *C. albicans*, arginine levels increased to 2–4-fold the concentration of arginine in lysogeny broth (Fig. 2g). No other amino acid concentration increased during co-culture (Supplementary Table 4). We next tested whether arginine regulates SPI-1 expression in *Salmonella*. Indeed, addition of L-arginine increased the expression of the SPI-1 regulator *hilA* in a dose-dependent manner (Extended Data Fig. 3f,g) and *invA* showed a trend for higher expression (Extended Data Fig. 3g). Transient exposure to L-arginine also increased invasion of *Salmonella* into epithelial cells in a dose-dependent manner (Fig. 2h and Extended Data Fig. 3h). Contrary to batch cultures in vitro, in the gut, metabolites such as arginine are in constant flux due to metabolism by the microbiome and the host. As observed for other metabolites utilized by *Salmonella* in vivo[35], metabolomic analysis showed no significant change in arginine concentrations in caecal content 24 h and 48 h post-infection (Extended Data Fig. 3i and Supplementary Tables 5 and 6). We therefore assessed whether *Salmonella* was expressing arginine catabolic genes during co-infection and found that *Salmonella* expression of *arcA*, *argF* and *STM4351* was tenfold higher in the caecum of mice when *C. albicans* was present (Fig. 2i). This indicates that *Salmonella* metabolizes arginine in the presence of *C. albicans* in vivo.

## *C. albicans* produces arginine

We next examined the source of the increased arginine concentrations during *Salmonella* and *C. albicans* co-cultures in vitro. As *Salmonella*

was catabolizing arginine in a monoculture (Fig. 2g and Supplementary Table 4), *C. albicans* remained the only possible producer. Indeed, expression of *C. albicans ARG1* and *ARG4* (Extended Data Fig. 4a) was highly increased in the presence of *Salmonella* in vitro (Fig. 3a and Extended Data Fig. 4b). *C. albicans ARG4* expression was also increased in the presence of *Salmonella* in vivo (Extended Data Fig. 4c). We hypothesized that a *C. albicans* mutant deficient in the production of arginine would not increase *Salmonella* expression of SPI-1 genes and its invasion into epithelial cells. A *C. albicans arg4*Δ/Δ strain indeed failed to increase the expression of SPI-1 genes (Extended Data Fig. 4d). The strain was still able to bind to *Salmonella* (Extended Data Fig. 4e) but did not increase *Salmonella* invasion into epithelial cells, whereas the revertant strain expressing *ARG4* increased *Salmonella* invasion to the same extent as WT *C. albicans* (Fig. 3b and Extended Data Fig. 4e). Arginine produced by *C. albicans* was thus required to increase *Salmonella* invasion into epithelial cells in vitro. We also tested whether the *C. albicans arg4*Δ/Δ strain would increase *Salmonella* virulence in vivo (Fig. 3c,d and Extended Data Fig. 4f,g). All *C. albicans* strains colonized the small intestine, caecum and colon or faeces equally well (Extended Data Fig. 4g). WT *C. albicans* and an *ARG4*-revertant strain increased *Salmonella* colonization in the caecum (Fig. 3d). However, in the presence of the *C. albicans arg4*Δ/Δ strain, *Salmonella* caecal colonization did not significantly increase (Fig. 3d). Similarly, WT and *ARG4*-revertant *C. albicans* showed a trend towards higher *Salmonella* dissemination to the spleen, whereas *arg4*Δ/Δ *C. albicans* did not (Fig. 3d). Similar to co-infection with WT *C. albicans* and *Salmonella*, arginine concentrations in vivo did not change (Extended Data Fig. 4h).

In vivo, *C. albicans*-produced arginine could be metabolized by the bacterial microbiota in addition to *Salmonella*. Population-level changes in composition or abundance of microbiota members in response to available arginine could therefore influence the observed phenotype. However, in germ-free mice (Extended Data Fig. 5a–d and Supplementary Table 7), as well as in mice colonized with an eight-member defined microbiota (altered Schaedler flora (ASF); Extended Data Fig. 5e–h and Supplementary Table 8), we observed an increase in *Salmonella* caecal colonization in the presence of *C. albicans* (Extended Data Fig. 5c,g). Experiments with germ-free mice had to be terminated at 24 h and showed no increase in dissemination at 24 h post-infection (Extended Data Fig. 5c), but in ASF mice, dissemination to the spleen and liver (Extended Data Fig. 5g) was increased in the presence of *C. albicans* at 48 h post-infection. We did not observe an increase of luminal arginine in the presence of *Salmonella* and *C. albicans* in germ-free or ASF mice (Extended Data Fig. 5i and Supplementary Tables 7 and 8). To test the effect of *C. albicans* presence and *C. albicans*-produced arginine on the microbiota of conventionally raised mice, we used internally transcribed spacer (ITS) and 16S sequencing (Supplementary Tables 9 and 10). As expected, all *C. albicans*-colonized mice clustered in ITS principal coordinate analysis (PCoA) plots (Extended Data Fig. 5j). In 16S PCoA plots, uninfected mice were separated from *Salmonella*-infected mice. However, *Salmonella*-infected mice did not cluster dependent on whether *C. albicans* was present (Extended Data Fig. 5j). Relative abundance plots indicate the presence of the same microbiota members in similar abundances across groups (Extended Data Fig. 5k). The bacterial microbiota thus does not seem to be significantly changing or contributing to the observed phenotype of enhancing *Salmonella* virulence in the presence of *C. albicans*.

## Effector triggers arginine production

*C. albicans* production of arginine was essential to increase *Salmonella* invasion into epithelial cells in vitro and in vivo. However, it was unclear why *C. albicans* would biosynthesize and release the amino acid when in contact with *Salmonella*. We hypothesized that *Salmonella* directly triggered this response in *C. albicans*. As a eukaryote, *C. albicans* shares many similarities with the mammalian cells that *Salmonella* has evolved

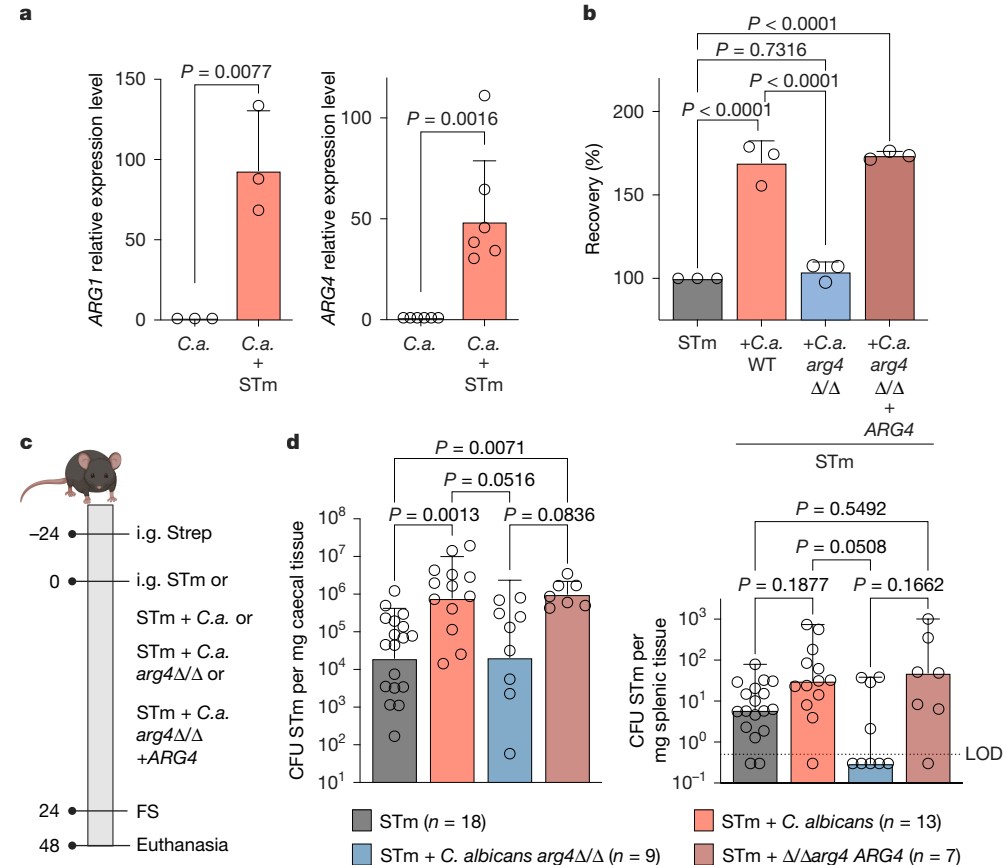

**Fig. 3 | Role of arginine production in *C. albicans* for *Salmonella* virulence. a**, rt-qPCR analysis of genes encoding *C. albicans* arginine biosynthesis from the *C. albicans* SC5314 or STm + *C. albicans* SC5314 cultures incubated for 2 h. Data are geometric mean ± s.d.; *n* = 3 (*ARG1*) and *n* = 6 (*ARG4*) independent experiments. Significance was determined by two-tailed unpaired Student's *t*-test for comparison. **b**, Invasion assay of STm-infected (MOI = 1) colonic epithelial cells (Caco2). STm either alone or with *C. albicans* SC5314 in a 10:1 (*Salmonella* to *C. albicans*) ratio were incubated for 2 h before the assay. Data are geometric mean ± s.d.; *n* = 3 independent experiments. Significance was determined using ordinary one-way ANOVA for comparison. **c**, Schematic

representation of the experimental set-up for STm or STm + *C. albicans* infection. Mice were either gavaged again with *C. albicans* at 24 h post-infection or the day 0 dose of *C. albicans* was increased to 1 × 10⁷ CFU ml⁻¹. The schematic was created in BioRender. Behnsen, J. (2025) https://biorender.com/xv05v3d. **d**, STm colonization in C57BL/6 mice infected with STm or STm + *C. albicans* SC5314 in the streptomycin pre-treatment model for 48 h post-infection. Data are geometric mean ± s.d. (for caecum) and median with range (for spleen) for comparison. *n* represents animals from four independent experiments. Significance was determined by Kruskal–Wallis test for comparison.

to interact with, and *Salmonella* T3SS effectors are functional when exogenously expressed in yeast (for example, *Saccharomyces cerevisiae*)[36]. We therefore tested whether the T3SS-1 that is required for *Salmonella* invasion into epithelial cells was also required to induce the expression of arginine biosynthesis genes in *C. albicans*. We incubated a *Salmonella* Δ*invA* strain, deficient in the assembly of the T3SS-1 needle, with *C. albicans* and tested whether the cell-free supernatant would increase WT *Salmonella* invasion into epithelial cells. This supernatant did not increase invasion of *Salmonella* (Extended Data Fig. 6a). We therefore measured the concentration of arginine and found that only co-cultures of *C. albicans* with WT *Salmonella*, but not with Δ*invA* *Salmonella*, had increased levels of arginine (Fig. 4a). The arginine biosynthesis genes *ARG1* (Fig. 4b) and *ARG4* (Extended Data Fig. 6b) showed high expression in the presence of WT *Salmonella*, modest expression with the *Salmonella* Δ*fim* mutant that cannot aggregate with *C. albicans*, and no increase in the presence of *Salmonella* Δ*invA*, which was able to aggregate with *C. albicans* (Extended Data Fig. 2e). Therefore, the *Salmonella* T3SS-1 was required to increase arginine biosynthesis in *C. albicans*. The *Salmonella* T3SS-1 can secrete many different effector proteins. To determine which effector might elicit the increase in arginine biosynthesis, we tested a mutant deficient in many effectors of T3SS-1 (*sipA*, *sopA*, *sopB*, *sopD* and *sopE2*) and it failed to increase arginine biosynthesis in *C. albicans* (Fig. 4c and Extended

Data Fig. 6c). Further tests with *Salmonella* mutants pinpointed that deletion of a single effector, *sopB*, was sufficient to abrogate induction of *C. albicans* arginine biosynthesis (Fig. 4c and Extended Data Fig. 6c). *Salmonella* Δ*sopB* showed no defect in binding to *C. albicans* (Extended Data Fig. 6d), but the co-culture supernatant did not increase invasion of *Salmonella* and did not contain measurable arginine levels (Fig. 4d and Extended Data Fig. 6e). *C. albicans* also did not increase invasion of a *Salmonella* strain deficient in SopB into epithelial cells (Fig. 4e). In vivo (Fig. 4f and Extended Data Fig. 6f–i), caecal colonization and splenic burden of a *Salmonella* Δ*sopB* strain was not changed in the presence of *C. albicans* WT or *arg4*Δ/Δ (Fig. 4f and Extended Data Fig. 6h). All our experiments were performed with the *Salmonella* strain IR715 (a Nal^R derivative of American Type Culture Collection (ATCC) 14028), but we also used the *Salmonella* SL1344 strain to confirm these findings (Extended Data Fig. 6j–l). All tested *Salmonella* SL1344 strains showed similar binding to *C. albicans* (Extended Data Fig. 6j). However, the function of SopB was redundant in SL1344, as the *Salmonella* SL1344 Δ*sopB* strain induced the same level of expression of *ARG1* and *ARG4* in *C. albicans* compared with WT *Salmonella* (Extended Data Fig. 6k). When we deleted both *sopB* and *sopE*, which has been shown in mammalian systems to have overlapping functions with SopB[37], the strain failed to induce arginine biosynthesis (Extended Data Fig. 6k). Overexpression of *sopB* in the Δ*sopB* strain resulted in three times higher

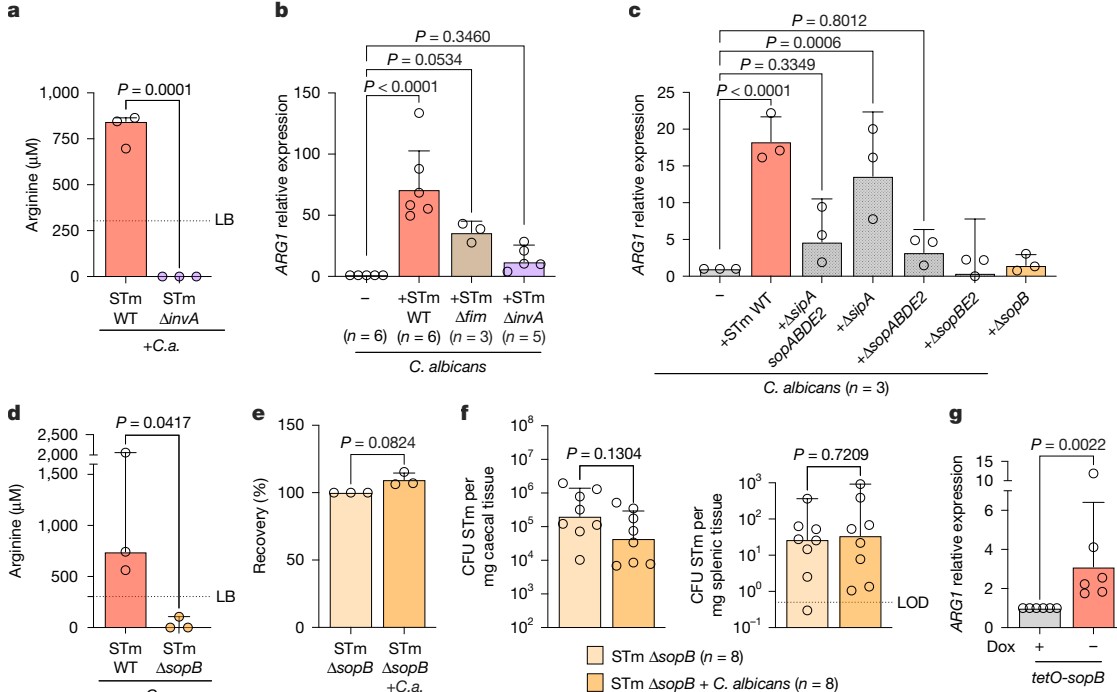

**Fig. 4 | *Salmonella* uses T3SS-1 effector SopB to trigger arginine production in *C. albicans*. a**, Arginine levels measured in cell-free supernatants of STm and *C. albicans* SC5314 cultures incubated for 2 h. Data are median with range; *n* = 3 independent experiments. Significance was determined usng a two-tailed unpaired Student's *t*-test for comparison. **b,c**, rt-qPCR analysis of genes encoding *C. albicans* arginine biosynthesis from the *C. albicans* SC5314 or STm + *C. albicans* SC5314 cultures incubated for 2 h. Data are geometric mean ± s.d. *n* represents independent experiments. Significance was determined by a Kruskal–Wallis test (**b**) and an ordinary one-way ANOVA (**c**) for comparison. **d**, Arginine levels measured in cell-free supernatants of STm and *C. albicans* SC5314 cultures incubated for 2 h. Data are represented as median with range; *n* = 3 independent experiments. Significance was determined by a one-tailed unpaired Student's *t*-test for comparison. **e**, Invasion assay of STm-infected (MOI = 1) colonic epithelial cells (Caco2). STm either alone or with *C. albicans* SC5314 in a 10:1 (*Salmonella* to *C. albicans*) ratio were incubated for 2 h before the assay. Data are geometric mean ± s.d.; *n* = 3 independent experiments. Significance was determined by two-tailed paired Student's *t*-test test for comparison. **f**, STm colonization in C57BL/6 mice infected with STm or STm + *C. albicans* SC5314 in the streptomycin pre-treatment model for 48 h post-infection. Data are geometric mean ± s.d. (for caecum) and median with range (for spleen). *n* represents animals from three independent experiments. Significance was determined using a Kruskal–Wallis test for comparison. **g**, rt-qPCR analysis of genes encoding *C. albicans* arginine biosynthesis from *C. albicans* SC5314 cultures expressing *tetO-sopB* incubated for 6 h with or without doxycycline (Dox). Data are geometric mean ± s.d.; *n* = 6 independent experiments. Significance was determined by a two-tailed Mann–Whitney *t*-test for comparison. Bars with no statistics have *P* > 0.9999. The dashed line indicates levels in LB (**a**,**d**) and LOD (**f**).

expression of *ARG1* and *ARG4 in C. albicans* than WT (Extended Data Fig. 6l). A functional *Salmonella* T3SS-1 is therefore required to increase arginine biosynthesis in *C. albicans* and SopB (and SopE) are the translocated effectors that can trigger this response. We also expressed SopB under control of the doxycycline-repressible *tetO* promoter in *C. albicans* (Extended Data Fig. 6m). Upon removal of doxycycline, *C. albicans ARG1* and *ARG4* expression was significantly induced (Fig. 4g and Extended Data Fig. 6n,o), confirming that the presence of SopB in *C. albicans* results in upregulation of arginine biosynthesis.

## Blunted immune response to co-infection

In addition to direct microbial interactions between *C. albicans* and *Salmonella*, we also found evidence that the presence of *C. albicans* modulates the host immune response to *Salmonella* infection. In the gastroenteritis mouse model (Fig. 5a), infection with *Salmonella* resulted in a strong inflammatory response in the caecum (Fig. 5b and Extended Data Fig. 7a,b). We expected an opportunistic pathogen such as *C. albicans*, which is known to induce IL-17 during candidiasis[38,39], to further increase the inflammatory response. However, the presence of *C. albicans* during *Salmonella* infection resulted in a blunted inflammatory response (Fig. 5b and Extended Data Fig. 7a,b). The decreased response was particularly pronounced early after infection at 24 h (Fig. 5b and Extended Data Fig. 7a) when *Salmonella* had not yet disseminated to the liver and the spleen, and colonization of the caecum

was equal between groups (Extended Data Fig. 7c–e). Expression of genes such as *Il17* and *Cxcl1* was 5–10-fold lower after infection with *Salmonella* and *C. albicans* than infection with *Salmonella* alone (Fig. 5b), with an increase in arginase II (*Arg2*) expression (Extended Data Fig. 8a). Decreased expression of inflammatory genes had functional consequences, as we detected significantly reduced levels of serum cytokines such as IFNγ (Extended Data Fig. 8b) and significantly reduced neutrophil infiltration at 48 h post co-infection (Extended Data Fig. 8c), as well as a slightly reduced pathology score (Extended Data Fig. 8d).

Results of our in vitro studies allowed us to mechanistically link the consequence of contact between *Salmonella* and *C. albicans* to this initially unexpected reduced inflammatory response. In vitro, *C. albicans* produced arginine in the presence of *Salmonella*. Although in vivo metabolomic analysis of caecal content did not indicate an overall increased availability of arginine, *Salmonella* gene expression indicated that there are probably locally increased levels of arginine (Fig. 2i). Arginine was previously shown to exert anti-inflammatory effects[40–42] and supplementation of arginine resulted in reduced inflammation after *Salmonella* infection in broiler chickens[43]. We therefore tested whether supplementation of L-arginine in the gastrointestinal environment (Fig. 5c) would phenocopy the presence of *C. albicans*. Indeed, addition of 2% L-arginine to the drinking water (adjusted to pH 7) resulted in a reduced inflammatory response (Fig. 5d, Extended Data Fig. 8e and Supplementary Tables 11 and 12), a trend towards higher *Salmonella* caecal colonization, and significantly increased *Salmonella*

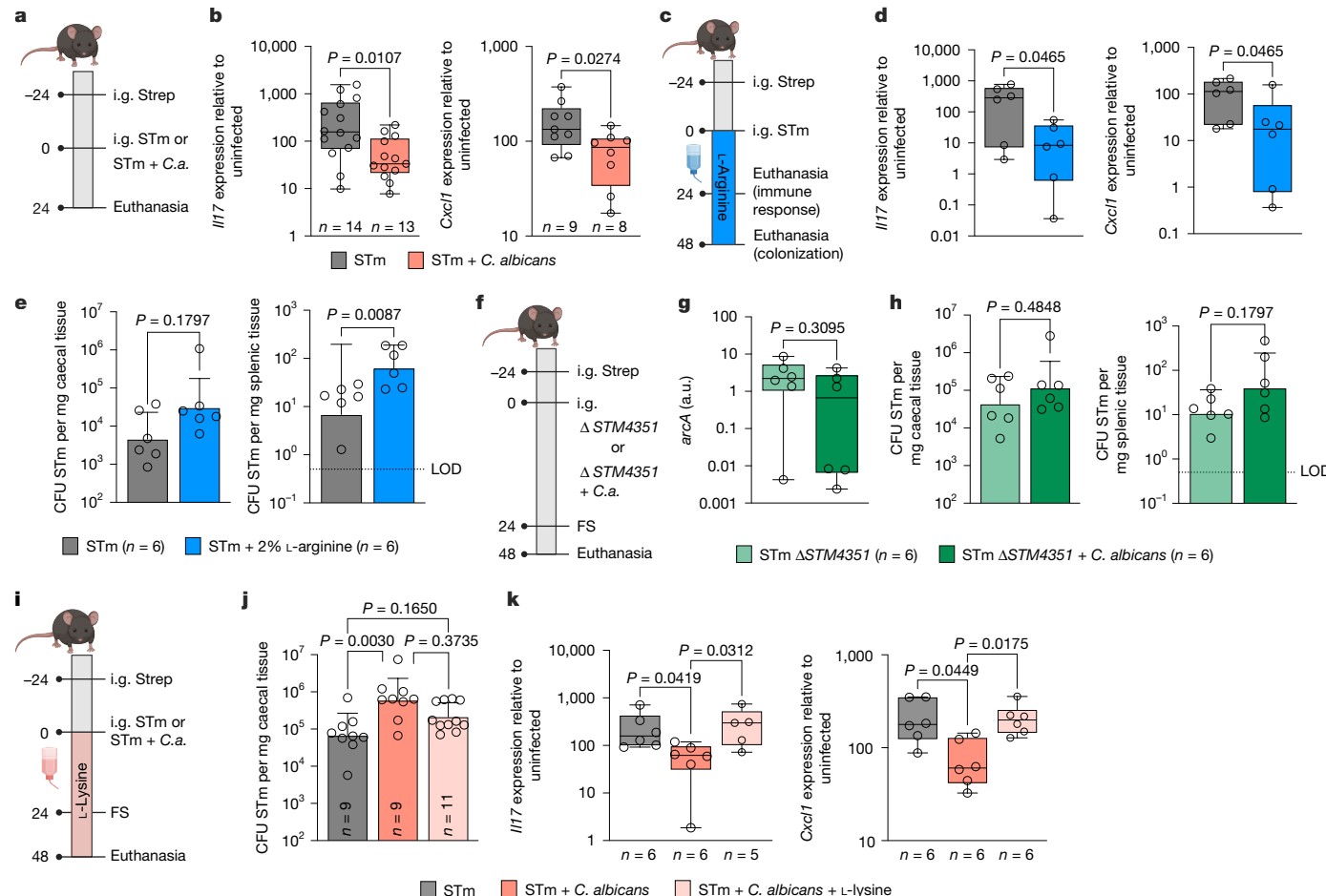

**Fig. 5 | Arginine uptake regulates host and *Salmonella* responses in vivo.**
**a**, Experimental schematic. **b**, rt-qPCR of host inflammatory genes from the caecal tissue of STm or STm + *C. albicans* ATCC infected mice 24 h post-infection; *n* represents animals from three independent experiments. Significance was determined using a Mann Whitney *U* test. **c**, Experimental schematic with 2% L-arginine in drinking water. **d**, rt-qPCR of host inflammatory genes from the caecal tissue of STm or STm with L-arginine-treated mice 24 h post-infection; *n* = 6 animals from two independent experiments. Significance was determined using a Mann Whitney *U* test. **e**, STm colonization in C57BL/6 mice infected with STm or STm treated with 2% L-arginine 48 h post-infection. Data are geometric mean ± s.d.; *n* = 6 animals from two independent experiments. Significance was determined using a Mann–Whitney *t*-test. **f**, Experimental schematic. **g**, rt-qPCR of STm arginine metabolism genes from the caecal content of STm or STm + *C. albicans* SC5314 infected mice 48 h post-infection; *n* = 6 animals from two independent experiments. Significance was determined using a two-tailed Mann–Whitney *t*-test. a.u., arbitrary units. **h**, STm colonization

in C57BL/6 mice infected with STm or STm + *C. albicans* SC5314 48 h post-infection. Data are geometric mean ± s.d.; *n* = 6 animals from two independent experiments. Significance was determined using a two-tailed Mann–Whitney *t*-test. **i**, Experimental schematic with 20 mM L-lysine in drinking water. The schematics in panels **a**,**c**,**f**,**i** were created in BioRender. Behnsen, J. (2025) https://biorender.com/xv05v3d. **j**, STm colonization in C57BL/6 mice infected with STm or STm + *C. albicans* SC5314 with or without 20 mM L-lysine 48 h post-infection. Data are geometric mean ± s.d.; *n* represents animals from three independent experiments. Significance was determined using a Kruskal–Wallis test. **k**, rt-qPCR of host inflammatory genes from the caecal tissue of mice infected with STm or STm + *C. albicans* SC5314 with or without L-lysine 48 h post-infection; *n* represents animals from two independent experiments. Significance was determined using a Kruskal–Wallis test. The box and whisker plots show the 25–75th percentile, median, minimum and maximum. All experiments used the streptomycin pre-treatment mouse model.

dissemination to the spleen and liver 48 h post-infection (Fig. 5e and Extended Data Fig. 8f). With L-arginine supplementation, induction of *Il17* and *Cxcl1* was significantly lower 24 h post-infection with *Salmonella* (Fig. 5d) and mice lost more weight (Extended Data Fig. 8g). L-Arginine in the drinking water did not change immune gene expression in uninfected mice (Supplementary Table 11) or arginine levels in the caecum (Extended Data Fig. 8h and Supplementary Table 13). Exogenous supply of arginine therefore reduced the inflammatory response to *Salmonella* infection and increased *Salmonella* systemic dissemination similar to co-infection with *C. albicans*. The reduced inflammatory response was dependent on the ability of *C. albicans* to produce arginine, as inflammatory genes such as *Ifng* and *Il17* were expressed significantly higher in mice infected with *Salmonella* and *arg4*Δ/Δ *C. albicans* than in mice infected with *Salmonella* and WT *C. albicans* (Fig. 3c and Extended Data Fig. 7b).

Our collective data thus far showed the twofold effects of *C. albicans*-produced arginine: (1) increased virulence gene expression in *Salmonella*, and (2) reduced inflammatory response to *Salmonella* infection. However, the relative importance of each response in vivo remained unknown. To separate the effects, we either diminished arginine uptake by *Salmonella* or arginine uptake by the host. *Salmonella* mutants deficient in well-characterized arginine uptake systems have defects in virulence[44,45]. We therefore investigated *STM4351*, which encodes a binding protein involved in the specific transport of arginine[31,46], whose expression was highly upregulated in the presence of *C. albicans* (Fig. 2e,i and Extended Data Fig. 3b). Deletion of *STM4351* in *Salmonella* did not result in growth defects in lysogeny broth (Extended Data Fig. 9a) or reduced binding to *C. albicans* (Extended Data Fig. 9b). In the presence of *C. albicans*, *Salmonella* Δ*STM4351* did not upregulate the expression of *arcA* in vitro (Extended Data Fig. 9c) and

in the caecum of mice (Fig. 5g), or *hilA* and *invA* in vitro (Extended Data Fig. 9c). Deletion of *STM4351* thus rendered *Salmonella* unresponsive to *C. albicans*-produced arginine. *Salmonella* Δ*STM4351* had no defects in caecal colonization and dissemination (Fig. 5f,h and Extended Data Fig. 9d) but induced a lower inflammatory response (Extended Data Fig. 9e). The presence of *C. albicans* only slightly and not significantly increased *Salmonella* Δ*STM4351* caecal colonization and dissemination to the spleen and liver (Fig. 5h and Extended Data Fig. 9d) and its presence did not change the inflammatory response (Extended Data Fig. 9e). L-Lysine is a competitive inhibitor of L-arginine uptake by the host gut epithelium[47,48] (Fig. 5j). L-Lysine administration in the drinking water did not alter *Salmonella* single infection parameters (Extended Data Fig. 9f,g). However, it partially reversed the increase of *Salmonella* caecal colonization (Fig. 5i,j) and dissemination to the spleen and liver (Extended Data Fig. 9h) triggered by the presence of *C. albicans*. L-Lysine administration also increased the inflammatory response to *Salmonella* during co-infection to levels similar to infection with *Salmonella* alone (Fig. 5k and Extended Data Fig. 9i). Both approaches, limiting *Salmonella* arginine uptake and limiting host arginine uptake, resulted in partial amelioration of the phenotype. We therefore conclude that *C. albicans*-produced arginine has significant effects on both *Salmonella* and the host.

Our collective data show that arginine has a crucial role to increase the virulence of *Salmonella* in the presence of *C. albicans*.

## Discussion

Our study delineates intricate molecular interactions that govern an interkingdom interaction in the gut between *Salmonella*, *C. albicans* and the host (model in Extended Data Fig. 10). Molecular details for fungal–bacterial interactions are only known for few interactions, such as *C. albicans*-induced *Staphylococcus aureus* toxicity in extraintestinal sites[49,50], or in the oral mucosa, where *Enterococcus faecalis* inhibits *C. albicans* morphogenesis by secreting the peptide EntV[51]. The amino acid arginine is emerging as a key metabolite in the gut, directly influencing microbial pathogenesis. *C. rodentium* upregulates virulence genes in the gut in the presence of arginine[32]. Conversely, lack of arginine due to *E. faecalis* catabolism serves as a metabolic cue for *Clostridioides difficile* to increase virulence gene expression[52]. Some pathogens deplete arginine to reduce its availability to iNOS[44,53,54]. However, during *Salmonella* infection, *C. albicans* arginine production or exogenously supplied L-arginine was anti-inflammatory and resulted in more severe disease.

Native translocation of SopB into *C. albicans* using the T3SS has been shown once before[55]. Considering the thickness of the *C. albicans* cell wall and the length of the T3SS needle, such activity should be physically impossible. However, other microorganisms are known to secrete fungal cell wall-degrading enzymes[56] and *Salmonella* chitinases[57] might result in local changes of the fungal cell wall that enable T3SS deployment. In *C. albicans*, SopB increased arginine biosynthesis, a phenotype absent in mammalian cells. Divergent evolution of a key protein, Arg82p, probably explains this differing phenotype[58]. SopB is an inositol phosphatase[37] and Arg82p is a kinase that catalyses the reverse reaction and additionally serves as a scaffolding protein of the arginine repressor complex[59]. SopB might affect arginine biosynthesis by feedback regulation or by making Arg82p physically unavailable to stabilize the arginine repressor complex.

Currently, the prevalence of *C. albicans* during *S.* Typhimurium infections of human patients is unknown. However, particularly in at-risk patients receiving antibiotic therapy that would allow fungal expansion, the presence of *C. albicans* in the gut could be an important but unappreciated risk factor. A study in Cameroon found that the recurrence of *Salmonella enterica* subsp. *enterica* serovar Typhi and Paratyphi infections increased fourfold when patients were colonized with *Candida* spp.[60]. In vulnerable populations, administration of an antimycotic might represent a novel approach to limiting sepsis

risk after *Salmonella* infection. In summary, our study has identified *C. albicans* as a susceptibility factor for *Salmonella* infection and arginine as a key metabolic cue in the interaction of *Salmonella*, *C. albicans* and the host.

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

## Methods

### Bacterial and fungal culture conditions

Fungal strains, bacterial strains and plasmids used in this study are listed in Supplementary Tables 14 and 15. *S.* Typhimurium strains were grown at 37 °C in lysogeny broth (LB) medium (per litre: 10 g tryptone, 5 g yeast extract and 10 g sodium chloride). *C. albicans* strains were grown at 37 °C in YPD media (per litre: 10 g yeast extract, 20 g peptone and 20 g dextrose). Antibiotics and antifungals were added at the following concentrations to LB agar or YPD agar, as needed: 100 mg l$^{-1}$ carbenicillin–ampicillin, 30 mg l$^{-1}$ chloramphenicol, 50 mg l$^{-1}$ kanamycin, 1 or 10 mg l$^{-1}$ doxycycline, 2.5 mg l$^{-1}$ amphotericin, 200 or 25 mg l$^{-1}$ nourseothricin (NAT) and 600 pr 75 mg l$^{-1}$ hygromycin B. All strains were cultured aerobically shaking at 200 rpm overnight except for in vitro experiments where *Salmonella* strains were grown overnight in liquid LB without shaking at 37 °C. Co-incubation of *S.* Typhimurium and *C. albicans* strains were done by quantifying the cell number by measuring the optical density at 600 nm (OD$_{600}$) of the cultures. Cells of *Salmonella* strains ($1 \times 10^9$) and cells of *C. albicans* strains ($1 \times 10^8$) were centrifuged and resuspended in 1 ml LB and co-incubated for 2 h in 37 °C without shaking, unless indicated otherwise. The cell-free supernatant from the 2 h co-incubations was filter sterilized and used to co-incubate with *Salmonella* to analyse the effect of secreted factors. When indicated, $1 \times 10^8$ cells of *C. albicans* were heat killed at 100 °C for 1 h, before co-incubation with *Salmonella*.

### Animal models

**Specific pathogen-free mice.** All animal experiments were reviewed and approved by the Institutional Animal Care and Use Committee at the University of Illinois Chicago in protocols 17-045, 20-016 and 22-192 and were in agreement with ethical regulations. Eight-to-nine-week-old C57BL/6J female or male WT mice (strain number 000664) and CBA/J female WT mice (strain number 000656) were obtained from Jackson Laboratories, maximum barrier, and were 9–10 weeks of age by the start of every experiment. Mice were housed in the Biologic Resources Laboratory facility at the University of Illinois Chicago in individual cages with filter tops (maximum occupancy of five mice) containing corn cob bedding and paper nesting material. The room the mice were housed in was kept under 14 h–10 h light–dark cycles, 70–76 °F and 30–70% humidity. Mice had access to food and autoclaved water ad libitum. Mice were fed chow LM485. Mice were randomized to experimental cages before the experiments. Experimenters were not blinded to experimental details in mouse experiments. No sample-size calculation was performed. Sample sizes were chosen based on past experience using the methods previously described[61–64]. Mouse experiments were performed with 2–6 mice per group and repeated for a minimal number of 5 mice per experimental condition. For infections with C57BL/6J mice, mice were pre-treated with streptomycin (0.1 ml of a 200 mg ml$^{-1}$ solution in sterile water) intragastrically 24 h before inoculation with *Salmonella* (WT) or *C. albicans* (WT) or *Salmonella* (Δ*sopB* or Δ*STM4351*) and *C. albicans* (WT, *arg4*Δ/Δ or *arg4*Δ/Δ + *ARG4*). For initial experiments, $1 \times 10^9$ colony-forming units (CFU) ml$^{-1}$ of *Salmonella* and $1 \times 10^8$ CFU ml$^{-1}$ of *C. albicans* were used (Figs. 1a–e and 5a,b and Extended Data Figs. 1c–e, 7a,c–e and 8a,c,d). The same dose, however, yielded significantly higher colonization and dissemination of *Salmonella* in single infected mice in subsequent experiments. We therefore maintained the same ratio of *Salmonella* to *C. albicans* but decreased the overall dose to $1 \times 10^7$ CFU ml$^{-1}$ of *Salmonella* and $1 \times 10^6$ CFU ml$^{-1}$ of *C. albicans*. This dose yielded *Salmonella* caecal colonization and dissemination levels similar to previous experiments (Figs. 2i, 3c,d, 4f and 5c–k and Extended Data Figs. 3h,i, 4f–i, 7b, 8b,e–h and 9d–i). For experiments with *C. albicans* arginine auxotroph (*arg4*Δ/Δ), mice were either gavaged again with *C. albicans* at 24 h post-infection or the day 0 dose of *C. albicans* was increased to $1 \times 10^7$ CFU ml$^{-1}$ (Figs. 3c,d and 4f and Extended Data Figs. 1f, 4f–h and 6f–i) and excluded the mice if

*C. albicans* did not colonize in the caecum. When required, mice were treated with 2% L-arginine (adjusted to approximately pH 7) or 20 mM L-lysine (adjusted to approximately pH 7) in their drinking water ad libitum. For infections with CBA/J mice, mice were colonized intragastrically with $1 \times 10^8$ CFU ml$^{-1}$ of *C. albicans* 529L or PBS control 3 and 1 days before and, to maintain equal colonization levels across mice, 4 days post intragastrically infection with $1 \times 10^9$ CFU ml$^{-1}$ of *Salmonella*. Inflammation develops slower in this model, with peak inflammation reached at 9–10 days post-infection[65]. Weights were monitored daily for both mouse models. Faecal samples were collected daily, and serial tenfold dilutions were plated for enumerating bacterial CFU on LB agar plates supplemented with either carbenicillin or kanamycin with amphotericin or YPD agar plates with chloramphenicol. Mice were euthanized at time points indicated. Caecal colonization and dissemination were assessed by homogenization and plating of the caecum, spleen, liver, Peyer's patches and mesenteric lymph nodes, respectively. Colon content was collected to enumerate *Salmonella* colonization and was flash frozen for microbiota analysis. Caecal luminal content was also collected, and flash frozen to either isolate RNA or perform metabolomic analysis. Caecal tissue was collected by flash freezing to analyse mouse inflammatory gene expression and by fixing it with formalin for histological examination. No statistical methods were used to pre-determine sample sizes, but our sample sizes are similar to those reported in previous publications[62–64].

**Gnotobiotic mice.** Swiss Webster (Tac:SW) germ-free WT mice were purchased from Taconic and C57BL/6 mice were purchased from Jax and bred at the BRL facility at the University of Illinois Chicago in a room with 14 h–10 h light–dark cycles and 70–76 °F and 30–70% humidity. C57BL/6 germ-free mice for some experiments were obtained from the University of Illinois Chicago Gnotobiotic core. Mice were kept in isolators purchased from Park Bioservices. Mice were fed autoclaved 5L79 chow and autoclaved super Q water ad libitum. Germ-free conditions were tested at least once a month with aerobic liquid cultures (brain heart infusion), solid cultures (blood agar plates; R01200, Thermo Sci Remel), fungal cultures (Sabouraud slants), anaerobic liquid (brain heart infusion) and solid cultures (Brucella agar; R01254, Thermo Sci Remel) from swabs of isolators, faecal samples and fungal traps placed inside the isolators. Faecal samples were also tested with Gram staining and qPCR to detect bacterial DNA. For ASF experiments, mice were stably colonized with ASF purchased from Taconic Biosciences. Stable colonization was assessed via species-specific PCR[66].

For immunofluorescence and *Candida* gene expression experiments, 10–15-week-old ASF-colonized Swiss Webster male and female mice were housed in a biosafety cabinet. Mice were then intragastrically infected with 0.1 ml of $1 \times 10^7$ CFU ml$^{-1}$ of *Salmonella* or $1 \times 10^7$ CFU ml$^{-1}$ of *Salmonella* and $1 \times 10^6$ CFU ml$^{-1}$ of *C. albicans*. Faecal samples were collected, and serial tenfold dilutions were plated for enumerating bacterial CFU on LB agar plates supplemented with carbenicillin and amphotericin or YPD agar plates with chloramphenicol. Mice were euthanized at 6 h and 24 h post-infection and caecal and colonic samples were collected by fixing it with formalin for immunofluorescence. For *Candida* gene expression experiments, mice were euthanized at 6 h and 12 h and luminal caecal content samples were collected and snap frozen.

For germ-free mouse experiments, 8–10-week-old C57BL/6 mice were housed in a biosafety cabinet. Mice were inoculated with $1 \times 10^7$ CFU ml$^{-1}$ of *C. albicans* intragastrically 24 h before infecting with $1 \times 10^4$ CFU ml$^{-1}$ of *Salmonella*. Mice were euthanized at 24 h post-*Salmonella* infection. For ASF mouse experiments, 8–10-week-old ASF-colonized C57BL/6 mice were housed in a biosafety cabinet. Mice were inoculated with $1 \times 10^7$ CFU ml$^{-1}$ of *C. albicans* intragastrically 24 h before infecting with $1 \times 10^7$ CFU ml$^{-1}$ of *Salmonella*. Mice were euthanized at 48 h post-*Salmonella* infection. Faecal samples were collected, and serial tenfold dilutions were plated for enumerating bacterial CFU on LB agar

plates supplemented with carbenicillin and amphotericin or YPD agar plates with chloramphenicol. Caecal colonization and dissemination were assessed by homogenization and plating of the caecum, spleen and liver, respectively. Colonic content was collected to enumerate *Salmonella* colonization. Caecal luminal content was also collected, and flash frozen to either isolate RNA or perform metabolomic analysis. Caecal tissue was collected by flash freezing to analyse mouse inflammatory gene expression and by fixing it with formalin for histological examination.

Both male and female mice were used in this study. Most experiments were conducted using female mice, with the repeat of key experiments in male mice to ensure reproducibility across sexes. Specifically, Fig. 1f includes five male and three female mice per group. Extended Data Fig. 5a–d includes five female and two male mice in the STm group and six female and two male mice in the STm + *C. albicans* group. Extended Data Fig. 5e–h includes three male and two female mice per group.

## Mycobiota and microbiota sequencing and analysis

ITS1 and 16S sequencing of mouse faecal samples were performed largely as previously described[12,61]. In brief, for DNA isolation, one to two mouse faecal pellets were resuspended in 0.5 ml lyticase buffer (50 mM Tris, 1 mM EDTA and 0.2% 2-mercaptoethanol), homogenized briefly and treated with 200 U lyticase (Sigma-Aldrich) for 30 min at 30 °C. Material was pelleted and resuspended in 0.4 ml stool DNA stabilizer (B Bridge International), mixed with 0.1 ml 0.1-mm silica beads (Biospec) and 0.3 ml 0.5-mm beads (Biospec), heated at 95 °C for 5 min and subjected to bead beating twice on high (VWR) for 1 min. DNA was then further purified using the QIAmp DNA mini kit (Qiagen) according to the manufacturer's instructions.

Primers used in this study are detailed in Supplementary Table 16. Fungal ITS1 amplicons were generated using primers with the following recombinant DNA targeting sequences: ITS1 forward (5′-CTTGGTCATT TAGAGGAAGTAA) and ITS2 reverse (5′-GCTGCGTTCTTCATCGATGC). ITS1 amplicons were generated with 35 cycles using Invitrogen AccuPrime PCR reagents at an annealing temperature of 48 °C. Bacterial 16S (V1–V3 region) amplicons were generated using primers with the following recombinant DNA-targeting sequences: 27 forward (5′-AGAGTTTGATCMTGGCTCAG) and 534 reverse (5′-ATTACCGCGGCT GCTGG). Amplicons were then used in the second PCR, using Illumina Nextera XT v2 barcoded primers to uniquely index each sample, and 2 × 300 paired-end sequencing was performed on the Illumina MiSeq.

ITS1 sequences were trimmed using ITSxpress (v1.7.4)[67] in QIIME 2 (v2019.7). Using the DADA2 package (v1.10.1) in R (v3.5.2), reads underwent further quality filtering as error rates were calculated and removed from the dereplicated reads. Where forward and reverse reads could be merged, they were, and where they could not, largely owing to ITS1 sequences sometimes being longer than sequence coverage, they were concatenated. An initial sequence table was constructed before chimeras were identified using the removeBimeraDenovo function. Finally, taxonomy was assigned using DADA2's native naive Bayesian classifier against the UNITE (v.8.2) database[68]. 16S sequences were trimmed using Cutadapt (v3.7) and taxonomy was assigned with the Green-Genes reference database (release of May 2013).

Statistical analyses were performed with R (v4.0.2)[69]. Figures were produced using the packages ggplot2 (ref. 70), dplyr[71], ape[72] and RColorBrewer[73]. Microbial communities were further analysed using the microbiome[74] and phyloseq[75] packages.

## Fluorescence microscopy

For in vivo experiments, tissue samples were fixed in 10% formalin for 24 h, stored in 70% ethanol and finally embedded in paraffin and sectioned at 7 μm with a microtome. Deparaffinization was performed by immersing the sections in xylene followed by decreasing concentrations of ethanol (100%, 90% and 70%). For antigen retrieval, the deparaffinized slides were immersed in 10 mM Na-citrate buffer and heated in a microwave for 20 min. Blocking was performed by flooding the slides with PBS–10% FBS for 30 min at 4 °C. *Salmonella* was stained with *Salmonella* antisera O-4 (1:100 in PBS; 294401, Hardy Diagnostics) for 1 h at room temperature. Following washing with PBS–0.3% Tween20, slides were incubated for 30 min at room temperature with Alexa Fluor 594 goat anti-rabbit antibody (1:1,000 in PBS; R37117, Invitrogen). Slides were washed with PBS–0.3% Tween20 and *C. albicans* was stained using a FITC-labelled rabbit anti-fungal antibody (1:250 in PBS–10% FBS; B65411F, Meridian Life Science) for 1 h at 37 °C. Slides were washed with PBS–0.3% Tween20, mounted using ProLong Diamond Antifade Mounting with DAPI (Invitrogen) and images were taken using the BZ-X710 All-in-One Fluorescence Microscope at ×60.

For in vitro experiments, to visualize agglutination, $OD_{600}$ of overnight *Salmonella*–mCherry and *C. albicans*–GFP was measured. STm ($4 × 10^9$) and *C. albicans* ($7.6 × 10^7$) were centrifuged and resuspended in 1 ml and 2 ml of PBS, respectively. On a microscope slide, 20 μl of *Salmonella* and 20 μl of *C. albicans* were added to the slide and mixed by swirling the slide for 1 min. Of the mixture, 5 μl was added to an agarose pad (0.75 g of agarose in 50 ml of $H_2O$). Images were taken using the BZ-X710 All-in-One Fluorescence Microscope at ×4–40. For quantification of *prgH-gfp*[+] cells, 5 μl of $1 × 10^8$ STm SB300 *prgh-gfp* cells (expressing mCherry) alone or with $1 × 10^8$ *C. albicans* were spotted onto a 1 × 1-cm-wide, 3-mm-thick agarose pad (1.5% agarose in M9 minimal medium) positioned on a microscope slide and allowed to dry for 10 min. Once cells were absorbed onto agarose pads, coverslips were added, sealed with nail polish and then slides were incubated for 2 h at 37 °C. Cells were visualized at ×60 on a BZ-X710 All-in-One Fluorescence Microscope. The acquired images were not always completely in focus. Therefore, a section of each image in focus was selected for counting. Each selected section contained at least ten *C. albicans* cells. In each section, the number of mCherry-positive (total) and GFP-positive (Prgh-expressing) cells were counted and presented as the percentage of GFP[+]:mCherry[+].

## Invasion (gentamicin protection) assay

The T84 colonic epithelial cell line (CCL-248, ATCC; RRID:CVCL_0555) or C2BBe1 colonic epithelial cell line (Caco2; CRL-2102, ATCC; RRID:CVCL_1096) were obtained directly from the manufacturer, confirmed visually and tested for mycoplasma contamination. Cells were seeded onto a 24-well tissue-culture-treated plate at a density of $5 × 10^5$ cells per well with media lacking antibiotics and/or antimycotics and incubated overnight (for T84) and for 5 days (for Caco2) at 37 °C. *Salmonella* strains alone or co-incubated with either *C. albicans* strains or the cell-free supernatant were centrifuged, resuspended in DMEM/F12 and serially diluted. Epithelial cells were infected with $5 × 10^5$ cells of *Salmonella* (MOI = 1) and $5 × 10^4$ cells of *C. albicans* (MOI = 0.1). Infected cells were incubated at 37 °C for 1 h. The inoculum was serially diluted and plated on LB agar with amphotericin to confirm bacterial numbers. After infection, the medium was removed via vacuum, and wells were washed three times with 500 μl PBS. Of DMEM/F12 + 10% FBS + 0.1 mg ml$^{-1}$ gentamicin, 500 μl was added to the wells and incubated at 37 °C for 1 h to kill extracellular bacteria. After incubation, the wells were washed with PBS and lysed by incubation with 1% Triton X-100 for 5–10 min. Cells were disrupted and harvested by scraping wells and pipetting, were serially diluted and plated on LB agar with amphotericin to quantify bacterial cells that invaded. The percentage of cells recovered relative to the inoculum was calculated.

## Sedimentation assay

$OD_{600}$ of overnight *Salmonella* and *C. albicans* cultures was measured. *Salmonella* ($4 × 10^9$) and *C. albicans* ($7.6 × 10^7$) were centrifuged and resuspended in 1 ml and 2 ml of PBS, respectively. In a 15-ml conical tube, 1 ml of *Salmonella* and 2 ml of *C. albicans* were added and vortexed for 15 s before taking 200 μl from the top of the mixture, which was

used to measure $OD_{600}$ (1:5 dilution) for the initial starting value. For a baseline control, 2 ml of *C. albicans* was mixed with 1 ml of PBS and the $OD_{600}$ was measured. Conical tubes (15 ml) were then incubated at 37 °C for 20 min without shaking; after that time, $OD_{600}$ was measured the same way as before. To calculate the percentage of sedimentation, the equation $((OD_{0\,min} - OD_{20\,min})/OD_{0\,min}) \times 100$ was used. The value of *Salmonella* and *C. albicans* was divided over the *C. albicans* alone to measure sedimentation.

### RNA extraction

For the in vivo experiment, caecal tissue collected from mice 24 h post-infection and 48 h post-infection were homogenized by mortar and pestle using liquid nitrogen. Because mice were treated with streptomycin 24 h before infection, we collected caecal tissue from uninfected mice after streptomycin treatment as a control. The homogenate was transferred to 1 ml of Tri-Reagent (Molecular Research Center) for RNA extraction. RNA was extracted with 0.1 ml of bromo-3-chloropro-pane, centrifuged and the upper phase was precipitated with 0.5 ml isopropanol. After centrifugation, pellets were washed twice with 1 ml of 75% ethanol in RNase-free water. The RNA pellet was then resuspended in RNase-free water. For caecal content, RNA was extracted from snap-frozen luminal caecal content samples collected during mouse experiments. RNA extraction was performed using the Qiagen RNeasy Power Microbiome kit. RNA was treated with DNase using the Turbo DNA-free kit (Invitrogen). For the in vitro experiments, RNA extraction was performed on the cell pellet from monocultures and co-cultures of *Salmonella* and *C. albicans* after 2 h of co-incubation, except to assess the expression of *sopB* in *C. albicans* in the *C. albicans tetO-sopB* strain. Here transformants containing CIptet-SopB or CIpSATtetTranstADH1 (empty vector) were grown overnight in liquid YPD media containing 10 µg ml⁻¹ doxycycline. Cells ($1 \times 10^5$ ml⁻¹) were transferred into 5 ml YPD with or without 10 µg ml⁻¹ doxycycline and incubated for 8 h at 30 °C with continuous shaking. In all cases, RNA extraction was performed using the hot-phenol method as previously described[76,77] followed by DNase treatment using the Turbo DNA-free kit (Thermo Fisher).

### RNA sequencing and analysis

For basic processing of RNA sequencing data, raw reads were aligned to the reference assembly (GCF_000022165.1) using BWA-MEM (v0.7.17)[78]. Expression levels of gene features, that is, coding DNA sequences regions from the reference assembly, were quantitated using FeatureCounts (v2.0.3) as raw read counts of the stranded libraries[79]. Differential analysis of quantitated gene features compared with treatment was performed using the software package edgeR on raw sequence counts[80]. Before analysis, the data were subsampled to a maximum depth of 1,750,000 counts per sample and filtered to remove any features that had less than 100 total counts summed across all samples. Data were normalized as counts per million and an additional normalization factor was computed using the trimmed mean of M values algorithm. Statistical tests were performed using the 'exactTest' function in edgeR. Adjusted *P* values (*q* values) were calculated using the Benjamini–Hochberg false discovery rate (FDR) correction[81]. Significant gene features were determined based on an FDR threshold of 5% (0.05). For enrichment analysis, the enrichment or over-representation of differentially expressed gene features in the various gene groups, that is, pathways, modules and BRITE categories, listed for the KEGG organism 'seo' (KEGG genome T01714) was determined using Fisher's exact test in R. In brief, a list of differentially expressed gene features was obtained from the results of the differential analysis based on a *q* value, that is, FDR-corrected *P* < 0.05. The enrichment of significantly different gene features as compared with all genes listed in KEGG for the organism 'seo' were then tested for the KEGG organism pathways, modules and BRITE level 1, 2 and 3 categories. Adjusted *P* values (*q* values) were calculated using the Benjamini–Hochberg FDR correction[81].

Significant enrichment of gene groups was determined based on an FDR threshold of 5% (0.05).

### rt-qPCR

Reverse transcription was performed with the High-Capacity cDNA Reverse Transcription Kit (Applied Biosystems). For *Salmonella* RNA, reactions were also performed without the addition of reverse transcriptase to confirm that there was no amplification of DNA in rt-qPCRs. Of RNA, 500 ng was used for the reverse transcription reaction. The reverse transcription cycle consisted of 10 min at 25 °C followed by 120 min at 37 °C and 5 min at 85 °C. rt-qPCR was performed using the Fast SYBR Green Master Mix (Applied Biosystems) on the Viia7 Real-time PCR system at the Genome Research core at the University of Illinois at Chicago. The rt-qPCR cycle consisted of 20 s at 95 °C followed by 40 cycles of 3 s at 95 °C and 30 s at 60 °C. Reactions were performed in duplicate.

For transcript levels of *sopB* in doxycycline-repressible strains, samples were normalized to 1 µg RNA and treated with Turbo DNase (Thermo Fisher). First-strand cDNA synthesis was performed using the RevertAid RT kit following the manufacturer's protocol. Amplification of approximately 20 ng cDNA was performed using 2X Maxima SYBR Green/ROX qPCR Master Mix (Thermo Fisher) and gene-specific primers for *sopB* (CaSopBDETF + CaSopBDETR). rt-qPCRs were monitored and analysed with a Bio-Rad CFX96 Real-Time System and software.

Relative expression was calculated based on the ΔCT values obtained by subtracting the CT value of the house-keeping gene with the gene of interest. *gapA*, *ActB* and *ACT1* were used as house-keeping genes to normalize *Salmonella*, mouse immune and *C. albicans* gene expression, respectively.

### Metabolomics analyses

Amino acids were quantified using a Waters Acquity uPLC System with an AccQ-Tag Ultra C18 1.7-µm 2.1 × 100-mm column and a Photodiode Detector Array. Faecal samples were homogenized in methanol (5 µl mg⁻¹ stool) and centrifuged twice at 13,000*g* for 5 min. Intestinal flushes were vortexed for 1 min and centrifuged twice at 13,000*g* for 5 min. Amino acids in the supernatant were derivatized using the Waters AccQ-Tag Ultra Amino Acid Derivatization Kit (Waters Corporation) and analysed using the UPLC AAA H-Class Application Kit (Waters Corporation) according to the manufacturer's instructions. Blanks and standards were run every eight samples. All chemicals and reagents used were mass spectrometry grade.

### β-Galactosidase assay

β-Galactosidase assays were performed as previously detailed[82]. Promoter activity was measured by monitoring β-galactosidase expression from chromosomal transcriptional reporter fusions, as previously described[83].

### Statistical analyses

Statistical analyses were performed using GraphPad (v10.2.2). All in vitro experiments were conducted at least in triplicate. In all graphs, each symbol represents an independent sample (*n*), and the individual sample size per experiment has also been indicated in the figure legends. Statistical test used in each experiment has been described in the figure legends.

### Reporting summary

Further information on research design is available in the Nature Portfolio Reporting Summary linked to this article.

## Data availability

Raw sequence reads have been deposited at the NCBI Sequence Read Archive under project PRJNA1143068, PRJNA1255633 and

PRJNA1285498, and the UNITE (v.8.2) database[68] was used for all microbiome analysis. Source data used for figure generation are provided in source data files for Figs. 1–5 and Extended Data Figs. 1–9. Uncropped images for Fig. 2c,f and Extended Data Fig. 2f,g, and images used for enumeration in Extended Data Fig. 3d are provided as Supplementary Information. Additional source data of main and extended figures (microbiome analysis, amino acid metabolomics, RNA sequencing analysis, KEGG analysis and rt-qPCR data) are available in Supplementary Tables 1–13. Source data are provided with this paper.

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

**Acknowledgements** This work was supported by grants from the US National Institute of Health to J.B. (R01AI143641 and R01AI175223) and to B.M.P. (R01AI177615, R01DK113788 and R01AI134796), and by startup funds from the Department of Microbiology and Immunology at the University of Illinois Chicago. O.A.T. was supported by the Training Program in Lung Biology and Pathobiology (T32HL007829). The authors thank A. Baumler for providing the *Salmonella* Δ*fim*, other fimbriae mutants and T3SS-1 effector deletions; L. Knodler for providing the plasmid pFPV-mCherry, *Salmonella* mutants and complemented strains in the background strain SL1344; J. Slaugh for providing the *hilA–lacZ* fusion *Salmonella* strain; A. Hockenberry for providing the *Salmonella* strain *prgh-gfp*; S. Saville for the vector CIpSATtetTrans; K. Perfecto for help maintaining the germ-free mice; and all laboratory members and L. Palmer for critical reading of the manuscript. RNA sequencing was performed by the University of Illinois Chicago Genomic Research Core. Bioinformatic analysis was performed by the University of Illinois Chicago Research Informatics Core, supported in part by NCATS through grant UL1TR002003. The University of Illinois Chicago Gnotobiotic Core provided some animals for experiments. Amino acid analysis was performed by the Microbial Culture and Metabolomics Core of the PennCHOP Microbiome Program and the Center for Molecular Studies in Digestive and Liver Diseases (National Institutes of Health P30DK050306). Histopathology was performed by the University of Illinois Chicago Research Histology Core. Figures were created in BioRender (https://biorender.com).

**Author contributions** J.B. conceived and administered the study and was the primary supervisor. K.J. additionally conceived and supervised parts of the study and was responsible for execution and analysis of the majority of the experimental work. O.A.T. performed additional in vitro experiments, aided in mouse experiments and performed extended colonization mouse experiments. R.C.F.A. established binding assays and performed the in vitro microscopy. W.S. performed the initial mouse experiments and in vivo imaging with gnotobiotic mice. C.M.D. performed the cytokine ELISAs and growth curves. E.T.E. performed the single-cell microscopy analysis. K.J., O.A.T., W.S., R.C.F.A., E.T.E., C.M.D. and J.B. were involved in the husbandry of germ-free and gnotobiotic mice. M.S. analysed the histopathology of tissue sections. D.M.U. was responsible for microbiome sequencing and analysis. S.P. generated all *C. albicans*-mutant strains for this study, except for SC5314 GFPy, which was generated by J.M. B.M.P. provided conceptual insight and supervised *C. albicans*-mutant generation. J.B. and K.J. wrote the original draft of the manuscript. J.B. and K.J. revised the draft with input from all authors.

**Competing interests** The authors declare no competing interests.

**Additional information**
**Correspondence and requests for materials** should be addressed to Judith Behnsen.

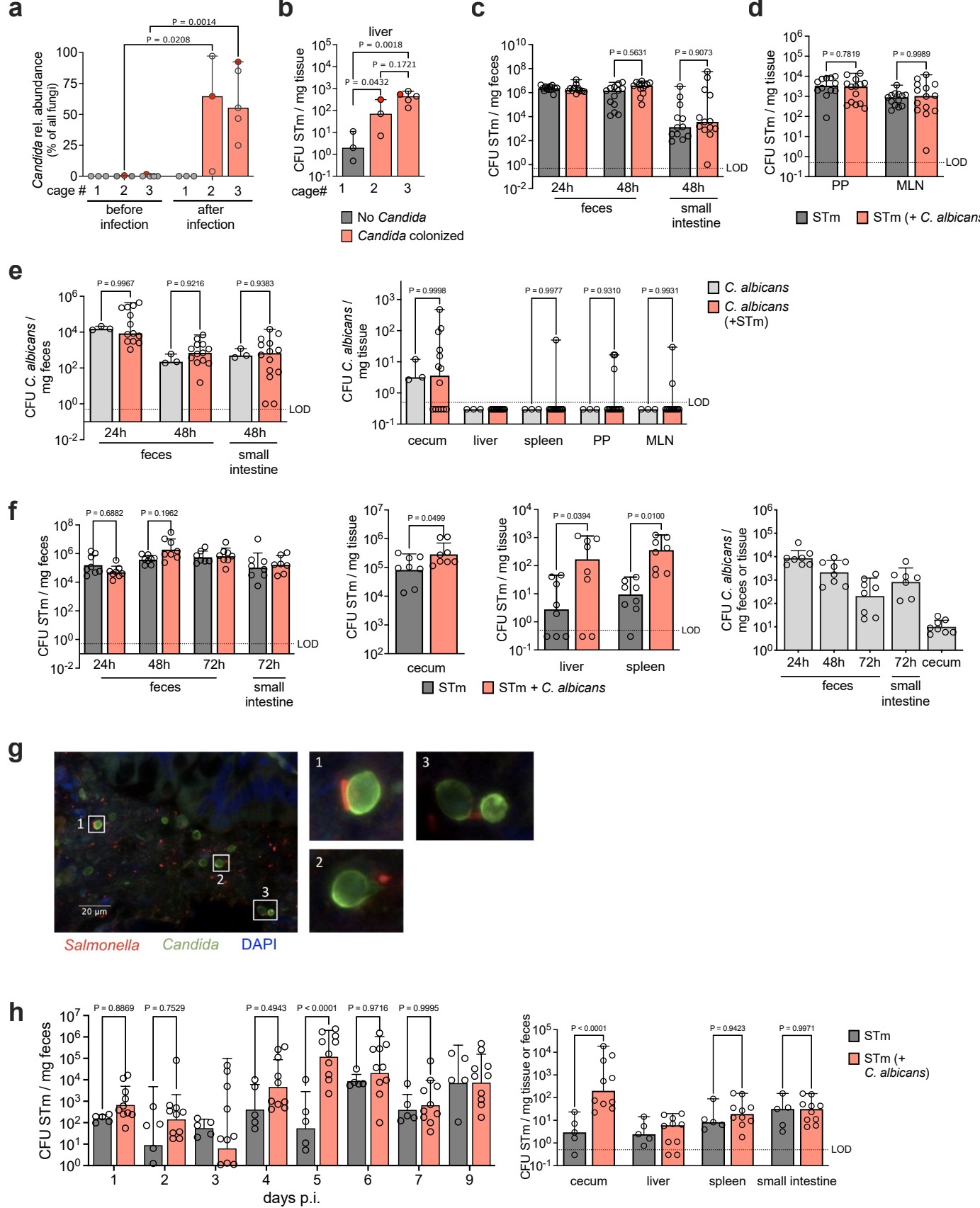

**Extended Data Fig. 1** | See next page for caption.

**Extended Data Fig. 1 | *C. albicans* increases *Salmonella* colonization and dissemination. a**, Relative abundance of *C. albicans* identified with ITS sequencing in fecal samples of mice before and after STm infection. Mice with reads for *C. albicans* before infection are represented as red circles. Data are median with range, n = 3 (Cage 1 and 2) and n = 5 (Cage 3) animals. Two-way ANOVA for comparison. **b**, STm colonization in liver 72 h p.i. Mice with reads for *C. albicans* before infection are represented as red circles. Data are median with range, n = 3 (Cage 1 and 2) and n = 5 (Cage 3) animals. Ordinary one-way ANOVA for comparison. **c,d**, STm colonization in C57BL/6 mice infected with STm or STm and *C. albicans* ATCC in the streptomycin pre-treatment model for 48 h p.i. Data are median with range. n = 14 (for STm + *C. albicans*: PP, MLN), n = 13 (for STm: feces, MLN and STm + *C. albicans*: feces, small intestine) and n = 12 (for STm: PP, small intestine) animals from 3 independent experiments. Two-way ANOVA for comparison. **e**, *C. albicans* colonization in C57BL/6 mice infected with *C. albicans* ATCC or *C. albicans* ATCC and STm in the streptomycin pre-treatment model for 48 h p.i. Data are median with range. n = 14 (for *C. albicans* + STm: small intestine, cecum, liver, spleen, PP, MLN), n = 13 (for *C. albicans* + STm: feces) and n = 3 (for *C. albicans* only samples) animals from 3 independent experiments. Two-way ANOVA for comparison. **f**, STm and *C. albicans* colonization in C57BL/6 mice (male and female) infected with a lower dose ($1 \times 10^{7}$ CFU) STm or ($1 \times 10^{7}$ CFU) STm and ($1 \times 10^{7}$ CFU) *C. albicans* ATCC in the streptomycin pre-treatment model for 72 h p.i. Data are geometric mean ± s.d. (for feces, small intestine and cecum) and median with range (for liver and spleen). n = 7 (for 72 h STm feces, STm + *C. albicans* small intestine and *C. albicans* small intestine) and n = 8 (for all other groups) animals from 2 independent experiments. Two-way ANOVA (feces and small intestine, liver and spleen) and two-tailed Mann–Whitney test (cecum) for comparison. **g**, Fluorescence image of lumen of colon during mouse infection with STm and *C. albicans* ATCC. Experiment was repeated independently with 4 mice. **h**, STm colonization in CBA/J mice infected with STm or STm and *C. albicans* 529 L in the absence of antibiotics for 9 d p.i. Data are geometric mean ± s.d (for feces) and median with range (for organs and small intestine). Two-way ANOVA for comparison. n = 5 (for STm) and n = 10 (for STm + *C. albicans*) animals. Bars with no statistics have p > 0.9999. CFU, colony-forming units; STm, *Salmonella*; PP, Peyer's patches; MLN, Mesenteric lymph nodes.

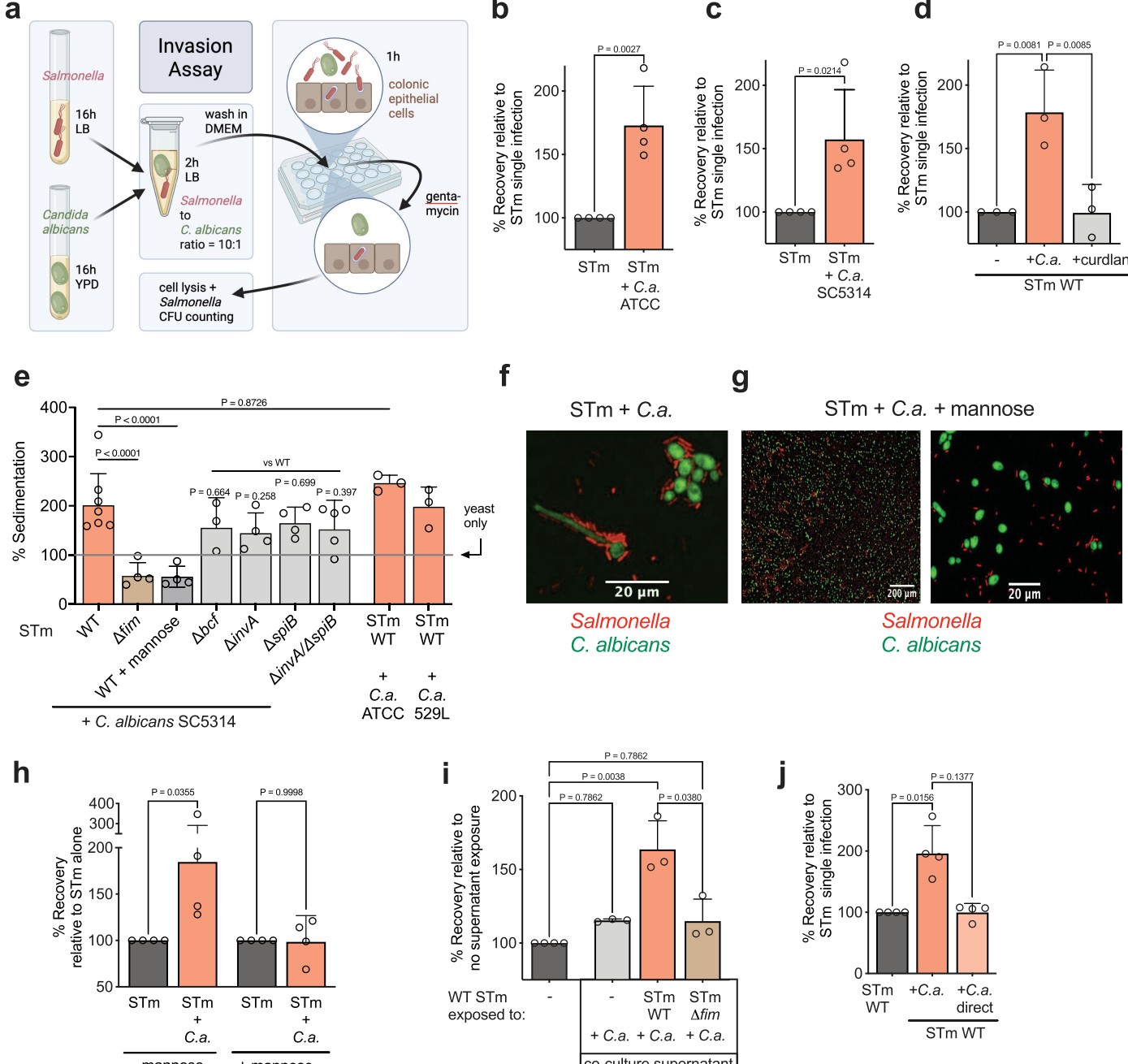

**Extended Data Fig. 2 | *Salmonella* invasion in the presence of *C. albicans*.**
**a**, Schematic representation of experimental setup for Invasion assay.
**b** & **c**, Invasion assay of STm infected (MOI = 1) colonic epithelial cells (Caco2).
STm either alone or with *C. albicans* ATCC (in **b**) *C. albicans* SC5314 (in **c**) in a 10:1
(*Salmonella* to *C. albicans*) ratio were incubated for 2 h prior to the assay. Data
are geometric mean ± s.d. n = 4 independent experiments. Two-tailed unpaired
t-test for comparison. **d**, Invasion assay of STm infected (MOI = 1) colonic
epithelial cells (T84). STm either alone or with *C. albicans* ATCC in a 10:1
(*Salmonella* to *C. albicans*) ratio or curdlan (ß-1,3-glucan) 100 μg/ml were
incubated for 2 h prior to the assay. Data are geometric mean ± s.d. n = 3
independent experiments. Ordinary one-way ANOVA for comparison.
**e**, Sedimentation assay of STm strains and *C. albicans*. Data are geometric
mean ± s.d. n = 6 (for WT), n = 4 (for Δ*fim*, WT+mannose, Δ*invA*, Δ*spiB*), n = 5
(for Δ*invA*Δ*spiB*) and n = 3 (for Δ*bcf*, +*C.a.*ATCC, and +*C.a.*529 L) independent
experiments. Ordinary one-way ANOVA for comparison. Assays were performed
with multiple mutants depicted in Fig. 2b and Extended Data Fig. 2e and one
STm WT control in parallel. For easier comparison, STm WT and Δ*fim* values

from Fig. 2b were also plotted in this graph. **f,g**, Fluorescence image of STm
(red) and *C. albicans* SC5314 (green) in vitro present as both yeast and hyphae
(in **f**) and in presence of 5% mannose (in **g**). **h**, Invasion assay of STm infected
(MOI = 1) colonic epithelial cells (Caco2). STm either alone or with *C. albicans*
ATCC in a 10:1 (*Salmonella* to *C. albicans*) ratio with or without mannose were
incubated for 2 h prior to the assay. Data are geometric mean ± s.d., n = 4
independent experiments. Two-way ANOVA for comparison. **i**, Invasion assay
of STm infected (MOI = 1) colonic epithelial cells (Caco2). STm either alone
or with the supernatant from *C. albicans* ATCC containing cultures were
incubated for 2 h prior to the assay. Data are geometric mean ± s.d. n = 4 (for
STm alone) and n = 3 (for other groups) independent experiments. Ordinary
one-way ANOVA for comparison. **j**, Invasion assay of STm infected (MOI = 1)
colonic epithelial cells (Caco2). STm either alone or with *C. albicans* SC5314 in a
10:1 (*Salmonella* to *C. albicans*) ratio incubated for 2 h prior to the assay. For
direct, both microbes were added to host cells without pre-incubation. Data
are geometric mean ± s.d. n = 4 independent experiments. Kruskal-Wallis test
for comparison. Bars with no statistics have p > 0.9999. STm, *Salmonella*.

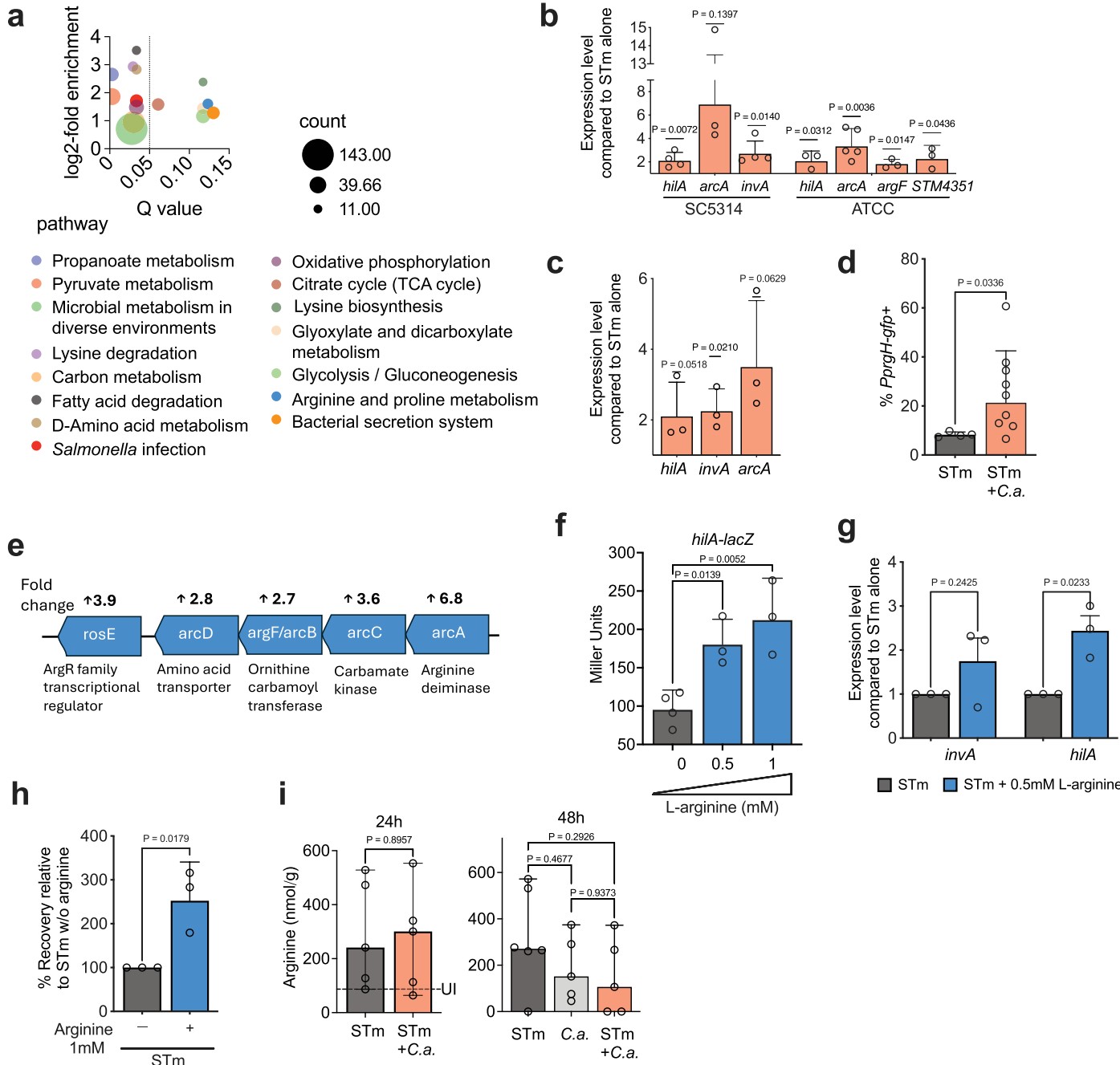

**Extended Data Fig. 3 | *Salmonella* transcriptomic changes in the presence of *C. albicans*. a**, Enrichment analysis showing upregulated KEGG pathways in STm in the presence of *C. albicans* ATCC. **b**, qRT-PCR analysis of STm genes from the STm or STm with either *C. albicans* ATCC or *C. albicans* SC5314 cultures incubated for 2 h. Data are geometric mean ± s.d. n = 4 (for SC5314: *hilA*, invA), n = 3 (for SC5314: *arcA* and ATCC: *hilA, argF, STM4351*) and n = 5 (for ATCC: *arcA*) independent experiments. One sample t-test for comparison. **c**, qRT-PCR analysis of STm genes from the STm SL1344 or STm SL1344 with *C. albicans* SC5314 cultures incubated for 2 h. Data are geometric mean ± s.d. n = 3 independent experiments. One sample t-test for comparison. **d**, Percentage of *Pprgh-gfp*+ cells in the presence of *C. albicans* SC5314 after incubation for 2 h. Data are geometric mean ± s.d. of 4 field of view (for STm) and 9 field of view (for STm + *C.a.*) from 3 independent experiments. Two-tailed Mann-whitney t-test for comparison. **e**, Genome organization of Arginine deiminase pathway and the corresponding fold increase in RNAseq analysis. **f**, ß-galactosidase assay

performed on STm carrying chromosomal fusion of *lacZ* with the promoter of *hilA* after incubation with L-arginine for 2 h. Data are geometric mean ± s.d. n = 4 (without arginine) and n = 3 (with arginine) independent experiments. Ordinary one-way ANOVA for comparison. **g**, qRT-PCR analysis of STm genes from STm alone or in the presence of L-arginine cultures incubated for 2 h. Data are mean ± SEM. n = 3 independent experiments. Two-way ANOVA for comparison. **h**, Invasion assay of STm infected (MOI = 1) colonic epithelial cells (T84). STm either alone or with L-arginine were incubated for 2 h prior to the assay. Data are geometric mean ± s.d. n = 3 independent experiments. Two-tailed unpaired t-test for comparison. **i**, Arginine levels in the cecum content of STm, *C. albicans* SC5314, or STm and *C. albicans* SC5314 infected mice 24 h and 48 h p.i. Data are represented as median with range. n = 6 (48 h: STm) and n = 5 (for all other groups) animals. Two-tailed unpaired t-test (24 h) and Ordinary one-way ANOVA (48 h) for comparison. Dashed line indicates levels in uninfected mice. STm, *Salmonella*.

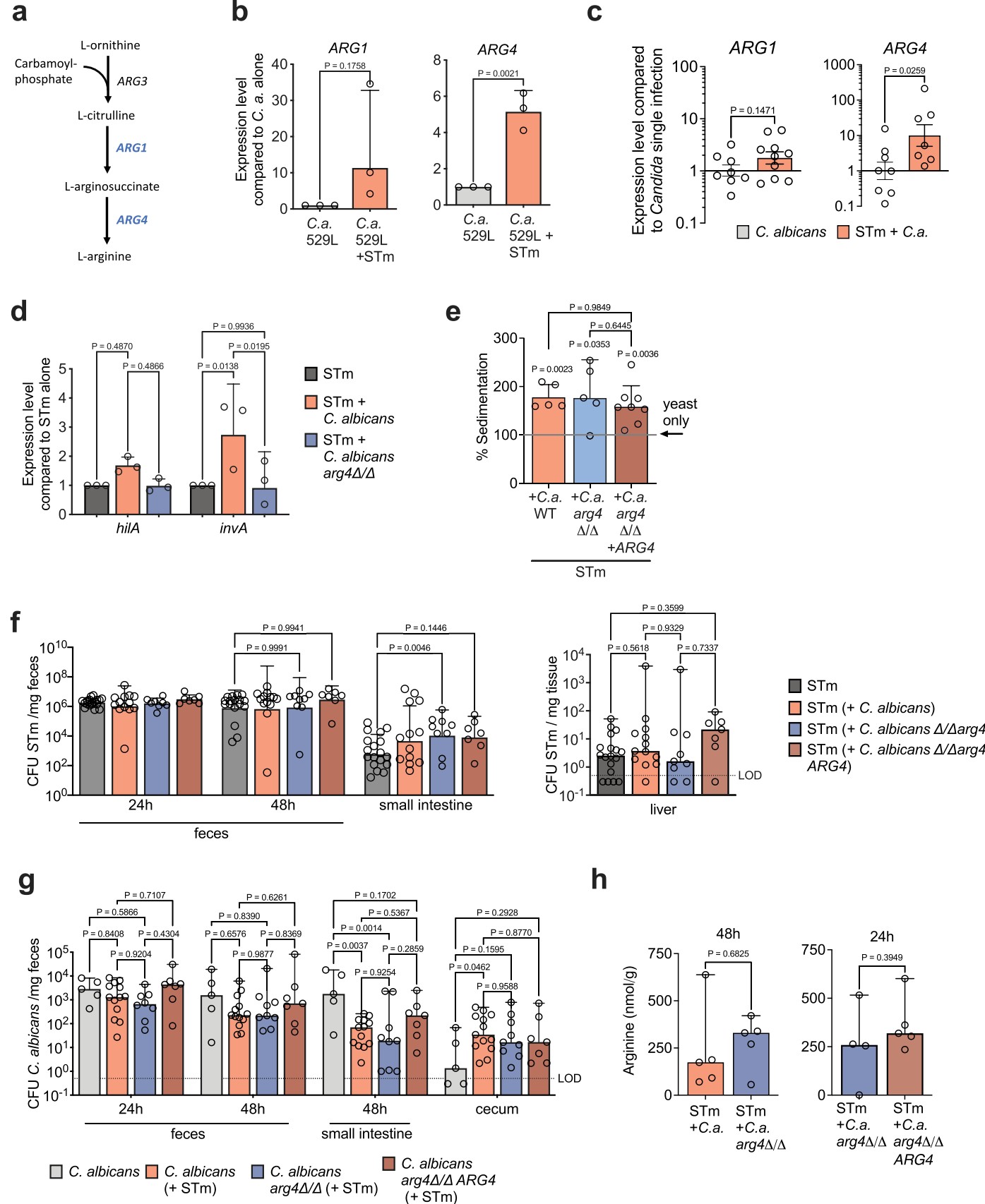

**Extended Data Fig. 4** | See next page for caption.

**Extended Data Fig. 4 | Role of *C. albicans* arginine biosynthesis for *Salmonella* virulence. a**, Schematic of arginine biosynthesis pathway in *C. albicans*. **b**, qRT-PCR analysis of genes encoding for *C. albicans* arginine biosynthesis from *C. albicans* 529L or STm and *C. albicans* 529L cultures incubated for 2 h. Data are geometric mean ± s.d. n = 3 independent experiments. Two-tailed unpaired t-test for comparison. **c**, qRT-PCR analysis of genes encoding for *C. albicans* arginine biosynthesis from the cecal content of *C. albicans* SC5314 or STm and *C. albicans* SC5314 infected mice at 6 and 12 h p.i. Data are mean ± SEM. n = 8 (for *C. albicans*: *ARG1* and *ARG4*) and n = 10 (for STM + *C.albicans*: *ARG1*) and n = 7 (for STM + *C.albicans*: *ARG4*) animals from 2 independent exp. Two-tailed Welch's t-test for comparison. **d**, qRT-PCR analysis of STm genes from the STm or STm with *C. albicans* SC5314 cultures incubated for 2h. Data are geometric mean ± s.d. n = 3 independent experiments. Two-way ANOVA for comparison. **e**, Sedimentation assay of STm and *C. albicans* SC5314. Data are geometric mean ± s.d. n = 5 (for *C.a.* WT and *C.a arg4Δ/Δ*) and n = 8 (for *C.a. arg4Δ/Δ + ARG4*) independent experiments. Ordinary one-way ANOVA for comparison among strains and one sample t-test for comparison with yeast only. **f**, STm colonization in C57BL/6 mice infected with STm or STm and *C. albicans* SC5314 in the streptomycin pre-treatment model for 48 h p.i. Data are geometric mean ± s.d. (for feces and small intestine) and median with range (for liver). n = 18 (for STm: 24 h feces, small intestine and liver), n = 13 (for STm+*C.a.*: 24 h and 48 h feces, small intestine and liver), n = 8 (for STm+*C.a. arg4Δ/Δ*: 24 h feces), n = 7 (for STm+*C.a. arg4Δ/Δ + ARG4*: 24 h and 48 h feces, small intestine and liver), n = 17 (for STm: 48 h feces), n = 9 (for STm+*C.a. arg4Δ/Δ*: 48 h feces, small intestine and liver) from 4 independent experiments. Ordinary one-way ANOVA (for liver) and Mixed-effects analysis (for fecal sample) for comparison. **g**, *C. albicans* colonization in C57BL/6 mice infected with *C. albicans* SC5314 alone or *C. albicans* SC5314 and STm in the streptomycin pre-treatment model for 48 h p.i. Data are median with range. n = 5 (for *C. albicans*: 24 h and 48 h feces, small intestine and cecum), n = 13 (for *C.a.*+STm: 24 h and 48 h feces, small intestine and cecum), n = 8 (for *C.a. arg4Δ/Δ*+STm: 24 h feces), n = 7 (for *C.a. arg4Δ/Δ + ARG4* + STm: 24 h and 48 h feces, small intestine and cecum), and n = 9 (for *C.a. arg4Δ/Δ*+STm: 48 h feces, small intestine and cecum) of 4 independent experiments. Two-way ANOVA for comparison. **h**, Arginine levels measured in the cecum content of STm and *C. albicans* SC5314 infected mice 24 h and 48 h p.i. Data are represented as median with range. n = 5 (for 48 h and STm+ *C.a. arg4Δ/Δ+ARG4* 24 h) and n = 4 (for STm + *C.a. arg4Δ/Δ* 24 h) animals. Two-tailed unpaired t-test for comparison. ns = not significant. Bars with no statistics have p > 0.9999. STm, *Salmonella;* CFU, colony-forming units.

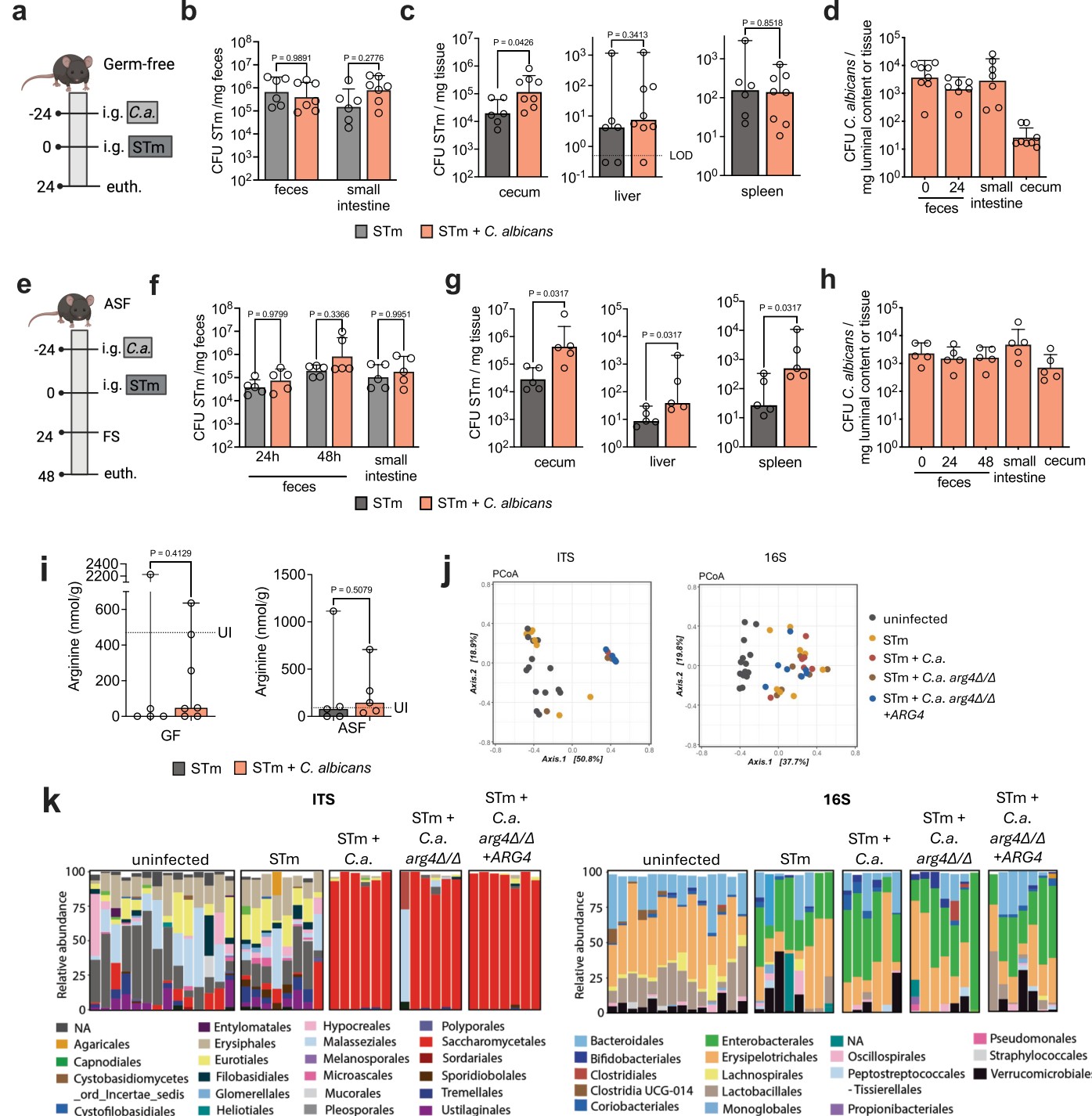

**Extended Data Fig. 5 | Gut microbiota does not play a major role in enhancing *Salmonella* virulence in the presence of *C. albicans*. a**, Schematic representation of experimental setup for STm or STm and *C. albicans* infection in germ-free mice. **b,c,d**, STm and *C. albicans* colonization in C57BL/6 germ-free mice infected with STm or STm and *C. albicans* SC5314 for 24 h p.i. Data are geometric mean ± s.d. (for feces, small intestine and cecum) and median with range (for liver and spleen). n = 6 (for STm in **b,c**), n = 7 (for STm + *C. albicans* in **b** and 24 h feces and small intestine in **d**) and n = 8 (for STm + *C. albicans* in **c** and 0 h feces and cecum in **d**) animals from 3 independent experiments. Two-way ANOVA (for **b**), two-tailed Mann–Whitney test (for **c**) for comparison. **e**, Schematic representation of experimental setup for STm or STm and *C. albicans* infection in ASF mice. **f,g,h**, STm and *C. albicans* colonization in C57BL/6 ASF mice infected with STm or STm and *C. albicans* SC5314 for 48 h p.i. Data are geometric

mean ± s.d. (for feces, small intestine and cecum) and median with range (for liver and spleen). n = 5 animal from 2 independent experiments. Two-way ANOVA (for **f**) and two-tailed Mann–Whitney test (for **g**) for comparison. **i**, Arginine levels measured in the cecum content of STm and *C. albicans* SC5314 infected germ-free and ASF mice 24 h and 48 h p.i., respectively. Data are represented as median with range. n = 5 (for GF: STm and ASF: STm and STm+ *C. albicans*) and n = 7 (for GF: STm+ *C. albicans*). Two-tailed Mann-whitney t-test for comparison. Dashed line indicates levels in uninfected mice **j**, PCoA plots of 16S and ITS sequencing data. DNA was extracted from fecal samples of C57BL/6 mice infected with STm or STm and *C. albicans* SC5314 and fecal samples were collected at 48 h p.i. **k**, Relative abundance of bacterial and fungal orders present in samples in **j**. i.g., intra-gastrically; STm, *Salmonella*; euth, euthanasia; CFU, colony-forming units; GF, germ-free; ASF, Altered Schaedler flora.

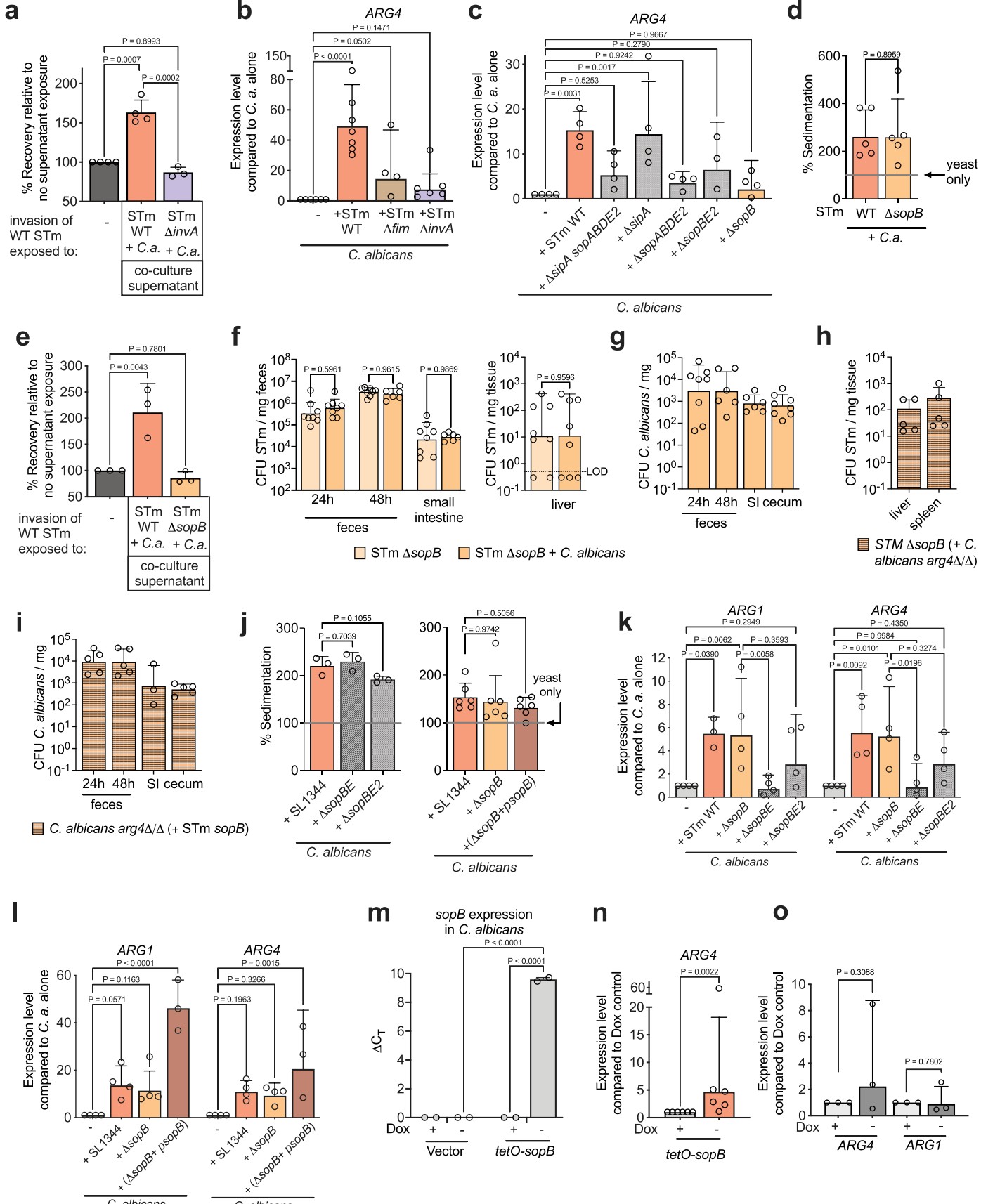

**Extended Data Fig. 6** | See next page for caption.

**Extended Data Fig. 6 | *Salmonella* uses T3SS-1 effector(s) to trigger arginine production in *C. albicans*. a**, Invasion assay of STm infected (MOI = 1) colonic epithelial cells (Caco2). STm either alone or with the supernatant from *C. albicans* ATCC containing cultures were incubated for 2 h prior to the assay. Data are represented as geometric mean ± s.d. n = 4 (for STm WT alone and with coculture supernatant of STm WT + *C.a.*) and n = 3 (for STm WT with coculture supernatant of STm Δ*invA* + *C.a.*) independent experiments. Ordinary one-way ANOVA for comparison. **b & c**, qRT-PCR analysis of genes encoding for *C. albicans* arginine biosynthesis from *C. albicans* SC5314 or STm and *C. albicans* SC5314 cultures incubated for 2 h. Data are geometric mean ± s.d. In **b**, n = 7 (for *C. albicans*, *C. albicans* + STm WT), n = 4 (for *C. albicans* + STm Δ*fim*) and n = 6 (for *C. albicans* + STm Δ*invA*) and in **c**, n = 3 (for *C. albicans* + STm Δ*sopBE2*) and n = 4 (for other groups) independent experiments. Kruskal-Wallis test (for **b**) and Ordinary one-way ANOVA (for **c**) for comparison. **d**, Sedimentation assay of STm and *C. albicans* SC5314. Data are geometric mean ± s.d. n = 5 independent experiments. Two-tailed unpaired t-test for comparison. **e**, Invasion assay of STm infected (MOI = 1) colonic epithelial cells (Caco2). STm either alone or with the supernatant from *C. albicans* SC5314 containing cultures were incubated for 2 h prior to the assay. Data are geometric mean ± s.d. n = 3 independent experiments. Ordinary one-way ANOVA for comparison. **f,g**, STm and *C. albicans* colonization in C57BL/6 mice infected with STm or STm and *C. albicans* SC5314 in the streptomycin pre-treatment model for 48 h p.i. Data are geometric mean ± s.d. In **f**, n = 8 (for 24 h feces, liver and cecum, and for 48 h STm Δ*sopB* feces and small intestine) and n = 6 (for 48 h STm Δ*sopB* + *C. albicans* feces and small intestine) animals and in **g**, n = 8 (for cecum), n = 7 (for 24 h feces) and n = 5 (for 48 h feces and small intestine) from 3 independent experiments. Two-way

ANOVA (for feces and small intestine) and Kruskal-Wallis test (for liver) are used for comparison. **h,i**, STm and *C. albicans* colonization in C57BL/6 mice infected with STm and *C. albicans* SC5314 in the streptomycin pre-treatment model for 48 h p.i. Data are geometric mean ± s.d. n = 5 (for 24 h and 48 h feces, cecum, liver and spleen), n = 3 (for 48 h small intestine), animals from 3 independent experiments. **j**, Sedimentation assay of STm SL1344 and *C. albicans* SC5314. Data are geometric mean ± s.d. n = 3 (for left panel) and n = 6 (for right panel) independent experiments. Ordinary one-way ANOVA for comparison. **k,l**, qRT-PCR analysis of genes encoding for *C. albicans* arginine biosynthesis from the *C. albicans* SC5314 or STm SL1344 and *C. albicans* SC5314 cultures incubated for 2 h. Data are geometric mean ± s.d. In **k**, n = 3 (for *ARG1 C. albicans* + STm) and n = 4 (for other groups) independent experiments and in **l**, n = 3 (for *C. albicans* + STm Δ*sopB* + p*sopB*) and n = 4 (for other groups) independent experiments. Mixed effect analysis (for **k**) and Two-way ANOVA (for **l**) for comparison. **m**, qRT-PCR analysis to confirm *sopB* expression in *C. albicans* SC5314 cultures expressing vector control or *tetO-sopB*. Data are mean ± SEM. n = 2 independent experiments. Two-way ANOVA for comparison. **n**, qRT-PCR analysis of genes encoding for *C. albicans* arginine biosynthesis from the *C. albicans* SC5314 cultures expressing *tetO-sopB* incubated for 6 h with or without doxycycline. Data are geometric mean ± s.d. n = 6 independent experiments. Two-tailed Mann-Whitney t-test for comparison. **o**, qRT-PCR analysis of genes encoding for *C. albicans* arginine biosynthesis from the *C. albicans* SC5314 cultures expressing vector control incubated for 6 h with or without doxycycline. Data are geometric mean ± s.d. n = 3 independent experiments. Unpaired two-tailed t-test for comparison. Bars with no statistics have p > 0.9999. STm, *Salmonella*; CFU, colony-forming units; LOD, limit of detection.

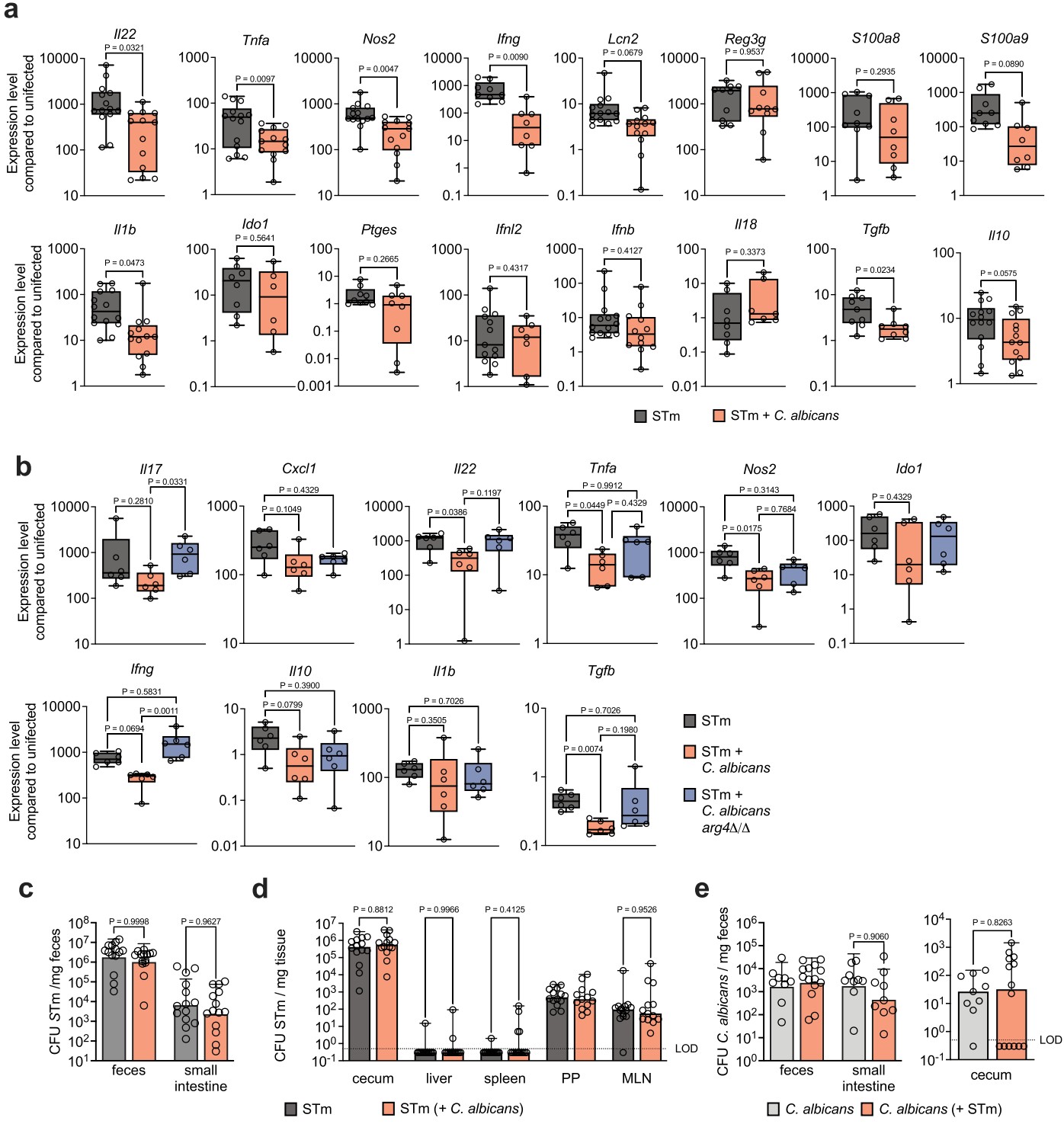

**Extended Data Fig. 7 | Immune response to *Salmonella* and *C. albicans* co-infection. a & b,** qRT-PCR analysis of genes encoding for host inflammatory response from the cecum tissue of (for **a**) STm or STm and *C. albicans* ATCC infected mice 24 h p.i. and (for **b**) STm or STm and *C. albicans* SC5314 infected mice 48 h p.i. In **a**, n = 16 (STm: *Il22*), n = 14 (STm: *Lcn2, Il1b, Ifnb, Il10, Tnfa, Nos*), n = 13 (STm: *Ifnl2* and STm + *C. albicans: Il22, Lcn2, Il1b, Il10, Tnfa, Nos*), n = 12 (STm: *Ifnb*), n = 10 (*Reg3g*), n = 9 (STm: *Ifng, S100a8, S100a9, Ptges, Tgfb*), n = 8 (STm: *Ido1, Il18* and STm + *C. albicans: Ifng, S100a8, S100a9, Ptges, Tgfb*), n = 7 (STm + *C. albicans: Ifnl2, Il18*), n = 6 (STm + *C. albicans: Ido1*) animals from 3 independent experiments. In **b**, n = 6 animals from 2 independent experiments. Data are presented as box and whiskers plot from 25th to 75th percentile, median,

minimum and maximum. Significance determined by two-tailed unpaired t-test (for **a**) and Kruskal-Wallis test (for **b**). **c,d,e,** STm and *C. albicans* colonization in C57BL/6 mice infected with STm, *C. albicans* ATCC, or STm and *C. albicans* ATCC in the streptomycin pre-treatment model for 24 p.i. Data are geometric mean ± s.d. (for fecal samples and SI content) and Median with range (for organ colonization). In **c** and **d**, n = 14 animals and in **e**, n = 9 (for *C. albicans*: all groups and *C. albicans* + STm: small intestine) and n = 14 (for *C. albicans* + STm: feces, cecum) animals from 3 independent experiments. Two-way ANOVA (for FS), mixed-effects analysis (for **d**) and Mann Whitney t-test (for cecum in **e**) for comparison. CFU, colony-forming units; STm, *Salmonella*; LOD, limit of detection.

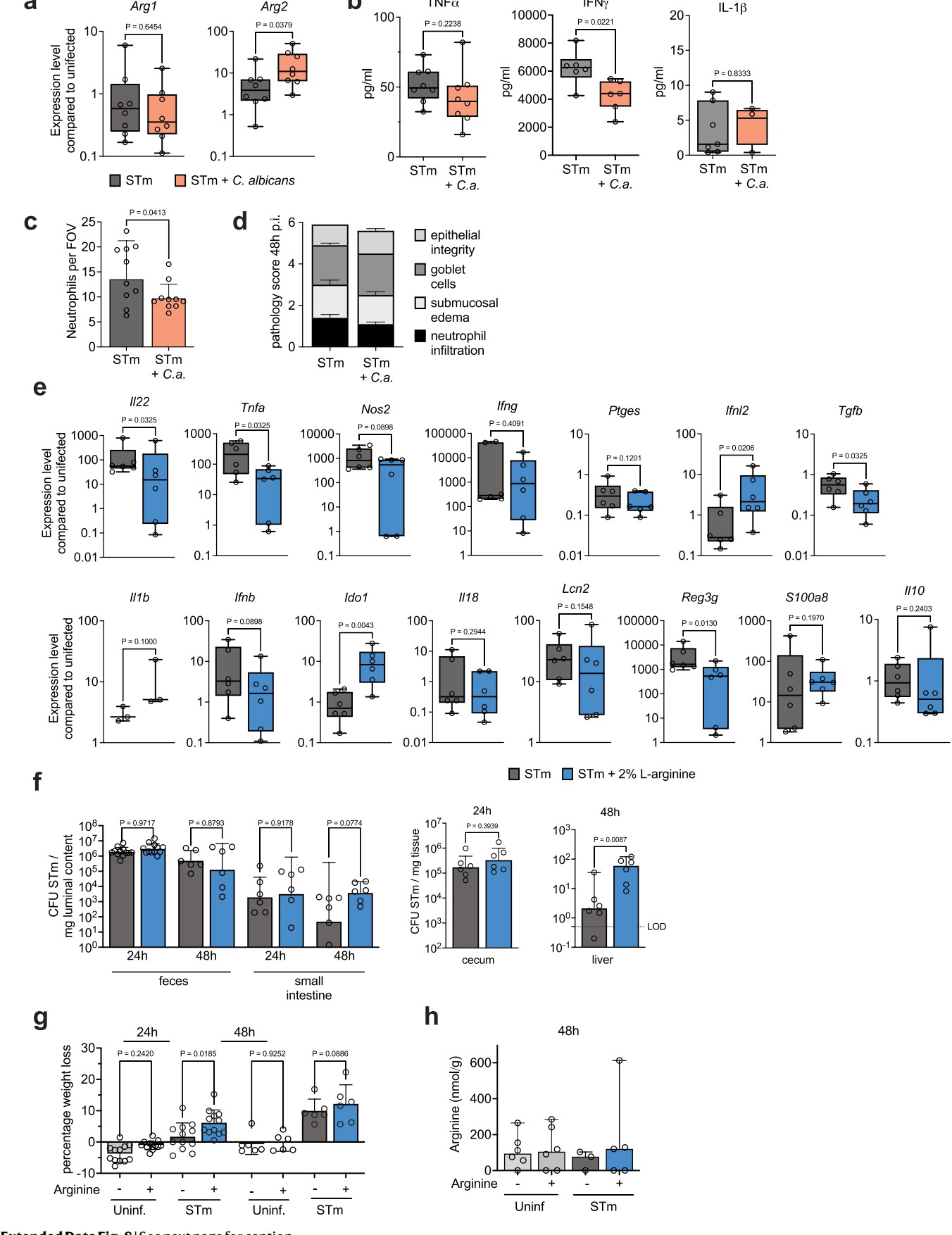

**Extended Data Fig. 8** | See next page for caption.

**Extended Data Fig. 8 | Effect of the presence of *C. albicans* and arginine supplementation during *Salmonella* infection. a**, qRT-PCR analysis of genes encoding for host arginase genes from the cecum tissue of STm or STm and *C. albicans* ATCC infected mice 24 h p.i. n = 8 animals from 2 independent experiments. Significance determined by two-tailed Mann–Whitney test. **b**, ELISA to detect cytokines from the serum of STm or STm and *C. albicans* SC5314 infected mice. n = 8 (for Tnfa), n = 7 (STm: IL-1b and Stm + *C. albicans*: IFNg), n = 6 (for IFNg STm) and n = 4 (for IL-1b STm + *C. albicans*) animals from 2 independent experiments. Data are presented as box and whiskers plot from 25th to 75th percentile, median, minimum and maximum. Significance determined by two-tailed Mann–Whitney test. **c**, Neutrophils counts for cecum 48 h p.i. from STm or STm and *C. albicans* ATCC infected mice. Data are geometric mean ± s.d. n = 10 animals. Welch's t-test for comparison. **d**, Pathological scores in cecum of STm or STm and *C. albicans* ATCC infected mice 48 h p.i. Data are mean ± SEM. n = 10 animals. **e**, qRT-PCR analysis of genes encoding for host inflammatory response from the cecum tissue of STm infected mice with or without L-arginine supplementation 24 h p.i. n = 3 (for *Il1b*) and n = 6 (for other genes) animals from 2 independent experiments. Data are presented as box and whiskers plot from 25th to 75th percentile, median, minimum and maximum. Significance determined by two-tailed Mann–Whitney test. **f,g**, STm colonization and Weight loss in STm or STm with L-arginine treated C57BL/6 mice in the streptomycin pre-treatment model for 24 h and 48 h p.i. Data are mean ± s.d. (for weight loss) and geometric mean ± s.d. (for fecal samples, SI content and cecum). n = 6 animals from 2 independent experiments. Mixed-effects analysis (for weight loss, fecal samples and SI content) and two-tailed Mann–Whitney test (for cecum) for comparison. **h**, Arginine levels measured in the cecum content of STm or STm with L-arginine treated mice 48 h p.i. Data are median with 95% CI. n = 6 (for uninfected), n = 5 (for STm+ arginine) and n = 3 (for STm) animals. Kruskal-Wallis test for comparison. Bars with no statistics have p > 0.9999. STm, *Salmonella*; FOV, field of view, CFU, colony-forming units.

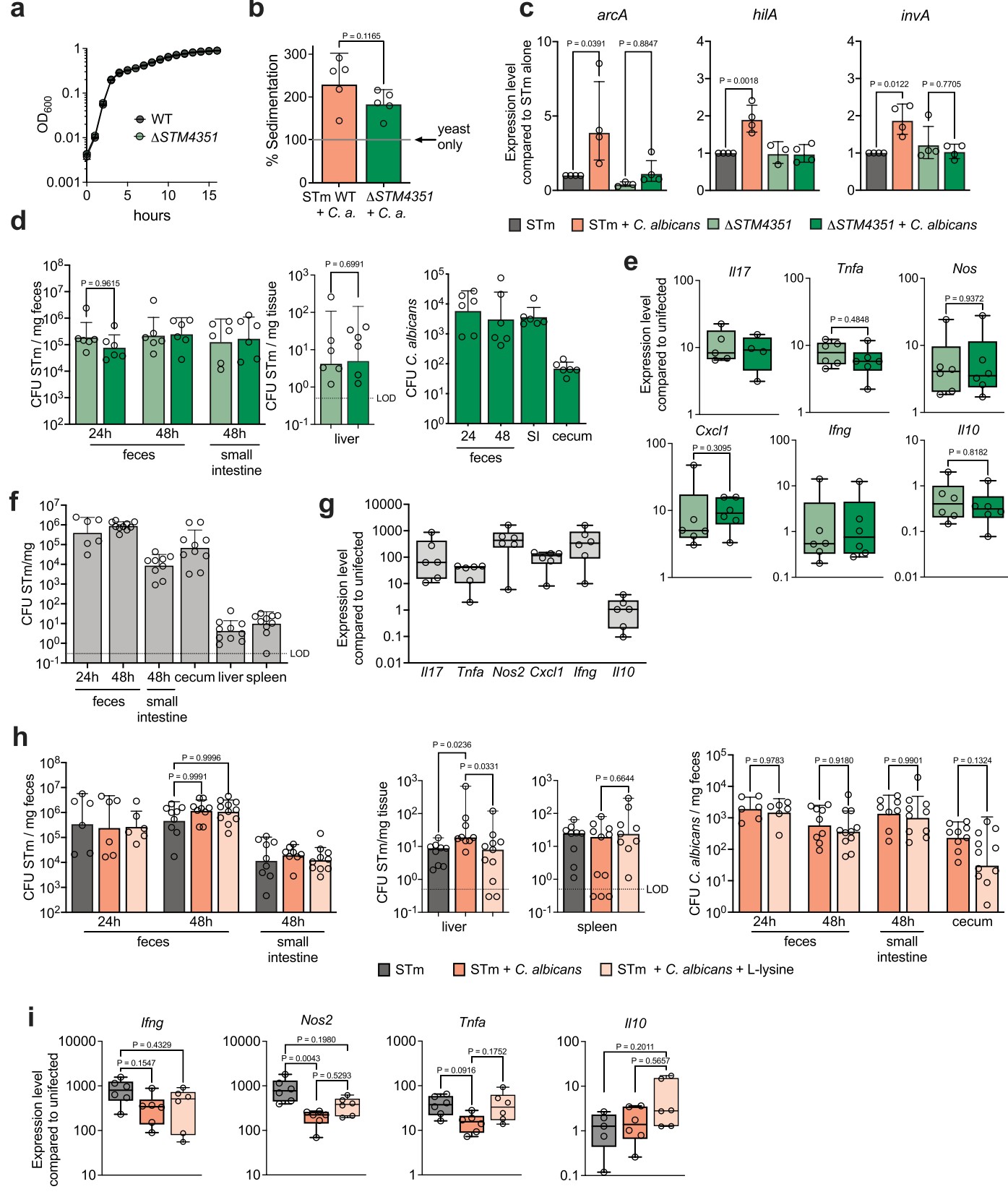

**Extended Data Fig. 9** | See next page for caption.

**Extended Data Fig. 9 | Arginine transport by *Salmonella* and host is required for *C. albicans* to affect *Salmonella* virulence. a**, Growth curve of STm WT and Δ*STM4351* strains in LB broth. Both curves overlapped. **b**, Sedimentation assay of STm strains and *C. albicans* SC5314. Data are geometric mean ± s.d. n = 5 independent experiments. Two-tailed unpaired t-test for comparison. **c**, qRT-PCR analysis of STm genes from the STm strains either alone or with *C. albicans* SC5314 cultures incubated for 2 h. Data are geometric mean ± s.d. n = 3 (for STm Δ*STm4351 arcA* and STm Δ*STM4351 hilA*) and n = 4 (for other groups). Ordinary one-way ANOVA for comparison. **d**, STm and *C. albicans* colonization in C57BL/6 mice infected with STm or STm and *C. albicans* SC5314 in the streptomycin pre-treatment model for 48 h p.i. Data are geometric mean ± s.d. n = 6 animals from 2 independent experiments. Two tailed Mann-Whitney t-test (for liver) and Two-way ANOVA (for fecal sample) for comparison. **e**, qRT-PCR analysis of genes encoding for host inflammatory response from the cecum tissue of STm or STm and *C. albicans* SC5314 infected mice 48 h p.i. n = 5 (for Δ*STm4351*: *Il17*), n = 4 (for Δ*STm4351* + *C. albicans*: *Il17*) and n = 6 (for other groups) animals from 2 independent experiments. Data are presented as box and whiskers plot from 25th to 75th percentile, median, minimum and maximum. Significance determined by two-tailed Mann-Whitney t-test. **f**, STm colonization in C57BL/6 mice infected with STm and treated with 20mM ʟ-lysine for 48 h p.i. Data are geometric mean ± s.d. n = 6 (for 24 h feces), n = 9 (for small intestine) and n = 10 (for 48 h feces, cecum, liver and spleen) animals from 3 independent experiments.

**g**, qRT-PCR analysis of genes encoding for host inflammatory response from the cecum tissue of STm infected mice treated with 20mM ʟ-lysine for 48 h p.i. n = 6 animals from 2 independent experiments. Data are presented as box and whiskers plot from 25th to 75th percentile, median, minimum and maximum. **h**, STm and *C. albicans* colonization in C57BL/6 mice infected with STm or STm and *C. albicans* SC5314 in the streptomycin pre-treatment model in the presence and absence of 20mM ʟ-lysine for 48 h p.i. Data are geometric mean ± s.d. n = 6 (for 24 h feces), n = 9 (for 48 h feces: STm, STm + *C. albicans*; small intestine: STm; liver: STm, STm + *C. albicans*; spleen: STm, STm + *C. albicans*; cecum: STm + *C. albicans*), n = 8 (for small intestine: STm + *C. albicans*), n = 10 (for small intestine: STm + *C. albicans* + ʟ-lysine), and n = 11 (for 48 h feces: STm + *C. albicans* + ʟ-lysine; liver: STm + *C. albicans* + ʟ-lysine; spleen: STm + *C. albicans* + ʟ-lysine; cecum: STm + *C. albicans* + ʟ-lysine) animals from 3 independent experiments. Two-way ANOVA (for feces and cecum), Kruskal-Wallis test (liver and spleen) for comparison. **i**, qRT-PCR analysis of genes encoding for host inflammatory response from the cecum tissue of mice infected with STm or STm and *C. albicans* SC5314 with or without ʟ-lysine for 48 h p.i. n = 4 (for STm *Il10*) and n = 6 (for other groups) animals from 2 independent experiments. Data are presented as box and whiskers plot from 25th to 75th percentile, median, minimum and maximum. Significance determined by Kruskal-Wallis test. Bars with no statistics have p > 0.9999. STm, *Salmonella*; CFU, colony-forming units; LOD, limit of detection.

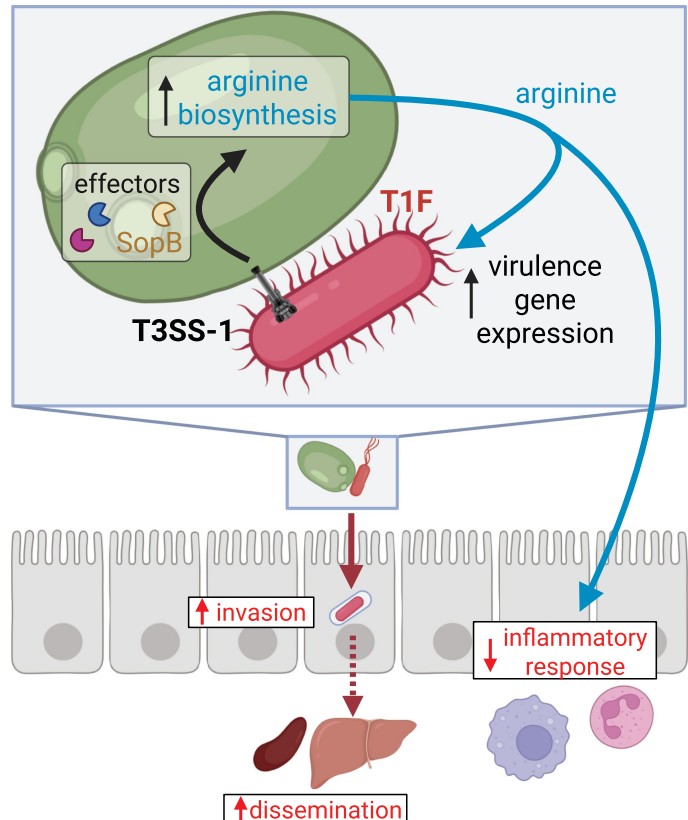

**Extended Data Fig. 10 | Proposed model of interactions between *Salmonella*, *C. albicans*, and the host during infection.** *Salmonella* binds to *C. albicans* via Type 1 fimbriae (T1F) and uses its Type 3 Secretion System (T3SS)-1 to deliver effector molecules, including SopB, into *C. albicans*. The effector SopB increases arginine biosynthesis in *C. albicans*, which is exported into the extracellular environment. *Salmonella* senses the released arginine and increases virulence gene expression, which results in increased invasion of epithelial cells. Arginine also decreases the inflammatory response to the infection, further facilitating *Salmonella* pathogenicity and dissemination to peripheral organs.

# Reporting Summary

## Statistics

For all statistical analyses, confirm that the following items are present in the figure legend, table legend, main text, or Methods section.

| n/a | Confirmed | |
|---|---|---|
| ☐ | ☒ | The exact sample size (*n*) for each experimental group/condition, given as a discrete number and unit of measurement |
| ☐ | ☒ | A statement on whether measurements were taken from distinct samples or whether the same sample was measured repeatedly |
| ☐ | ☒ | The statistical test(s) used AND whether they are one- or two-sided *Only common tests should be described solely by name; describe more complex techniques in the Methods section.* |
| ☐ | ☒ | A description of all covariates tested |
| ☐ | ☒ | A description of any assumptions or corrections, such as tests of normality and adjustment for multiple comparisons |
| ☐ | ☒ | A full description of the statistical parameters including central tendency (e.g. means) or other basic estimates (e.g. regression coefficient) AND variation (e.g. standard deviation) or associated estimates of uncertainty (e.g. confidence intervals) |
| ☐ | ☒ | For null hypothesis testing, the test statistic (e.g. *F*, *t*, *r*) with confidence intervals, effect sizes, degrees of freedom and *P* value noted *Give P values as exact values whenever suitable.* |
| ☒ | ☐ | For Bayesian analysis, information on the choice of priors and Markov chain Monte Carlo settings |
| ☒ | ☐ | For hierarchical and complex designs, identification of the appropriate level for tests and full reporting of outcomes |
| ☒ | ☐ | Estimates of effect sizes (e.g. Cohen's *d*, Pearson's *r*), indicating how they were calculated |

*Our web collection on statistics for biologists contains articles on many of the points above.*

## Software and code

Policy information about availability of computer code

| Data collection | No custom code was used in this study. |
|---|---|
| Data analysis | No custom code was used in this study. |

For manuscripts utilizing custom algorithms or software that are central to the research but not yet described in published literature, software must be made available to editors and reviewers. We strongly encourage code deposition in a community repository (e.g. GitHub). See the Nature Portfolio guidelines for submitting code & software for further information.

## Data

Policy information about availability of data

All manuscripts must include a data availability statement. This statement should provide the following information, where applicable:
- Accession codes, unique identifiers, or web links for publicly available datasets
- A description of any restrictions on data availability
- For clinical datasets or third party data, please ensure that the statement adheres to our policy

Raw sequence reads have been deposited at NCBI Sequence Read Archive under project PRJNA1143068, PRJNA1255633, and PRJNA1285498 and the UNITE (v.8.2) database64 was used for all microbiome analysis.

# Research involving human participants, their data, or biological material

Policy information about studies with [human participants or human data](). See also policy information about [sex, gender (identity/presentation), and sexual orientation]() and [race, ethnicity and racism]().

| | |
|---|---|
| Reporting on sex and gender | Human research participants is not applicable to this study |
| Reporting on race, ethnicity, or other socially relevant groupings | Human research participants is not applicable to this study |
| Population characteristics | Human research participants is not applicable to this study |
| Recruitment | Human research participants is not applicable to this study |
| Ethics oversight | Human research participants is not applicable to this study |

Note that full information on the approval of the study protocol must also be provided in the manuscript.

# Field-specific reporting

Please select the one below that is the best fit for your research. If you are not sure, read the appropriate sections before making your selection.

☒ Life sciences  ☐ Behavioural & social sciences  ☐ Ecological, evolutionary & environmental sciences

For a reference copy of the document with all sections, see [nature.com/documents/nr-reporting-summary-flat.pdf]()

# Life sciences study design

All studies must disclose on these points even when the disclosure is negative.

| | |
|---|---|
| Sample size | No sample-size calculation was performed. Sample sizes were chosen based on past experience using the methods described (see Santus et al, Nature Microbiology 2022; Behnsen et al, Immunity 2014; Raffatellu et al, Cell Host Microbe 2009; Deriu et al 2013, Cell Host Microbe). Each in vitro experiment was repeated with at least 3 individual biological replicates. Mouse experiments were performed with 2-6 mice per group and repeated for a minimal number of 5 mice per experimental condition. |
| Data exclusions | As mice are not a natural host for C. albicans, some mice did not maintain C. albicans cecum colonization during the experiment. These mice were excluded from further analysis. |
| Replication | Each in vitro experiment was performed at least in triplicate. Mouse experiments were performed 2-4 times. All replication attempts performed under the same conditions were successful. |
| Randomization | Mice were received from vendors and randomly assigned to experimental groups. For in vitro experiments, samples could not be randomized, as each was a specific experimental condition that needed to be defined. |
| Blinding | Blinding was not performed. No measurements sensitive to subjective interpretation were performed. |

# Reporting for specific materials, systems and methods

We require information from authors about some types of materials, experimental systems and methods used in many studies. Here, indicate whether each material, system or method listed is relevant to your study. If you are not sure if a list item applies to your research, read the appropriate section before selecting a response.

## Materials & experimental systems

| n/a | Involved in the study |
|---|---|
| ☐ | ☒ Antibodies |
| ☐ | ☒ Eukaryotic cell lines |
| ☒ | ☐ Palaeontology and archaeology |
| ☐ | ☒ Animals and other organisms |
| ☒ | ☐ Clinical data |
| ☒ | ☐ Dual use research of concern |
| ☒ | ☐ Plants |

## Methods

| n/a | Involved in the study |
|---|---|
| ☒ | ☐ ChIP-seq |
| ☒ | ☐ Flow cytometry |
| ☒ | ☐ MRI-based neuroimaging |

# Antibodies

| | |
|---|---|
| Antibodies used | Goat anti-Rabbit IgG (H+L) Cross-Adsorbed ReadyProbes™ Secondary Antibody, Alexa Fluor™ 594; Invitrogen, Cat# R37117, Lot# 1938330<br>FITC-labeled rabbit anti-fungal antibody; Meridian Life Science, Cat# B65411F, Lot# 2D11517 |
| Validation | Both antibodies used in this study were validated by the manufacturers. Comprehensive validation statements, alongside relative details such as species reactivity, applications, and supporting citations, are accessible through the provided website links. Essential information pertaining to the antibodies employed in this study are outlined below:<br><br>Goat anti-Rabbit IgG (H+L) Cross-Adsorbed ReadyProbes™ Secondary Antibody, Alexa Fluor™ 594; Invitrogen<br>Species reactivity: Rabbit<br>Application (in this study): Immunocytochemistry (ICC/IF)<br>Validation statements and other details (including citations): https://www.thermofisher.com/antibody/product/Goat-anti-Rabbit-IgG-H-L-Cross-Adsorbed-Secondary-Antibody-Polyclonal/R37117<br><br>FITC-labeled rabbit anti-fungal antibody; Meridian Life Science<br>Application (in this study): Immunocytochemistry (ICC/IF)<br>Validation statements and other details (including citations): https://www.meridianbioscience.com/uploads/ls-uploads/coa/B65411F.pdf |

# Eukaryotic cell lines

Policy information about cell lines and Sex and Gender in Research

| | |
|---|---|
| Cell line source(s) | T84 colonic epithelial cell line; ATCC Cat# CCL-248; human carcinoma cell line derived from a lung metastasis of a colon carcinoma in a 72-year-old male; RRID:CVCL_0555<br><br>C2BBe1 [clone of Caco-2]; ATCC Cat# CCL-2102; enterocytes isolated from the large intestine (colon) of a patient with colorectal adenocarcinoma; RRID:CVCL_1096 |
| Authentication | Cell lines used were not authenticated |
| Mycoplasma contamination | Both T84 and C2BBe1 cell lines tested negative for mycoplasma contamination |
| Commonly misidentified lines<br>(See ICLAC register) | Commonly misidentified cell lines were not used |

# Animals and other research organisms

Policy information about studies involving animals; ARRIVE guidelines recommended for reporting animal research, and Sex and Gender in Research

| | |
|---|---|
| Laboratory animals | conventional C57BL/6 mice 8-9 weeks old ; gnotobiotic Swiss Webster mice 12-15 weeks old ; CBA/J mice 8-9 weeks old, gnotobiotic C57BL/6 mice 8-10 week old |
| Wild animals | The study did not involve wild animals. |
| Reporting on sex | Both male and female mice were used in the study. |
| Field-collected samples | The study did not involve samples collected from the field. |
| Ethics oversight | All animal experiments were reviewed and approved by the Institutional Animal Care and Use Committee at the University of Illinois Chicago (protocols 17-045, 20-016 and 22-192). |

Note that full information on the approval of the study protocol must also be provided in the manuscript.

# Plants

Seed stocks

N/A- This study do not involve plants

Novel plant genotypes

N/A- This study do not involve plants

Authentication

N/A- This study do not involve plants

