## [Peer Review file · Nature]

Commensal Yeast Promotes Salmonella Typhimurium Virulence

Corresponding Author: Dr Judith Behnsen

Version 0:

Reviewer comments:

Referee #1

(Remarks to the Author)

This study by Jaswal et al. demonstrates a bidirectional relationship between the commensal (and opportunistic pathogen) *Candida albicans* and the gut pathogen *Salmonella enterica* serovar Typhimurium. This work stems from a finding that fecal abundance of *Candida* spp were correlated with *Salmonella* dissemination. Consistent with this observation, animals inoculated with *Salmonella* and *Candida* together had greater weight loss and increased *Salmonella* burden. Delving further, the authors used in vitro infection of colonic epithelial cell lines to show direct interaction dependent on *Salmonella* Type 1 Fimbriae binding to mannose residues. *Salmonella* genes upregulated in response to coculture with *Candida* included T3SS structural proteins and genes encoding proteins that contribute to arginine catabolism and transport. Indeed, arginine levels are increased in *salmonella* and *candida* cocultures, and *Candida* strains deficient in arginine production were unable to increase *Salmonella* invasion or dissemination. *Candida* arginine production was then shown to require a functional T3SS and effectors SopB (and SopE). Finally, the authors show a blunted immune response in animals exposed to *Salmonella* and *Candida* but not *salmonella* alone.

These findings represent a novel mechanism for *Salmonella* pathogenicity in which the T3SS and its effectors increase arginine biosynthesis by *Candida* early during infection, which subsequently enhances *Salmonella* colonization and invasion. I found the sequence of events to be convincing and appreciated the genetic approaches that allowed mixing of *Candida* and *Salmonella* mutants. Addressing a few specific points can further support their conclusions.

Specific points:

1. I found the proposed effect of *Candida*-derived arginine on host immunity to be problematic for two reasons. First, it is unclear whether the gene expression changes are functionally important and contributing to increased *Salmonella* burden. Second, if *Candida* is dampening host immunity, as the authors suggest, then it is possible that the effect of fungal colonization on *Salmonella* burden does not involve direct action of arginine on the bacterium in vivo. This three-way interaction is tough to tease apart in the animal experiments but is also the crux of the manuscript. Can the authors use a *Salmonella* mutant that is unresponsive to exogenous arginine to determine the extent to which *Candida*-derived arginine acts on the bacterial pathogen versus the host in vivo?
2. Related to the above point, could *Candida*/arginine be acting by changing microbiota composition or factors produced by the microbiota? Is *Salmonella* the main or only consumer of *Candida*-derived arginine (potentially due to its physical association)? It would be helpful if the authors can find a way to increase confidence that the arrow coming back out of *Candida* is really directly connecting back to *Salmonella*.
3. The authors were unable to detect alterations in arginine levels in the cecum. Could this be due to timing or perhaps intracellular vs extracellular levels?
4. I was confused by the schematic in fig 3c, where at 24h, there is a time point for "FS and ig C.a.". Did the authors give a second gavage of *candida* to these animals? If so, any specific reason?

Referee #2

(Remarks to the Author)

In this exciting manuscript, Jaswal et al describe a novel mechanism of how commensal yeast, as part of the gut microbiome, interacts with a bacterial pathogen. The initial observation is that oral infection of mice with *Salmonella Typhimurium* results in blooms of *Candida* in the gut, and *Candida* blooms correlate with increased disease severity. This is recapitulated through experimental *Candida-Salmonella* co-infection in mice. Using an in vitro culture system, the authors investigate potential interactions between *Candida* and *Salmonella*, and find that *Salmonella* binds to yeast using type 1 fimbriae. Furthermore, they discover that a secreted molecule, later to be revealed to be L-arginine, enhances *Salmonella* invasiveness and expression of the invasion-associated type three secretion (SPI1). A *Candida* mutant unable to produce arginine is unable to increase *Salmonella* virulence in cell culture and in mouse models. Curiously, production of the secreted effector protein SopB was necessary for *Salmonella* to induce arginine release by *Candida*. Furthermore, *Candida* colonization and L-arginine supplementation reduce inflammatory responses towards *Salmonella*, and result in increased dissemination.

This study sheds light on to a very understudied aspect of the microbiome, the mycobiome, and its effect on disease processes. As such, the discovery that *Candida* and *Salmonella* interact metabolically through L-arginine cross-feeding is timely, exciting, and markedly extends our knowledge on the gut microbiome. Most exciting to me is the fact that *Salmonella* appears to be using a type three secretion system effector to manipulate *Candida*; to my knowledge, this is the first evidence that non-mammalian cells are targeted by the *Salmonella* type three secretion system. This observation alone should open up new avenues of research. The overall conclusions are well-supported by the data. The manuscript is well written and easy to follow for a general audience.

There are some minor concerns regarding the detailed mechanism(s) that may require the authors' attention.

1. The experiment shown in Fig. 3 demonstrates that arginine production by *Candida* is necessary to enhance *Salmonella* virulence. Based on Fig 2, the authors argue that L-arginine is a cue for the invasion-associated type three secretion system, while Fig. 5 suggests that arginine supplementation changes inflammatory responses. This raises the question as to whether the phenotype reported in Fig. 3 is due to altered virulence gene regulation, or due to altered immune responses. How does arginine influence *S. Tm* infection in vivo, directly via gene regulation of virulence factors, or indirectly via the host response? While this concern does not chip away from the conceptual novelty outlined above, this key aspect of the mechanism still needs to be resolved.

2. Along these lines, it would be important to show that SopB is required for inducing L-arginine production by *C. albicans* in the mouse model. Since L-arginine levels do not substantially change (the authors argue that local levels might), the appropriate readout might be *S. Tm* colonization. The prediction would be that in the absence of SopB, the presence of *Candida* or the Arg4 mutant would not influence invasiveness and colonization (see Fig 3d,e). Similarly, the authors should evaluate as to whether type one fimbriae are required for the in vivo phenotype.

Referee #3

(Remarks to the Author)

The manuscript by Jaswal et al examines interactions between *Salmonella* and *Candida* cells in the gastrointestinal tract. The authors reveal that *Candida* can exacerbate infection by *Salmonella*, and this involves the secretion of arginine from fungal cells which upregulates the Type 3 Secretion System (T3SS) in *Salmonella* and results in increased host invasion. *Salmonella* cells also enhance arginine secretion by *Candida* cells (as T3SS introduces the SopB effector protein into *Candida* cells), thereby effectively enhancing their own infectivity by this mechanism.

The paper is interesting in providing novel mechanistic insights into bacterial-fungal crosstalk in the gut, and the discovery that fungal-produced arginine can impact bacterial pathogenesis (as well as gut inflammation) is particularly intriguing. There were several areas, however, which were underdeveloped and should be addressed:

Major points:

1. The in vivo interaction data in the gut not overly convincing (Fig. 1f). There is really only one example where there appears to be a direct interaction between a bacterium and a fungal cell. Is there additional evidence (microscopic or otherwise) that these species physically interact in the gut? Is this interaction only occurring in the cecum or in other GI organs? Are filamentous *Candida* cells also present in the gut and interacting with *Salmonella* cells under these conditions (or in vitro)? These are key questions to be addressed.

2. For the experiment using *C. albicans* strain 529L in Fig. S1e it should be explicitly stated in the text and legend that antibiotics were not used in those experiments. Also, the levels of *Salmonella* in these mice only reached 10²-10³ CFU/mg of cecal tissue which is likely why they did not disseminate (compared to 10⁴-10⁵ in the antibiotic models) yet this difference is not mentioned/discussed? I also did not see the data showing that peak inflammation is reached at 9-10 days in this model? Does 529L also produce arginine in the presence of *Salmonella*?

3. For the sedimentation data in Fig. 2b, are the *C. albicans* cells filamenting in these assays or are they completely in the yeast state? Does filamentation influence interactions with *Salmonella* cells? The same question is true for the microscopy data in Fig. 2c? And were the sedimentation assays in each panel done on the same day? (as the WT + Ca data points in Fig. S2d looks identical to that in Fig. 2b).

4. The authors show that mannose interferes with fungal-bacterial interactions but it would be relevant to test other sugars or fungal cell wall components to test the specificity of this inhibition. And does adding mannose to the drinking water of mice influence the STm phenotype (while realizing that this may be confounded by other effects of mannose on the GI environment?)

5. In some places the statistical comparisons between experimental groups should include additional comparisons. For example, in Fig 2e, is there a significant difference between STm+Ca co-culture and STm fim+Ca groups? Similarly, later on in Fig. S5b, is there a significant difference in expression of *invA* between STm+Ca and STm + Ca(*arg4*) groups? Also in Fig S5c, is Ca(*arg4*) or Ca(+ARG4) significantly different from the yeast only control for sedimentation?

6. The authors use *Candida* strains ATCC and SC5314 extensively throughout the paper but in some places use ATCC and other places use SC5314. For example, in Fig. 2 the invasion assays are done with ATCC and the adherence assays with SC5314. While the use of multiple *Candida* strains is a strength of the manuscript, it does seem necessary to perform key experiments with both strains to show they have the same phenotypes. Does SC5314 also increase STm invasion similar to ATCC? And does ATCC show the same physical interactions with STm cells as SC5314? (The authors do show data with both *Candida* strains for Fig. S2j but not for most experiments).

7. In Fig 3 the authors indicate that the STm + Ca(*arg4*) mutant does not increase STm colonization (Fig 3) and invasion. But is the *arg4* mutant substantially different from the WT Ca in Fig 3D? Can the authors include the ARG4 complemented strain for the in vivo experiment as well as the in vitro data? And even the WT Ca strain did not significantly increase STm levels in the spleen in this data set (no data for the liver was shown) which does raise the question about the scale and reproducibility of the *Candida*-enhanced STm phenotype.

8. On line 351, the authors state that "Upon removal of doxycycline *C. albicans* ARG1 and ARG4 genes were induced (Fig. 4h and S6h), confirming that presence of SopB in *C. albicans* results in upregulation of arginine biosynthesis." However, there are no significant differences in gene expression in Fig. S6h and in Fig. 4h there is no comparison between the +/- dox samples for the tetO-sopB strain.

9. For the data in Figure S7, the authors state that inflammatory genes are higher with the Ca(*arg4*) strain than the Ca(WT) strain but the statistical analysis does not directly compare these groups (only compares them to the no *Candida* control). A direct comparison is again required here.

10. Does addition of arginine in the drinking water alter the levels of this amino acid in the cecum? Does arginine only alter inflammatory responses in the cecum or also in other host tissues?

11. Line 446 of the discussion says "yeast cells" but should say *Candida* cells as it is not clear that SopB would have the same effect on *S. cerevisiae* cells.

12. In terms of the overall model, is the decrease in inflammation thought to drive increased colonization by *Salmonella* in the cecum? And is increased invasion envisaged to be the result of both increased colonization and increased T3SS-1 expression? Clarifying these points in the model (Fig. S9) would help the reader.

13. I was not able to access the find the raw sequence data at NCBI or project PRJNA1143068.

14. It is essential to also provide catalog numbers for key resources like the antibodies used in the study.

Minor points:

-How exactly was *Candida albicans* identified as the "contaminant" in the mice in Fig. 1b,c?

-In many cases it is says that *Salmonella* and *C.albicans* are used in a 10:1 ratio but it is not clear which is in excess – this should be clarified throughout.

- The colors used in many places are confusing. For example, in Fig S1, the colors used for *C. albicans* cells and for *Salmonella* cells are very similar in some of the panels (light gray v. dark gray). Fig. S1a,b has no key (although it is assumed to be the same as in Fig. S1c?)

- a schematic would for fig2a to show how bacteria/fungal cells were cultured and tested would be a benefit.

- sentence starting on Line 191 is confusing "In either case....."

Version 1:

Reviewer comments:

Referee #1

(Remarks to the Author)

The authors have successfully addressed previous concerns. I particularly appreciated the completeness of the response and believe the evidence in the mouse model is substantially stronger in the revised manuscript.

Referee #2

(Remarks to the Author)

The authors have adequately addressed my concerns.

Referee #3

(Remarks to the Author)

The revised manuscript by Jaswal et al. has addressed the major concerns of this reviewer and the authors should be commended for their additional experiments and analyses that have added to the robustness of their findings. Some minor points are listed below and I am grateful that the authors have gone to considerable lengths to address the points of the three reviewers.

1. On line 169 the text states that "presence of *C. albicans* increased *Salmonella* colonization in the cecum (Fig. S1h)". However, given that this difference is only significant at one time point (5 dpi) I would add this to the text to be transparent.
2. Could the authors alter the contrast in the images in Fig. 2g to show the yeast cells more clearly as these are hard to see? Also, the panel on the right seems to show that there are both T3SS-positive and -negative cells in close contact with *Candida* cells? How do the authors account for this? Another way to look at this could be to use their sedimentation assay and test if sedimenting *Salmonella* cells that are presumably in closer contact with *Candida* cells activate T3SS expression more than those cells left in suspension.
3. The authors use a yeast only control to normalize the sedimentation data but does the presence of *Salmonella* also influence the OD reading (independent of its effect on sedimentation?). This will not change the relative results of the assays but could affect how the absolute numbers look?
4. Line 482. I would make clearer in the text that lysine can be used as a competitive inhibitor of arginine uptake (this is inferred but not 100% clear in this sentence).
5. One line 564, it could be emphasized that the activity of Arg82 / IPMK has diverged between fungi and mammalian cells? I believe that is what is being suggested here.
6. The authors state that *Candida* cells are mostly in the yeast state in the gut. Given the interest in fungal morphology and gut colonization I would suggest adding this statement to the main text. Also, it is very difficult to see hyphal cells in Fig. S2f – could the green signal be turned up to better see the fungal cells?
7. The authors state that mice of both sexes were used. It would be helpful to note in the figure legends which sexes were used for a specific experiment and the numbers of each involved.

RESPONSE TO REFEREES

Notes: Due to the extensive addition of additional data and limitations to allowable figure numbers, we had to delete the original figures S3 and S4 from the manuscript. The data can still be found in the manuscript, but they are now displayed in table form in S4 and S5.

Sequencing data reviewer links:

<https://dataview.ncbi.nlm.nih.gov/object/PRJNA1143068?reviewer=f7a5b02g47f7mp8012v8kff8> and <https://dataview.ncbi.nlm.nih.gov/object/PRJNA1255633?reviewer=nn1l2b38lt7tosdqbbkn7bp5gq>

REFEREE #1:

Referee expertise: gut microbiome, microbe-immune system interactions

This study by Jaswal et al. demonstrates a bidirectional relationship between the commensal (and opportunistic pathogen) *Candida albicans* and the gut pathogen *Salmonella enterica* serovar Typhimurium. This work stems from a finding that fecal abundance of *Candida* spp were correlated with *Salmonella* dissemination. Consistent with this observation, animals inoculated with *Salmonella* and *Candida* together had greater weight loss and increased *Salmonella* burden. Delving further, the authors used in vitro infection of colonic epithelial cell lines to show direct interaction dependent on *Salmonella* Type 1 Fimbriae binding to mannose residues. *Salmonella* genes upregulated in response to coculture with *Candida* included T3SS structural proteins and genes encoding proteins that contribute to arginine catabolism and transport. Indeed, arginine levels are increased in *salmonella* and *candida* cocultures, and *Candida* strains deficient in arginine production were unable to increase *Salmonella* invasion or dissemination. *Candida* arginine production was then shown to require a functional T3SS and effectors SopB (and SopE). Finally, the authors show a blunted immune response in animals exposed to *Salmonella* and *Candida* but not *salmonella* alone.

These findings represent a novel mechanism for *Salmonella* pathogenicity in which the T3SS and its effectors increase arginine biosynthesis by *Candida* early during infection, which subsequently enhances *Salmonella* colonization and invasion. I found the sequence of events to be convincing and appreciated the genetic approaches that allowed mixing of *Candida* and *Salmonella* mutants. Addressing a few specific points can further support their conclusions.

We thank the reviewer for their positive assessment of the novelty of our research and our approaches to dissect this cross-kingdom interaction.

Specific points:

1. I found the proposed effect of *Candida*-derived arginine on host immunity to be problematic for two reasons. First, it is unclear whether the gene expression changes are functionally important and contributing to increased *Salmonella* burden. Second, if *Candida* is dampening host immunity, as the authors suggest, then it is possible that the effect of fungal colonization on *Salmonella* burden does not involve direct action of arginine on the bacterium in vivo. This three-way interaction is tough to tease apart in the animal experiments but is also the crux of the manuscript. Can the authors use a *Salmonella* mutant that is unresponsive to exogenous arginine to determine the extent to which *Candida*-derived arginine acts on the bacterial pathogen versus the host in vivo?

We agree with the reviewer that such complex interactions with three different participants (or with the microbiota four participants, see our answer to point #2) are difficult to tease apart experimentally. Our experiments support the hypothesis that arginine affects virulence gene expression of *Salmonella* as well as the host response. However, as the reviewer points out,

there remained the possibility that responses are differentially important in an *in vivo* setting. We therefore aimed to assess the relative contribution of each to the overall phenotype we observed. We performed experiments in which we limited either host uptake of arginine or Salmonella uptake of arginine. In support of our hypothesis, these experiments showed intermediate phenotypes, suggesting that both the effect of arginine on the host as well as the effect on virulence gene expression in *Salmonella*, is important. Our detailed replies to individual comments are found below.

“I found the proposed effect of Candida-derived arginine on host immunity to be problematic for two reasons. First, it is unclear whether the gene expression changes are functionally important and contributing to increased Salmonella burden”

We agree with the reviewer that changes in gene expression do not necessarily result in functional differences. We therefore also measured systemic cytokine levels. While intestinal inflammation is required for *Salmonella* to efficiently colonize the gut lumen, it is known that inflammation is also required to limit pathogen dissemination to peripheral sites. Administration of IFN γ was shown to reduce *Salmonella* burden in the spleen

(Muotiala & Mäkelä, 1990). We analyzed cytokine levels in the blood of mice infected with either *Salmonella* alone or *Salmonella* and *C. albicans*. We found that cytokine levels were lower in mice co-infected with *Salmonella* and *C. albicans*. Specifically, IFN γ was significantly lower in the blood of mice infected with *Salmonella* and *C. albicans* compared to mice infected with *Salmonella* alone (Fig S8b). There was a trend for lower levels of TNF α and no change in IL-1 β (Fig S8b).

Fig. S8b (new). Cytokine levels in blood of mice collected 48h post infection with either *Salmonella* or *Salmonella* and *C. albicans*.

Fig. S8c. Neutrophils in the cecum of mice 48h post infection with either *Salmonella* or *Salmonella* and *C. albicans*.

We also examined neutrophil recruitment to the gut in our first submission. Neutrophil recruitment is an early and powerful response to *Salmonella* infection and absence of neutrophils exacerbates disease (e.g. (Gül et al., 2023)). A main signal recruiting neutrophils is a Cxcl-1 gradient. We observed significantly reduced Cxcl1 expression 24h post infection in co-infected mice and

statistically significant reduction of neutrophils in the cecum 48h post infection (Fig S8c). Reduced cytokine levels in the blood and neutrophil recruitment during co-infection thus indicate that the gene expression changes we observed are functionally important.

“it is possible that the effect of fungal colonization on Salmonella burden does not involve direct action of arginine on the bacterium in vivo. Can the authors use a Salmonella mutant that is unresponsive to exogenous arginine to determine the extent to which Candida-derived arginine acts on the bacterial pathogen versus the host in vivo?”

We thank the reviewer for this excellent suggestion. We initially thought this impossible to achieve, as *Salmonella* is known to require arginine transporters for full virulence in mice (Das et al., 2010; Margolis et al., 2023). However, the comment prompted us to examine

alternative options to dissect the relative importance of arginine on the pathogen versus the host. Our experiments led to exciting results that we think further enhances our manuscript.

SPI-1 activation in *Salmonella* cells close to *C. albicans*

Our microscopy images of the gut indicate that *Salmonella* and *C. albicans* are in close proximity and we postulated that the cells close to *C. albicans* would utilize arginine and upregulate SPI-1. While this is technically challenging to show *in vivo*, it is very feasible *in vitro* to test. We visualized SPI-1 induction on a single cell level, utilizing reporter construct in which the promoter of the T3SS-gene *prgH* regulates the expression of GFP. *Salmonella* and *C. albicans* were incubated for 2h on agarose pads, which limited the dispersal of arginine compared to liquid culture. We found a significant increase in *PprgH-gfp* positive cells in the presence of *C. albicans*, specifically in the cells in close proximity to *C. albicans* (Fig. 2g). Quantification shows a higher number of positive cells (9 fields of view) when *C. albicans* is present than when it is not (Fig S3d). We aimed to use the reporter strain *in vivo* but the high background fluorescence of the cecal lumen rendered these experiments impossible to interpret. Nevertheless, this *in vitro* experiment highlights that SPI-1 induction occurs in cells that are close to *C. albicans*. To test *in vivo* whether arginine utilization by *Salmonella* is required to enhance virulence in the presence of *C. albicans*, we performed the experiment that the reviewer suggested. We rendered *Salmonella* unresponsive to exogenous arginine, as described in the following.

Inhibiting *Salmonella* arginine uptake (Arginine transporter STM4351)

Deletion of the arginine transporter encoded by *artPIQM-artJ* results in *Salmonella* growth defects in mice (Margolis et al., 2023). Deletion of *argT*, encoding the periplasmic binding protein of the *argT-hisJQMP* gene cluster, resulted in reduced *Salmonella* survival in macrophages (Das et al., 2010). An *argT* mutant also showed reduced growth in a mouse model (Margolis et al., 2023) and reduced dissemination to the spleen and liver (Das et al., 2010). These confounding factors would make data generated using these arginine transporter uptake mutants impossible to interpret.

We therefore sought an alternative to inhibit or significantly reduce *Salmonella* arginine uptake. *Salmonella* highly upregulated expression of periplasmic binding protein involved in arginine transport (Stamp et al., 2011), STM4351, in the presence of *Candida in vitro*. The *Salmonella* deletion library screen in mice with >50 mutants in arginine metabolism did not include this mutant (Margolis et al., 2023) and to our knowledge no other study had tested an STM4351 mutant in mice. The *Salmonella* strain Δ STM4351 had no growth defects *in vitro* (Fig S9a) and bound to *C. albicans* similar to WT (Fig S9b). It was also not responsive to *C. albicans* for the parameters we tested. Expression of genes coding for arginine deiminase ArcA, the SPI1 regulator HilA and the T3SS-1 structural protein InvA was not increased in the presence of *C. albicans* (Fig S9c). In the presence of *C. albicans*, WT *Salmonella* upregulates expression of *arcA* in the cecum of mice. For Δ STM4351 we

Fig. 2g (new). Fluorescence images of STm expressing mCherry (constitutive promoter) and *Pprgh-gfp* and *C. albicans* SC5314 (brightfield) *in vitro*. T3SS+ cells are yellow due to mCherry and *Pprgh-gfp* fluorescence overlap.

Fig. S3d (new). Percentage of *Pprgh-gfp*+ cells in the presence of *C. albicans* SC5314 after incubation for 2h.

did not observe an upregulation. We instead observed a trend for lower expression of this gene, indicating that *Salmonella* likely does not metabolize arginine in the gut (Fig. 5h). Importantly, the strain colonized small intestine, cecum, and feces, and disseminated to spleen and liver to the same level as the WT (Fig. 5i and Fig. S9d). The levels for WT colonization can be found in Fig. 1e and S1c-e. Curiously, $\Delta STM4351$ caused a lower inflammatory response (Fig S9e) yet colonized and disseminated without any defect.

In the presence of *C. albicans*, $\Delta STM4351$ cecum colonization and dissemination to the spleen was slightly but not significantly elevated. As rendering *Salmonella* unresponsive to arginine prevents the postulated feed-forward loop of increased T3SS-dependent interaction with *C. albicans* and thus arginine production, we did not see a change in inflammatory gene expression during infection with $\Delta STM4351$ alone compared to $\Delta STM4351$ and *C. albicans* (Fig S9e). Arginine uptake by *Salmonella* is thus enhancing virulence of *Salmonella* in mice.

Fig. S9a-e (new). Characterization of *Salmonella* $\Delta STM4351$ for (a) *in vitro* growth, (b) binding to *C. albicans*, (c) gene expression in the presence of *C. albicans in vitro*, (d) gut colonization and dissemination in mice, (e) gene expression in the cecum.

Fig. 5g-i (new). During co-infection of mice, *C. albicans* does not (h) increase *arcA* expression in $\Delta STM4351$, or (i) cecum colonization or dissemination to the spleen.

Limiting host uptake of arginine (Lysine supplementation)

As a complementary approach to limiting arginine utilization by *Salmonella*, we sought to limit arginine uptake by the host. Lysine is known to competitively inhibit arginine uptake via cationic amino acid transporters (Kadirvel & Kratzer, 1974; Singh et al., 2012). On the other hand, it does not inhibit arginine uptake by *Salmonella*, as *Salmonella* possesses the specific arginine transport system artPIQM-artJ that is not transporting lysine (Torres Montaguth et al., 2019) and STM4351, which is not involved in lysine transport (Stamp et al., 2011).

We used 4 groups of mice infected/treated with the following (Fig 5j): 1) *Salmonella*, 2) *Salmonella* and lysine, 3) *Salmonella* and *C. albicans*, 4) *Salmonella*, *C. albicans*, and lysine. Competitive inhibition of arginine uptake of the host by excess lysine resulted in partial amelioration of *C. albicans*-enhanced virulence of *Salmonella*. While the presence of *C. albicans* still enhanced *Salmonella* cecum colonization in the presence of lysine, the magnitude of this enhancement was decreased (Fig 5k). Dissemination to the liver was significantly reduced, similar to infection with *Salmonella* alone (Fig S9h), while

dissemination to spleen was also reduced but not significantly (Fig S9h). Importantly, addition of lysine greatly increased inflammatory gene expression when mice were infected with *Salmonella* and *C. albicans*. Expression levels of genes such as *Il17*, and *Cxcl1* were similar to infection with *Salmonella* alone, others were trending higher, such as *Tnfa* and *Nos2* (Fig 5l), while *Ifng* and *Il10* did not change (Fig S9i). Inhibiting arginine uptake by the host thus ameliorates *C. albicans* enhanced virulence of *Salmonella*.

Overall, our results from experiment in which we limited arginine uptake by *Salmonella* or limited arginine uptake by the host showed intermediate phenotypes. This suggests that both the effect of arginine on the host as well as its effect on virulence gene expression in *Salmonella*, is important to enhance *Salmonella* colonization and dissemination in the presence of *C. albicans*.

2. Related to the above point, could Candida/arginine be acting by changing microbiota composition or factors produced by the microbiota? Is *Salmonella* the main or only consumer of Candida-derived arginine (potentially due to its physical association)? It would be helpful if the authors can find a way to increase confidence that the arrow coming back out of Candida is really directly connecting back to *Salmonella*.

The reviewer raises an important question. In our study of the three-way-interaction between *Salmonella*, *C. albicans*, and the host, we thus far had ignored the bacterial microbiota and its contribution to the observed phenotype was therefore unclear. We agree with the reviewer that a likely explanation for the effect of *C. albicans*-produced arginine on *Salmonella* is higher consumption due to physical association (see also SPI-1 reporter images in response to point #1). However, many species in the gut have the capability to catabolize arginine in addition to *Salmonella*. To elucidate whether changes in the bacterial microbiota could be contributing to the phenotypes we observed in mice, we sequenced the microbiome of conventionally raised mice infected with *Salmonella* or *Salmonella* and *C. albicans*. Additionally, we performed experiments with germ-free mice and mice with a limited microbiome.

Germ-free mice do not have a microbiota. In these mice, the presence of *C. albicans* (Fig S5a) increased *Salmonella* cecum colonization (Fig S5c), but we did not see an increased

Fig. S5a-d and i (new). (a) Germ-free mice were colonized with *C. albicans* and 24h later infected with *Salmonella*. (b-d) Colonization levels of (c) *Salmonella* and (d) *C. albicans*. (i) Arginine levels in feces from germ-free mice 24h after infection and from Altered Schaedler Flora (ASF)-colonized mice 48h post infection. UI=uninfected.

dissemination to spleen and liver (Fig S5c). Experiments in germ-free mice are limited to 24h, as mice succumb to *Salmonella* infection rapidly. Extended experiment time might have resulted in increased dissemination, but it is unfortunately not possible to test in this model. Infection of GF mice with *Salmonella* resulted in greatly reduced arginine levels (frequently below the detection limit) compared to uninfected mice (UI) with levels of ~450nmol/g (Fig. S5i), indicating *Salmonella* catabolism of this amino acid (or increased host uptake). In the presence of STm and *C. albicans*, levels in some mice were slightly higher, but still below the level of uninfected mice (Fig. S5i).

Altered Schaedler Flora (ASF) colonized mice have a limited microbiota of only 8 member species. In ASF-colonized mice (Fig. S5e-h), presence of *Candida* increased *Salmonella* cecum colonization similar to GF mice (Fig. S5g). Here, we also observed significantly increased dissemination of *Salmonella* to the liver and spleen (Fig. S5g). Arginine levels in uninfected ASF mice were ~100nmol/g and thus lower than the ~450nmol/g in GF mice (Fig S5i). This indicates that the microbiota is metabolizing arginine. Arginine levels did not change upon *Salmonella* infection or with *Salmonella* and *C. albicans* (Fig S5i, see image above).

Fig. S5e-h (new). (a) ASF mice were colonized with *C. albicans* and 24h later infected with *Salmonella*. (e-g) Colonization levels of (c) *Salmonella* and (h) *C. albicans*.

Conventionally raised mice. We also sequenced the microbiome of conventionally raised mice to determine whether microbiome composition changes in response to *C. albicans*-derived arginine could account for phenotypes we observed.

PCoA plots of ITS sequencing data show clear clustering of *C. albicans* colonized mice compared to uninfected and *Salmonella* infected mice (Fig S5j). Plotting of relative abundance highlights the dominance of *C. albicans* in mice that received it (Fig S5k). PCoA plots of 16S data show a clear separation of uninfected and *Salmonella*-infected animals. However, *Salmonella*-infected animals did not further separate based on whether they were colonized with *C. albicans* or not (Fig S5j). This is also visualized in Fig. S5k, showing similar relative abundances of the same species across groups. *Salmonella* infection therefore changes the bacterial microbiome but the presence of *C. albicans* does not result in any further changes, regardless of whether *C. albicans* can produce arginine.

Our data from germ-free mice, ASF mice and conventionally raised mice highlight that shifts in the bacterial microbiome are an unlikely cause for the phenotypes we observed. Combined with data provided for this reviewer's point #1, we feel confident that the arrow in our model in Fig S10 pointing to *Salmonella* is accurate also in an *in vivo* setting.

Fig. S5j-k (new). Fecal samples of uninfected mice or mice infected with *Salmonella*, *Salmonella* and *C. albicans* WT, *C. albicans arg4Δ/Δ* or *ARG4* complemented strain were sequenced. (j) PCoA plots of ITS and 16S sequencing data. (k) Relative abundance of fungal orders (ITS) or bacterial orders (16S).

3. The authors were unable to detect alterations in arginine levels in the cecum. Could this be due to timing or perhaps intracellular vs extracellular levels?

We were curious about this as well and thus performed more experiments to address this question. Samples from the lumen of the cecum 24h post co-infection showed no changes in arginine levels (Fig S3h). Additional data from samples collected at 48h post infection also confirmed our previous findings of no change in arginine levels (Fig S4h). We also sent samples from germ-free and ASF mice for analysis, as discussed related to this reviewer's point #2 (Fig S5i). In germ-free mice, we found that the ~450nmol/g arginine detected in uninfected mice was rapidly (within 24h) reduced to below the detection limit in most mice

Fig. S3h and S4h (new). Arginine concentration in the cecum of mice. (g) 24h post infection. (h) 48h post infection.

Fig. S5i (new, copied from response to point #2). Germ-free or ASF-colonized mice were colonized with *C. albicans* and 24h later infected with *Salmonella*. (i) Arginine levels in feces from germ-free mice 24h after infection and from Altered Schaedler Flora (ASF)-colonized mice 48h post infection. UI=uninfected

(Fig S5i, line for UI=uninfected compared to STm). In ASF mice, arginine levels did not significantly change upon *Salmonella* infection or infection with *Salmonella* and *C. albicans*. We agree with the reviewer that intracellular concentrations of arginine might be changing. However, we are unaware of any technique that would allow us to conclusively determine intracellular abundance of a de novo in situ synthesized amino acid in only one specific species present in a complex microbiome.

In summary, we can detect arginine abundance changes in very simple systems, i.e. uninfected germ-free mice versus infected germ-free mice. However, as soon as there are consumers of arginine present, changes in arginine flux in the system cannot be adequately determined by measuring absolute metabolite levels. We therefore rely on gene expression changes in *Salmonella* (Fig. 2k) and *C. albicans* (new Fig. S4c) as indicators whether consumption or production of arginine is occurring.

4. I was confused by the schematic in fig 3c, where at 24h, there is a time point for "FS and ig C.a.". Did the authors give a second gavage of candida to these animals? If so, any specific reason?

Yes, the reviewer is correct. We did administer a second dose of *C. albicans* to these animals, as we did not know how a *C. albicans* arginine auxotroph would colonize the gut. We wanted to exclude potential colonization defects of the *Candida arg4* mutant that would limit interpretation of our results. We have now repeated the experiment giving a single dose of *Salmonella* and *C. albicans* without an additional dose of *C. albicans*. At the request of Reviewer #3, we also included the revertant *ARG4* strain. Results of these new experiments replicate the results of the initial experiments with the additional dose of *C. albicans* (updated Fig 3d).

Fig. 3d (updated) Mice were infected with *Salmonella* and *C. albicans* WT, *C. albicans arg4Δ/Δ* or the complemented ARG4 mutant. CFU counts were enumerated 48h post infection.

REFEREE #2:

Referee expertise *Salmonella* pathogenesis and the microbiome

In this exciting manuscript, Jaswal et al describe a novel mechanism of how commensal yeast, as part of the gut microbiome, interacts with a bacterial pathogen. The initial observation is that oral infection of mice with *Salmonella* Typhimurium results in blooms of *Candida* in the gut, and *Candida* blooms correlate with increased disease severity. This is recapitulated through experimental *Candida*-*Salmonella* co-infection in mice. Using an in vitro culture system, the authors investigate potential interactions between *Candida* and *Salmonella*, and find that *Salmonella* binds to yeast using type 1 fimbriae. Furthermore, they discover that a secreted molecule, later to be revealed to be L-arginine, enhances *Salmonella* invasiveness and expression of the invasion-associated type three secretion (SPI1). A *Candida* mutant unable to produce arginine is unable to increase *Salmonella* virulence in cell culture and in mouse models. Curiously, production of the secreted effector protein SopB was necessary for *Salmonella* to induce arginine release by *Candida*. Furthermore, *Candida* colonization and L-arginine supplementation reduce inflammatory responses towards *Salmonella*, and result in increased dissemination.

This study sheds light on to a very understudied aspect of the microbiome, the mycobiome, and its effect on disease processes. As such, the discovery that *Candida* and *Salmonella* interact metabolically through L-arginine cross-feeding is timely, exciting, and markedly extends our knowledge on the gut microbiome. Most exciting to me is the fact that *Salmonella* appears to be using a type three secretion system effector to manipulate *Candida*; to my knowledge, this is the first evidence that non-mammalian cells are targeted by the *Salmonella* type three secretion system. This observation alone should open up new avenues of research. The overall conclusions are well-supported by the data. The manuscript is well written and easy to follow for a general audience.

There are some minor concerns regarding the detailed mechanism(s) that may require the authors' attention.

We are delighted that the reviewer is as excited about our study as we are. We thank the reviewer for their positive assessment of our manuscript and address their minor concerns in the comments below.

1. The experiment shown in Fig. 3 demonstrates that arginine production by *Candida* is necessary to enhance *Salmonella* virulence. Based on Fig 2, the authors argue that L-arginine is a cue for the invasion-associated type three secretion system, while Fig. 5 suggests that arginine supplementation changes inflammatory responses. This raises the question as to whether the phenotype reported in Fig. 3 is due to altered virulence gene regulation, or due to altered immune responses. How does arginine influence *S. Tm* infection in vivo, directly via gene regulation of virulence factors, or indirectly via the host response? While this concern does not chip away from the conceptual novelty outlined above, this key aspect of the mechanism still needs to be resolved.

Thank you very much for your important comment. Referee 1 raised similar questions, which we addressed in detail above. As the reply is extensive, we opted to not copy the response here and kindly refer you to the response to Point#1 by referee 1 (at the top of this document).

2. Along these lines, it would be important to show that SopB is required for inducing L-arginine production by *C. albicans* in the mouse model. Since L-arginine levels do not substantially change (the authors argue that local levels might), the appropriate readout might be *S. Tm* colonization. The prediction would be that in the absence of SopB, the presence of *Candida* or the Arg4 mutant would not influence invasiveness and colonization (see Fig 3d,e). Similarly, the authors should evaluate as to whether type one fimbriae are required for the in vivo phenotype.

We thank the reviewer for the suggestion of this important control experiment. We performed an *in vivo* experiment in which we did exactly what the reviewer proposed and infected mice either with *Salmonella ΔsopB* or *Salmonella ΔsopB* and *C. albicans*. As expected, the presence of *C. albicans* did not increase colonization or dissemination of *Salmonella ΔsopB* (Fig 4h). Similarly, the presence of the *C. albicans arg4Δ/Δ* mutant did not alter *Salmonella ΔsopB* colonization or dissemination. Interestingly, *C. albicans* colonization in the cecum (Fig S6e) was higher in the presence of the *Salmonella ΔsopB* mutant than during single colonization or co-infection with WT *Salmonella* (Fig S1f in the manuscript), indicating that SopB activity might be detrimental for *C. albicans* over long co-colonization times.

Fig. 4h (new). Presence of *C. albicans* WT or *arg4* mutant does not lead to increased cecum colonization or dissemination of *Salmonella*.

Fig. S6d,e (new). (d) Presence of *C. albicans* WT or *arg4Δ/Δ* mutant does not lead to changes in fecal or small intestinal colonization of *ΔsopB* *Salmonella*, or dissemination to the liver. (e) *C. albicans* colonization in feces and cecum in the presence of *ΔsopB* *Salmonella*.

To address the requirement for type I fimbriae (T1F), we have performed the suggested experiment with the *Salmonella Δfim* mutant in mice. Unfortunately, T1F are important for adhering to host tissues. In our hands, *Salmonella Δfim* colonized mice very poorly. In the two experiments whose results we combined and appended in Fig R2, we gavaged the mice with 10-fold more *Salmonella Δfim* than WT *Salmonella* but the *Δfim* mutant colonized feces 1000-fold and the cecum 100-fold less than *Salmonella* WT. *C. albicans* presence did not enhance *Salmonella* colonization. This could mean that T1F are important for binding of *Salmonella* to *C.*

albicans in vivo. However, we opted to not include this data in the manuscript, as the colonization defect of this mutant is very apparent. However, we are happy to add the data to our manuscript if the reviewer considers this addition worthwhile.

[REDACTED]

REFEREE #3

Expertise: *C. albicans* pathogenesis and commensalism

The manuscript by Jaswal et al examines interactions between *Salmonella* and *Candida* cells in the gastrointestinal tract. The authors reveal that *Candida* can exacerbate infection by *Salmonella*, and this involves the secretion of arginine from fungal cells which upregulates the Type 3 Secretion System (T3SS) in *Salmonella* and results in increased host invasion. *Salmonella* cells also enhance arginine secretion by *Candida* cells (as T3SS introduces the SopB effector protein into *Candida* cells), thereby effectively enhancing their own infectivity by this mechanism.

The paper is interesting in providing novel mechanistic insights into bacterial-fungal crosstalk in the gut, and the discovery that fungal-produced arginine can impact bacterial pathogenesis (as well as gut inflammation) is particularly intriguing. There were several areas, however, which were underdeveloped and should be addressed:

Major points:

1. The in vivo interaction data in the gut not overly convincing (Fig. 1f). There is really only one example where there appears to be a direct interaction between a bacterium and a fungal cell. Is there additional evidence (microscopic or otherwise) that these species physically interact in the gut? Is this interaction only occurring in the cecum or in other GI organs?

The reviewer makes a valid argument and we aimed to address this concern in two different ways. First, we took more images and replaced the original figure in the manuscript. We also realized a small error occurred in the description, as the figure was not from the cecum but instead from the colon (now featured in Fig S1g). We now show images of the cecum (Fig 1f), the site of highest *Salmonella* colonization boost by *C. albicans* at 6h post infection. Numerous *Salmonella* are in very close proximity to *C. albicans* cells (Fig 1f).

f
Fig. 1f (new). Mice were co-infected with *Salmonella* and *C. albicans*. Thin sections of the cecum were stained with antibodies against the *Salmonella* O antigen and Alexa Fluor 594 secondary antibody. They were additionally stained with FITC-labeled antibody against *C. albicans*.

We also quantified interactions between *Salmonella* and *C. albicans* in multiple images but realized that this analysis is inherently subjective. We therefore did not consider this analysis suitable as an unbiased proof for interaction. Instead, we focused on consequences of *Salmonella* and *C. albicans* interaction. *In vitro* we showed that ARG4 expression is a result of physical interaction of *Salmonella* and *C. albicans*. We therefore determined ARG1 and ARG4 expression in *C. albicans* during single colonization or during co-infection with *Salmonella*. ARG1 expression was not significantly increased but ARG4 expression was significantly increased in the presence of *Salmonella*. This indicates that *Salmonella* interacts with *C. albicans* and induces the expression of arginine biosynthesis genes *in vivo*.

c
Fig. S4c (new). Mice were colonized with *C. albicans* or co-infected with *Salmonella* and *C. albicans*. Cecum samples were collected at 6h post infection to recover high enough numbers of *C. albicans* for qRT-PCR analysis of ARG1 and ARG4 expression in *C. albicans*.

In addition to cecum and colon, we also aimed to visualize interaction of *Salmonella* and *C. albicans* in the terminal ileum. However, colonization levels of both *Salmonella* and *C. albicans* are so low at this site that we could not image more than a couple *Salmonella* in each section. We are aiming to develop tissue clearing and 3D imaging to process larger areas of the gut to address interactions in this organ in the future.

Are filamentous *Candida* cells also present in the gut and interacting with *Salmonella* cells under these conditions (or *in vitro*)? These are key questions to be addressed.

In vitro, *Salmonella* is binding to both yeast and hyphal form of *C. albicans* (Fig S2f). To our knowledge, filamentous *C. albicans* in the gut are primarily observed in models that continuously administer antibiotics to the mice. In our model, we only administer a single dose

of streptomycin 24h before infection. In this model, we observe *C. albicans* primarily in the yeast form in the gut (see also Fig 1f). We do not see long hyphae, but we occasionally detect *C. albicans* cells with short germ tubes (Fig R5). Researchers are now beginning to study *C. albicans* commensalism in the gut in the absence of antibiotics, but we did not find any relevant images in the literature to compare our findings. However, our *in vitro* experiments show that it is highly likely that *Salmonella* would bind to both yeast and hyphal form of *Candida* in the gut.

f

Fig. S2f (new). *C. albicans* (primed for hyphal formation by growing at 37°C) and *Salmonella* were co-incubated for 2h in LB.

2. For the experiment using *C. albicans* strain 529L in Fig. S1e it should be explicitly stated in the text and legend that antibiotics were not used in those experiments.

Thank you for the suggestion. It is an important point to highlight and we now mention in both the text and the legend that no antibiotics were used. The figure legend now states: "STm colonization in CBA/J mice infected with STm or STm and *C. albicans* 529L in the absence of antibiotics for 9d p.i."

Also, the levels of *Salmonella* in these mice only reached 10²-10³ CFU/mg of cecal tissue which is likely why they did not disseminate (compared to 10⁴-10⁵ in the antibiotic models) yet this difference is not mentioned/discussed?

We now mention that the lower colonization level of *C. albicans* in this model is likely the reason for why *C. albicans*, contrary to antibiotic treated models, does not disseminate. The statement now reads: "However, contrary to antibiotic-treated models, *Salmonella* dissemination to peripheral organs was not increased, likely due to the lower colonization level of *Salmonella* in the cecum."

I also did not see the data showing that peak inflammation is reached at 9-10 days in this model?

The reviewer is correct that we do not show data to support the statement "peak inflammation". This time point and statement was based off examples in the literature. These show that CBA/J mice develop high inflammation in mice day 7-10 post infection with *Salmonella* WT IR715, the same that we use in our study (Spiga et al., 2017). We are now citing this paper. Appended below is a comparison of inflammation in C57BL/6 mice at 48h post infection with CBA/J mice at 9 days post infection (Fig R6). Inflammation levels are comparable in these two models.

Fig. R6. Expression of inflammatory genes in cecum tissue determined by qRT-PCR. C57BL/6 mice at 2 days and CBA/J mice at 9 days post infection.

Does 529L also produce arginine in the presence of *Salmonella*?

Thank you for your comment, which helped to further increase the rigor of our study. We confirmed using qRT-PCR that *Salmonella* indeed induces arginine biosynthesis in *Candida albicans* strain 529L. Data for induction of ARG1 and ARG4 can now be found in the manuscript in Fig S4b.

Fig. S4b (new). Expression of ARG1 and ARG4 in *C. albicans* strain 529L in the presence or absence of *Salmonella*.

3. For the sedimentation data in Fig. 2b, are the *C. albicans* cells filamenting in these assays or are they completely in the yeast state? Does filamentation influence interactions with *Salmonella* cells? The same question is true for the microscopy data in Fig. 2c? And were the sedimentation assays in each panel done on the same day? (as the WT + Ca data points in Fig. S2d looks identical to that in Fig. 2b).

Thank you for your clarifying questions. We are sorry that this was not more clearly stated. *C. albicans* was grown at 37°C for 16h for these assays. Most cells were in yeast form, but we did observe an occasional short hypha or pseudohypha under these conditions (Fig R7a). We repeated the sedimentation assays with *C. albicans* grown at 30°C and sedimentation rates were the same (Fig R7b). We did not detect any differences in the binding of *Salmonella* to yeast or hyphal cells of *C. albicans*.

The reviewer is correct in their assumption that the assays were performed in parallel with one WT control and multiple mutants, but not all were shown in the main Fig. 2b. The WT data points from Fig. 2b were repeated in figure S2d (now Fig S2e) to make comparisons easier for the reader. The data duplication was indicated in the figure legend, but we added appropriate justification for why they are shown again. We now mention “Assays were performed with multiple mutants depicted in Fig 2b and FigS2e and one STm WT control in parallel. For easier comparison, STm WT and Δ fim values from Fig. 2b were also plotted in this graph”.

Fig. R7. (a) Image of *C. albicans* grown over night at 37°C. [REDACTED]

4. The authors show that mannose interferes with fungal-bacterial interactions but it would be relevant to test other sugars or fungal cell wall components to test the specificity of this inhibition. And does adding mannose to the drinking water of mice influence the STM phenotype (while realizing that this may be confounded by other effects of mannose on the GI environment?)

As Type 1 fimbriae (T1F) are generally referred to as mannose-binding or mannose-sensitive lectins, we only tested mannose in our initial experiments. However, the reviewer question did prompt us to dive into the extensive literature on Type 1 fimbriae. Indeed, other sugars can bind to T1F, albeit with lower affinity. Fructose binds 15-fold less and glucose 4000-fold less compared to mannose (Bouckaert et al., 2006). Sugars such as pentoses do not inhibit the binding. We therefore tested the hexoses glucose and fructose, as well as the pentose arabinose in our *Salmonella-Candida* binding assay. Glucose and fructose inhibited the binding similar to mannose, whereas arabinose did not inhibit the binding (Fig R8).

We initially considered performing the mouse experiment that the reviewer suggested and add mannose to the drinking water of mice. Our literature search, however, revealed that such an experiment would have too numerous confounding factors to yield interpretable results, as the reviewer also acknowledged.

1) Mannose supplementation was shown to alter the bacterial microbiome composition (Sharma et al., 2018) and significantly reduce *Salmonella* colonization (Oyofe et al., 1989).

2) Mannose supplementation was shown to reduce *Candida* colonization.

These factors alone would have still made it worthwhile to study the effect of mannose in mice. However, we decided to not pursue mannose supplementation because of a third important point.

3) The supplementation of 2% mannose in the drinking water resulted in only 6 μmol mannose per gram of feces in mice (Sharma et al., 2018). This is a 20-fold reduction of the added mannose during transition through the GI tract. Given the changes in the bacterial microbiome composition after addition of mannose, most of it is likely metabolized by the bacterial microbiota. We tested if the remaining amount, the equivalent to roughly 0.1% mannose is sufficient to inhibit binding. Unfortunately, our data showed that this remaining amount is not sufficient to inhibit binding of *Salmonella* to *Candida*. We therefore did not pursue this experiment.

However, we did perform an experiment with the T1F-deficient Δfim *Salmonella* strain (see also reviewer #2) that does not bind to *C. albicans*. This experiment therefore addresses the question the reviewer asked. Colonization of this Δfim strain did not increase in the presence of *Salmonella* (Fig R2, copied below). However, this strain did not colonize mice well, even though mice were gavaged with 10-fold higher amount of *Salmonella* compared to WT *Salmonella*. Fimbriae are necessary for *Salmonella* to bind to host tissues, which is the likely explanation for the low colonization of this mutant.

[REDACTED]

[REDACTED]

5. In some places the statistical comparisons between experimental groups should include additional comparisons. For example, in Fig 2e, is there a significant difference between STm+Ca co-culture and STm fim+Ca groups?

Thank you for alerting us to this important but missing comparison. It has now been added and the difference is significant.

Similarly, later on in Fig. S5b, is there a significant difference in expression of *invA* between STm+Ca and STm + Ca(*arg4*) groups?

This comparison has also now been added. The difference in *invA* expression is statistically significant (now Fig S4d).

Also in Fig S5c, is Ca(*arg4*) or Ca(+ARG4) significantly different from the yeast only control for sedimentation?

We performed one sample t tests and the difference is significant over yeast only. The results of the statistical analysis is now depicted in Fig S4e.

6. The authors use *Candida* strains ATCC and SC5314 extensively throughout the paper but in some places use ATCC and other places use SC5314. For example, in Fig. 2 the invasion assays are done with ATCC and the adherence assays with SC5314. While the use of multiple *Candida* strains is a strength of the manuscript, it does seem necessary to perform key experiments with both strains to show they have the same phenotypes. Does SC5314 also increase STm invasion similar to ATCC?

Yes, indeed both *C. albicans* strains increase invasion of *Salmonella* into epithelial cells. The requested data were shown in supplementary figures S2a and S2b of the original manuscript (now Fig S2b and S2c). We used two *C. albicans* strains as often as possible. Other WT strains were not used in experiments where *C. albicans* mutants were employed, as the mutants were made in SC5314. However, key experiments, such as binding to *Salmonella* (Fig. 2b SC5314, Fig. S2e ATCC and 529L), *Salmonella* invasion into epithelial cells in the presence of *C. albicans* (Fig. S2b,c SC5314 and ATCC), expression of *ARG1/4* in the presence of *Salmonella* (Fig. 3a SC5314, Fig. S4b 529L), and reduction of the inflammatory response (Fig. 5b ATCC, Fig.

Fig. S2b,c. Invasion assay with *C. albicans* strains ATCC and SC5314

S7b SC5314) is featured for two *C. albicans* strains. In this resubmission we aimed to make this as clear as possible in the figure legends.

And does ATCC show the same physical interactions with STm cells as SC5314? (The authors do show data with both *Candida* strains for Fig. S2j but not for most experiments).

As requested, we performed an additional assay and now show that binding of the ATCC strain to *C. albicans* is similar to binding to SC5314. We also added sedimentation data for 529L to further increase rigor. We updated Fig S2e with this data.

Fig. S2e. Sedimentation assay with *C. albicans* strains SC5314, ATCC and 529L and *Salmonella*.

7. In Fig 3 the authors indicate that the Stm + Ca(arg4) mutant does not increase Stm colonization (Fig 3) and invasion. But is the arg4 mutant substantially different from the WT Ca in Fig 3D? Can the authors include the ARG4 complemented strain for the in vivo experiment as well as the in vitro data? And even the WT Ca strain did not significantly increase STm levels in the spleen in this data set (no data for the liver was shown) which does raise the question about the scale and reproducibility of the *Candida*-enhanced STm phenotype.

Thank you for these important comments on rigor and reproducibility.

We performed additional experiments to include the ARG4 complemented strain in our experiments. As shown in Fig. 3d, the complemented strain replicated the phenotypes obtained with WT *C. albicans*. We also added the requested comparison to the graph.

The reviewer was correct in their observation that the dissemination phenotype in Fig. 3 was not as strong as we observed in experiments in Fig. 1. We shared the reviewer's concern about this apparent amelioration of the original phenotype. The liver data was shown in Fig S5e (now Fig S4f) but there was not a significant difference between *Salmonella* and *Salmonella* + *C. albicans*. We therefore performed additional experiments and were able to conclusively show

Fig. 3d (updated) Mice were infected with *Salmonella* and *C. albicans* WT, *C. albicans* arg4Δ/Δ or the complemented ARG4 mutant. CFU counts were enumerated 48h post infection.

that the enhancement of *Salmonella* dissemination by presence of *C. albicans* remained reproducible. As mentioned in the Materials and Methods, we had to adjust our *Salmonella* infectious dose after about half of the experiments for the manuscript were performed. Mice were getting reproducibly sicker from single infection with *Salmonella* than we had observed in 8 years performing *Salmonella* mouse experiments at UIC. This was likely due to differences in the mice shipped from Jax, as other researchers experienced differences in their infection models as well (personal communication). A 100-fold lower dose of *Salmonella* seemed to overall replicate the *Salmonella* single infection parameters in these mice that the original higher dose used to achieve. With this *Salmonella* (1×10^7 cfu) and *C. albicans* (1×10^6 cfu) dose we replicated the originally observed phenotype of higher colonization in the cecum in the presence of *C. albicans* at 48h. However, what we did not sufficiently appreciate was that the lower dose of *Salmonella* resulted in delayed dissemination to the spleen and liver. We estimate that the phenotype shifted by about 12 hours (Fig. R9). On the one hand this allowed us to observe significant differences in inflammation at 48h that were previously only observed at 24h. On the other hand, dissemination was still so low at 48h that we did not observe any differences. We aimed to visualize the changes in phenotypes in Fig R9. To test our hypothesis, we performed additional experiments and took samples at 72h. The dissemination phenotype is readily apparent at this time point (Fig S1f).

Fig R9. Visualization of phenotype shift due to lower STm infectious dose

Fig. S1f (new). Mice were infected with 1×10^7 *Salmonella* and 1×10^7 *C. albicans*. Fecal samples were collected every 24h. Mice were euthanized **72h post infection** and CFU in small intestine, cecum, liver and spleen were determined.

8. On line 351, the authors state that “Upon removal of doxycycline *C. albicans* ARG1 and ARG4 genes were induced (Fig. 4h and S6h), confirming that presence of SopB in *C. albicans* results in upregulation of arginine biosynthesis.” However, there are no significant differences in gene expression in Fig. S6h and in Fig. 4h there is no comparison between the +/- dox samples for the tetO-sopB strain.

We apologize for this inaccurate description and statistical analysis. We analyzed the data again and added the required statistical comparison, which shows a statistically significant increase in expression for ARG1 and ARG4 (now Fig 4i and S6j).

9. For the data in Figure S7, the authors state that inflammatory genes are higher with the Ca(arg4) strain than the Ca(WT) strain but the statistical analysis does not directly compare these groups (only compares them to the no Candida control). A direct comparison is again required here.

We corrected this oversight on our part and added these comparisons (Fig S7b). Expression of cytokines like *Il17* and *Ifng* are significantly increased in samples of mice infected with *Salmonella* and *C. albicans arg4ΔΔ* compared to samples infected with *Salmonella* and *C. albicans* WT.

10. Does addition of arginine in the drinking water alter the levels of this amino acid in the cecum?

Thank you for this interesting and relevant question. We tested arginine levels in the cecum of mice infected with *Salmonella* or uninfected mice that were supplied oral L-arginine 48h post infection. It was our hypothesis that we would detect increased arginine levels in mice that received oral L-arginine. Interestingly, we did not observe significant changes of arginine in the cecum even after administration of 2% L-arginine in the drinking water (Fig S8h), indicating that the combined activity of the host, the microbiota (and *Salmonella* in infected animals) metabolized all orally supplied arginine.

Fig. S8h (new). Conventionally raised mice were infected with *Salmonella* or not (uninf). Mice were given 2% L-arginine with the drinking water and arginine levels in the cecum were determined 48h post infection.

We also performed an experiment with germ-free and Altered Schaedler Flora (ASF) colonized mice (8 member limited microbiota), addressing a comment of reviewer 1. This experiment shows that the presence of a microbiota reduces arginine levels in the cecum of uninfected mice (UI), from ~450nmol/g to less than 100nmol/g. In germ-free mice, infection with *Salmonella* reduces arginine levels from ~450nmol/g to below the detection limit in most mice, indicating rapid consumption of this amino acid.

Fig. S5i (new, copied from response to referee #2). Germ-free or ASF-colonized mice were colonized with *C. albicans* and 24h later infected with *Salmonella*. (i) Arginine levels in feces from germ-free mice 24h after infection and from Altered Schaedler Flora (ASF)-colonized mice 48h post infection. UI=uninfected

In light of this data, rapid consumption of arginine by the host, *Salmonella* and the microbiota, is a likely explanation for why we do not observe a change in detectable arginine in the cecum during arginine supplementation.

Does arginine only alter inflammatory responses in the cecum or also in other host tissues?

We tested cecum as the main site of *Salmonella* gastroenteritis in mice. We now performed additional experiments to include colon and ileum. We did not observe the decrease in inflammatory gene expression in the presence of *C. albicans* at these sites (Fig R10).

Fig. R10. Conventionally raised mice were infected with *Salmonella* or with *Salmonella* and *C. albicans*. qRT-PCR was performed on colon and ileum tissue harvested 48h post infection.

11. Line 446 of the discussion says “yeast cells” but should say *Candida* cells as it is not clear that *SopB* would have the same effect on *S. cerevisiae* cells.

We corrected the wording as suggested.

12. In terms of the overall model, is the decrease in inflammation thought to drive increased colonization by *Salmonella* in the cecum? And is increased invasion envisaged to be the result of both increased colonization and increased T3SS-1 expression? Clarifying these points in the model (Fig. S9) would help the reader.

Reviewer #1 and reviewer #2 had similar questions that prompted us to perform extensive experimentation to address these questions. In lieu of repeating the same information three times, we opted to address this important point in detail in response Point#1 by referee 1 (at the top of this document).

13. I was not able to access the find the raw sequence data at NCBI or project PRJNA1143068.

We apologize that this was not yet made available. The (now two) projects PRJNA1143068 and PRJNA1255633 will be available at the time of publication under the accession numbers provided. Here are the reviewer links to the data:

<https://dataview.ncbi.nlm.nih.gov/object/PRJNA1143068?reviewer=f7a5b02q47f7rnp8012v8kff8> and <https://dataview.ncbi.nlm.nih.gov/object/PRJNA1255633?reviewer=nn1I2b38lt7tosdqbbkn7bp5go>

14. It is essential to also provide catalog numbers for key resources like the antibodies used in the study.

The catalog numbers for antibodies were provided in the reporting summary. As requested, we also added these numbers as well as others to the manuscript and will rely on editorial guidance on where such information should be disclosed.

Minor points:

-How exactly was *Candida albicans* identified as the “contaminant” in the mice in Fig. 1b,c?

We identified *C. albicans* in sequencing data (Fig S1a and Table S1). Upon re-reading the sentence we realize that readers might deduce that additional identification was performed. Analysis of sequencing data for this experiment was performed months after samples were collected, so no culture of *C. albicans* could be performed. We re-stated this sentence in the revised manuscript and it now reads: “The species identified by sequencing in mice was *Candida albicans*, which is a frequent commensal of humans but not a known commensal of mice.”

-In many cases it is says that *Salmonella* and *C.albicans* are used in a 10:1 ratio but it is not clear which is in excess – this should be clarified throughout.

We have now clarified this throughout the manuscript.

- The colors used in many places are confusing. For example, in Fig S1, the colors used for *C. albicans* cells and for *Salmonella* cells are very similar in some of the panels (light gray v. dark gray). Fig. S1a,b has no key (although it is assumed to be the same as in Fig. S1c?)

Thank you for this feedback. We changed the color for *C. albicans* in Fig S1. The omittance of the key for S1a,b was an oversight that is now corrected.

- a schematic would for fig2a to show how bacteria/fungal cells were cultured and tested would be a benefit.

Thank you for this suggestion. We added a scheme depicting culture and testing of fungal and bacterial cells. It is appended to the right and can also be found as Fig S2a.

- sentence starting on Line 191 is confusing “In either case.....”

This has been corrected and now reads: “*C. albicans* did not increase invasion of *Salmonella* Δ *fim* or *Salmonella* WT in the presence of mannose, indicating that *Salmonella* binding to *C. albicans* is required to increase *Salmonella* invasion into epithelial cells (Fig. 2d and S2h).”

REFERENCES

(only for this Response to Reviewers document)

- Bouckaert, J., Berglund, J., Schembri, M., De Genst, E., Cools, L., Wuhrer, M., Hung, C. S., Pinkner, J., Slattegard, R., Zavialov, A., Choudhury, D., Langermann, S., Hultgren, S. J., Wyns, L., Klemm, P., Oscarson, S., Knight, S. D., & De Greve, H. (2006). Receptor binding studies disclose a novel class of high-affinity inhibitors of the *Escherichia coli* FimH adhesin. *The Journal of Urology*, 175(4), 1570–1570.
- Das, P., Lahiri, A., Lahiri, A., Sen, M., Iyer, N., Kapoor, N., Balaji, K. N., & Chakravorty, D. (2010). Cationic amino acid transporters and *Salmonella* Typhimurium ArgT collectively regulate arginine availability towards intracellular *Salmonella* growth. *PLoS One*, 5(12), e15466.
- Gül, E., Enz, U., Maurer, L., Abi Younes, A., Fattinger, S. A., Nguyen, B. D., Hausmann, A., Furter, M., Barthel, M., Sellin, M. E., & Hardt, W.-D. (2023). Intraluminal neutrophils limit epithelium damage by reducing pathogen assault on intestinal epithelial cells during *Salmonella* gut infection. *PLoS Pathogens*, 19(6), e1011235.
- Kadirvel, R., & Kratzer, F. H. (1974). Uptake of L-arginine and L-lysine by the small intestine and its influence on arginine-lysine antagonism in chicks. *The Journal of Nutrition*, 104(3), 339–343.
- Margolis, A., Liu, L., Porwollik, S., Till, J. K. A., Chu, W., McClelland, M., & Vázquez-Torres, A. (2023). Arginine metabolism powers *Salmonella* resistance to oxidative stress. *Infection and Immunity*, 91(6), e0012023.
- Muotiala, A., & Mäkelä, P. H. (1990). The role of IFN-gamma in murine *Salmonella* typhimurium infection. *Microbial Pathogenesis*, 8(2), 135–141.
- Oyoyo, B. A., DeLoach, J. R., Corrier, D. E., Norman, J. O., Ziprin, R. L., & Mollenhauer, H. H. (1989). Prevention of *Salmonella* typhimurium colonization of broilers with D-mannose. *Poultry Science*, 68(10), 1357–1360.
- Sharma, V., Smolin, J., Nayak, J., Ayala, J. E., Scott, D. A., Peterson, S. N., & Freeze, H. H. (2018). Mannose alters gut microbiome, prevents diet-induced obesity, and improves host metabolism. *Cell Reports*, 24(12), 3087–3098.
- Singh, K., Coburn, L. A., Barry, D. P., Boucher, J.-L., Chaturvedi, R., & Wilson, K. T. (2012). L-arginine uptake by cationic amino acid transporter 2 is essential for colonic epithelial cell restitution. *American Journal of Physiology. Gastrointestinal and Liver Physiology*, 302(9), G1061–G1073.
- Spiga, L., Winter, M. G., Furtado de Carvalho, T., Zhu, W., Hughes, E. R., Gillis, C. C., Behrendt, C. L., Kim, J., Chessa, D., Andrews-Polymenis, H. L., Beiting, D. P., Santos, R. L., Hooper, L. V., & Winter, S. E. (2017). An oxidative central metabolism enables

salmonella to utilize Microbiota-derived succinate. *Cell Host & Microbe*, 22(3), 291-301.e6.

Stamp, A. L., Owen, P., El Omari, K., Lockyer, M., Lamb, H. K., Charles, I. G., Hawkins, A. R., & Stammers, D. K. (2011). Crystallographic and microcalorimetric analyses reveal the structural basis for high arginine specificity in the *Salmonella enterica* serovar Typhimurium periplasmic binding protein STM4351. *Proteins*, 79(7), 2352–2357.

Torres Montaguth, O. E., Bervoets, I., Peeters, E., & Charlier, D. (2019). Competitive repression of the artPIQM operon for arginine and ornithine transport by arginine repressor and leucine-responsive regulatory protein in *Escherichia coli*. *Frontiers in Microbiology*, 10, 1563.

RESPONSE TO REFEREES

Referee #3:

The revised manuscript by Jaswal et al. has addressed the major concerns of this reviewer and the authors should be commended for their additional experiments and analyses that have added to the robustness of their findings. Some minor points are listed below and I am grateful that the authors have gone to considerable lengths to address the points of the three reviewers.

We thank the reviewer for their highly positive view on our resubmission and gladly answer their remaining comments below.

1. On line 169 the text states that “presence of *C. albicans* increased *Salmonella* colonization in the cecum (Fig. S1h)”. However, given that this difference is only significant at one time point (5 dpi) I would add this to the text to be transparent.

The reviewer refers to the left panel of Fig S1h, which depicts *Salmonella* levels in feces. However, we are referring to colonization levels in the cecum, which can only be determined at the end of an experiment and is depicted together with all parameters determined after necropsy in the panel on the right side of Fig. S1h. We would have liked to describe all panels in greater detail but due to the space limitations we were unfortunately unable to do so.

2. Could the authors alter the contrast in the images in Fig. 2g to show the yeast cells more clearly as these are hard to see? Also, the panel on the right seems to show that there are both T3SS-positive and -negative cells in close contact with *Candida* cells? How do the authors account for this? Another way to look at this could be to use their sedimentation assay and test if sedimenting *Salmonella* cells that are presumably in closer contact with *Candida* cells activate T3SS expression more than those cells left in suspension.

As suggested, we adjusted the contrast to show the yeast more clearly as expected. *Salmonella* populations have 10-30% T3SS-1 positive cells, dependent on the culturing conditions used. We therefore included a *Salmonella* only control for our quantification in Fig. S3d. For quantification we counted all *Salmonella* in a field of view (or portion thereof that was in focus). We do not claim that all *Salmonella* that are bound to *C. albicans* activate the T3SS. 100% activation, to our knowledge, has only been achieved with constitutively active mutant *Salmonella*. However, we do show in Fig. S3d that significantly more *Salmonella* activate T3SS-1 when *C. albicans* is present. We chose imaging on an agar pad to limit diffusion of arginine and its action on *Salmonella* cells not in vicinity of *C. albicans*. Nevertheless, there was likely still some diffusion of arginine through the agar, as also some cells not in immediate contact showed higher percentage of T3SS activation than in the *Salmonella* alone control. The diffusion of arginine in the culture medium was also the reason we opted to not perform the suggested experiment. We propose that arginine upregulates T3SS-1 in *Salmonella*. In our sedimentation assay, both *Salmonella* bound to *C. albicans* as well as *Salmonella* not bound to *C. albicans* would be exposed to arginine produced by *C. albicans*. Exposure would likely be even more uniform than on an agar pad. We therefore would not expect a difference in T3SS-1 positive cells in the two populations.

3. The authors use a yeast only control to normalize the sedimentation data but does the presence of *Salmonella* also influence the OD reading (independent of its effect on sedimentation?). This will not change the relative results of the assays but could affect how the *absolute* numbers look?

Yes, the reviewer is correct in their assessment. The influence of *Salmonella* on the OD reading can be indirectly seen in the samples with the *Salmonella* Δfim mutant (Fig. 2b). We normalized our readings to “yeast only”. The reason for why the reading for the Δfim mutant is not 100% but 50% is that *Salmonella* are present in the culture and they increase the OD reading. We tried multiple different ways of displaying the data of our assay but found displaying percent sedimentation as a positive value to be the most intuitive for readers to understand. Absolute OD values (top of the culture after incubation) are highest for *C. albicans* + Δfim (as *Salmonella* does not bind to *C. albicans* and does not sediment), intermediate for *C. albicans* alone (as no *Salmonella* is present to increase the OD), and lowest for *C. albicans* + WT *Salmonella* (as they bind and sediment quite rapidly together).

4. Line 482. I would make clearer in the text that lysine can be used as a competitive inhibitor of arginine uptake (this is inferred but not 100% clear in this sentence).

We slightly reworded the sentence to clarify this point as suggested: “*L-lysine is a competitive inhibitor of L-arginine uptake by the host gut epithelium (Fig 5j). L-lysine administration in the drinking water did not alter Salmonella single infection parameters [...]*”

5. One line 564, it could be emphasized that the activity of Arg82 / IPMK has diverged between fungi and mammalian cells? I believe that is what is being suggested here.

Yes, the reviewer is correct in that we hypothesize that the two enzymes diverged and now exhibit different functions in mammalian cells and fungal cells. We would have loved to extend our discussion of Arg82/IPMK. However, due to page limitations, we were unfortunately unable to expand on our hypothesis. If required and permitted by the journal, we could provide a supplementary discussion file.

6. The authors state that *Candida* cells are mostly in the yeast state in the gut. Given the interest in fungal morphology and gut colonization I would suggest adding this statement to the main text. Also, it is very difficult to see hyphal cells in Fig. S2f – could the green signal be turned up to better see the fungal cells?

We added the requested information. The sentence now reads: “*Imaging showed C. albicans predominantly in yeast form in the gut and confirmed that Salmonella and C. albicans interact with each other in the lumen of the cecum (Fig. 1f) and in proximity to colonic epithelial cells (Extended Fig. 1g).*” The intensity of the green channel in Fig S2f has been increased and the hypha is now more clearly visible.

7. The authors state that mice of both sexes were used. It would be helpful to note in the figure legends which sexes were used for a specific experiment and the numbers of each involved.

We are happy to provide this information. As we were limited in word counts for figure legends, we provide the requested information in the Materials and Methods section in section “Animal models”. We have now mentioned “*Both males and females are used in this study. Most experiments are conducted using female mice, with the repeat of key experiments in males to ensure reproducibility across sexes. Specifically, Fig. 1f includes 5 male and 3 female mice per group. Extended Data Figure 5a–d includes 5 females and 2 males in the STm group and 6 females and 2 males in the STm + C. albicans group. Extended Fig. 5e–h includes 3 males and 2 females per group.*”